# SoPo: Text-to-Motion Generation Using Semi-Online Preference Optimization

**Xiaofeng Tan**[1,2]     **Hongsong Wang** [1,2*]     **Xin Geng**[1,2]     **Pan Zhou**[3]

[1]Department of Computer Science and Engineering, Southeast University, Nanjing, China
[2]Key Laboratory of New Generation Artificial Intelligence Technology and Its Interdisciplinary Applications (Southeast University), Ministry of Education, Nanjing, China
[3] Singapore Management University
{xiaofengtan, hongsongwang, xgeng}@seu.edu.cn, panzhou@smu.edu.sg,

## Abstract

Text-to-motion generation is essential for advancing the creative industry but often presents challenges in producing consistent, realistic motions. To address this, we focus on fine-tuning text-to-motion models to consistently favor high-quality, human-preferred motions—a critical yet largely unexplored problem. In this work, we theoretically investigate the DPO under both online and offline settings, and reveal their respective limitation: overfitting in offline DPO, and biased sampling in online DPO. Building on our theoretical insights, we introduce Semi-online Preference Optimization (SoPo), a DPO-based method for training text-to-motion models using "semi-online" data pair, consisting of unpreferred motion from online distribution and preferred motion in offline datasets. This method leverages both online and offline DPO, allowing each to compensate for the other's limitations. Extensive experiments demonstrate that SoPo outperforms other preference alignment methods, with an MM-Dist of 3.25% (vs e.g. 0.76% of MoDiPO) on the MLD model, 2.91% (vs e.g. 0.66% of MoDiPO) on MDM model, respectively. Additionally, the MLD model fine-tuned by our SoPo surpasses the SoTA model in terms of R-precision and MM Dist. Visualization results also show the efficacy of our SoPo in preference alignment. Project page: https://xiaofeng-tan.github.io/projects/SoPo/.

## 1   Introduction

Text-to-motion generation aims to synthesize realistic 3D human motions based on textual descriptions, unlocking numerous applications in gaming, filmmaking, virtual and augmented reality, and robotics [1–4]. Recent advances in generative models [5–7], particularly diffusion models [1, 2, 8–14], have significantly improved text-to-video generation. However, text-to-motion models often encounter challenges in generating consistent and realistic motions due to several key factors.

Firstly, models are often trained on diverse text-motion pairs where descriptions vary widely in style, detail, and purpose. This variance can cause inconsistencies, producing motions that do not always meet realism or accuracy standards [15–17]. Secondly, text-to-motion models are probabilistic, allowing diverse outputs for each description. While this promotes variety, it also increases the chances of generating undesirable variations [4]. Lastly, the complexity of coordinating multiple flexible human joints results in unpredictable outcomes, increasing the difficulty of achieving smooth and realistic motion [16]. Together, these factors limit the quality and reliability of current methods of text-to-motion generation.

---

[*]Corresponding Author

39th Conference on Neural Information Processing Systems (NeurIPS 2025).

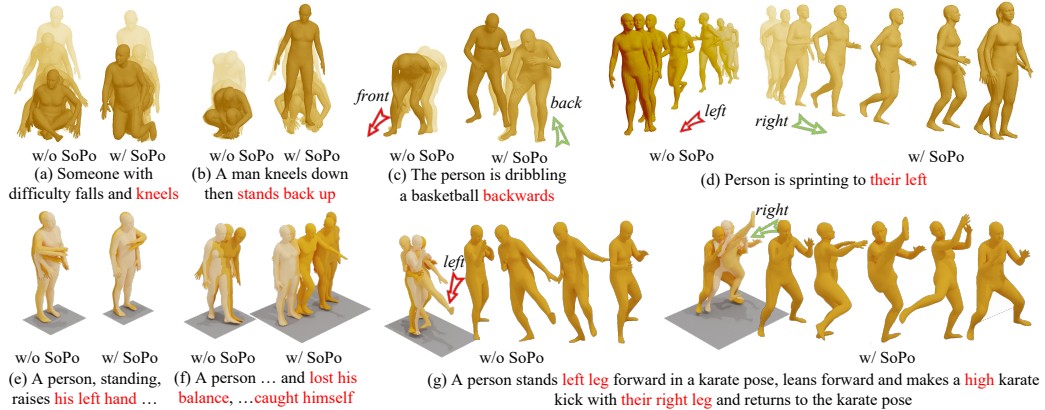

w/o SoPo  w/ SoPo
(a) Someone with
difficulty falls and kneels

w/o SoPo  w/ SoPo
(b) A man kneels down
then stands back up

front  back
w/o SoPo  w/ SoPo
(c) The person is dribbling
a basketball backwards

left  right
w/o SoPo    w/ SoPo
(d) Person is sprinting to their left

w/o SoPo  w/ SoPo
(e) A person, standing,
raises his left hand …

w/o SoPo  w/ SoPo
(f) A person … and lost his
balance, …caught himself

left  right
w/o SoPo    w/ SoPo
(g) A person stands left leg forward in a karate pose, leans forward and makes a high karate
kick with their right leg and returns to the karate pose

Figure 1: Visual results on HumanML3D dataset. We integrate our SoPo into MDM [13] and MLD [1], respectively. Our SoPo improves the alignment between text and motion preferences.

In this work, we focus on refining text-to-motion models to consistently generate high-quality and human-preferred motions, a largely unexplored but essential area given its wide applicability. To our knowledge, MoDiPO [9] is the only work directly addressing this. MoDiPO applies a preference alignment method, DPO [18], originally developed for language and text-to-image models, to the text-to-motion domain. This approach fine-tunes models on datasets where each description pairs with both preferred and unpreferred motions, guiding the model toward more desirable outputs. Despite MoDiPO's promising results, challenges remain, as undesired motions continue to arise, as shown in Fig. 1. Unfortunately, this issue is still underexplored, with limited efforts directed at advancing preference alignment approaches to mitigate it effectively.

**Contributions.** Building upon MoDiPO, this work addresses the above problem, and derives some new results and alternatives for text-to-motion generation alignment. Particularly, we theoretically investigate the limitations of online and offline DPO, and then propose a Semi-Online Preference Optimization (SoPo) to solve the alignment issues in online and offline DPO for text-to-motion generation. Our contributions are highlighted below.

Our first contribution is the explicit revelation of the limitations of both online and offline DPO. Online DPO is constrained by biased sampling, resulting in high-preference scores that limit the preference gap between preferred and unpreferred motions. Meanwhile, offline DPO suffers from overfitting due to limited labeled preference data, especially for unpreferred motions, leading to poor generalization. This leads to inconsistent performance in aligning preferences for existing methods.

Inspired by our theory, we propose a novel and effective SoPo method to address these limitations. SoPo trains models on "semi-online" data pairs that incorporate high-quality preferred motions from offline datasets alongside diverse unpreferred motions generated dynamically. This blend leverages the offline dataset's human-labeled quality to counter online DPO's preference gap issues, while the dynamically generated unpreferred motions mitigate offline DPO's overfitting.

Finally, extensive experimental results like Fig. 1 show that our SoPo significantly outperforms the SoTA baselines. For example, on the HumanML3D dataset, integrating our SoPo into MLD brings 0.222 in Diversity and 3.25% in MM Dist improvement. By comparison, combining MLD with MoDiPO only bring 0.091 and $-0.01\%$ respectively. These results underscore SoPo's effectiveness in improving human-preference alignment in text-to-motion generation.

## 2   Related Works

**Text-to-Motion Generation.** Text-to-motion generation [10, 19–21] is a key research area with broad applications in computer vision. Recently, diffusion-based models have shown remarkable progress by enhancing both the quality and diversity of generated motions with stable training [2, 11–13]. MotionDiffuse [14] is a pioneering text-driven diffusion model that enables fine-grained body control

and flexible, arbitrary-length motion synthesis. Tevet et al. [13] propose a transformer-based diffusion model using geometric losses for better training and performance. Chen et al. [1] improve efficiency by combining latent space and conditional diffusion. Kong et al. [8] enhance diversity with a discrete representation and adaptive noise schedule. Dai et al. [2] present a real-time controllable model using latent consistency distillation for efficient and high-quality generation. Despite these advances, generating realistic motions that align closely with text remains challenging. Despite significant progress in skeleton-based motion understanding achieved by unified foundational models [22, 23], these generative models still exhibit limitations in the semantic and spatial complexities understanding. ***Thus, how to enhance the generative ability by discriminative model remain necessary to explore.***

**Direct Preference Optimization.** RLHF [24] aims to align model distributions over pre-defined preference distributions under the same conditions. As a representative RL method, Direct Preference Optimization (DPO) has shown great success in large language models (LLMs) [18, 25], text-to-3D [26], and image generation [27–31], offering a promising solution to the aforementioned issue. Existing methods are broadly categorized into offline [27, 32] and online DPO [28–31]. Offline DPO trains on fixed datasets with preference labels from humans [27] or AI feedback [9]. In contrast, online DPO generates data online using a policy [31] or a reference model [29], and forms preference pairs via human [28] or AI feedback [32]. While effective in text-to-image generation, DPO methods for text-to-motion—e.g., MoDiPO [9]—remain underexplored and face challenges such as overfitting and insufficient preference gaps. More discussion about recent RL research are shown in App. D.

## 3 Motivation: Rethink Offline & Online DPO

**Preliminaries.** Here we analyze DPO in MoDiPO to explain its inferior alignment performance for text-to-motion generation. To this end, we first briefly introduce DPO [18]. Let $\mathcal{D}$ be a preference dataset which comprises numerous triples, each containing a text condition $c$ and a motion pair $x^w \succ x^l$ where $x^w$ and $x^l$ respectively denote the preferred motion and unpreferred one. With this dataset, Reinforcement Learning from Human Feedback (RLHF) [33] first trains a reward model $r(x, c)$ to access the quality of $x$ under the condition $c$. Then RLHF maximizes cumulative rewards while maintaining a KL constraint between the policy model $\pi_\theta$ and a reference model $\pi_{\text{ref}}$:

$$\max_{\pi_\theta} \mathbb{E}_{c \sim \mathcal{D}, x \sim \pi_\theta(\cdot|c)} \left[ r(x, c) - \beta D_{\text{KL}} \left( \pi_\theta(x|c) \, \| \, \pi_{\text{ref}}(x|c) \right) \right]. \tag{1}$$

Here one often uses the frozen pretrained model as the reference model $\pi_{\text{ref}}$ and current trainable text-to-motion model as the policy model $\pi_\theta$.

Building upon RLHF, DPO [18] analyzes the close solution of problem in Eq. (1) to simplify its loss:

$$\mathcal{L}_{\text{DPO}}(\theta) = \mathbb{E}_{(x^w, x^l, c) \sim \mathcal{D}} \left[ -\log \sigma \left( \beta \mathcal{H}_\theta(x^w, x^l, c) \right) \right], \tag{2}$$

where $\mathcal{H}_\theta(x^w, x^l, c) = h_\theta(x^w, c) - h_\theta(x^l, c)$, $h_\theta(x, c) = \log \frac{\pi_\theta(x|c)}{\pi_{\text{ref}}(x|c)}$, and $\sigma$ is the logistic function. When there are multiple preferred motions (responses) under a condition $c$, i.e., $x^1 \succ x^2 \succ \cdots \succ x^K$ ($K \geq 2$), by using Plackett-Luce model [34], DPO can be extended as:

$$\mathcal{L}_{\text{off}}(\theta) = -\mathbb{E}_{(x^{1:K}, c) \sim \mathcal{D}} \left[ \log \prod_{k=1}^{K} \frac{\exp(\beta h_\theta(x^k, c))}{\sum_{j=k}^{K} \exp(\beta h_\theta(x^j, c))} \right]. \tag{3}$$

When $K = 2$, $\mathcal{L}_{\text{off}}$ degenerates to $\mathcal{L}_{\text{DPO}}$. Since MoDiPO uses multiple preferred motions for alignment, we will focus on analyze the general formulation in Eq. (3).

### 3.1 Offline DPO

**Analysis.** In Eq. (3), its training data are sampled from an offline dataset $\mathcal{D}$. So DPO in Eq. (3) is also called "offline DPO". Here we analyze its preference optimization with its proof in App. C.1

**Theorem 1.** *Given a preference motion dataset $\mathcal{D}$, a reference model $\pi_{\text{ref}}$, and ground-truth preference distribution $p_{\text{gt}}$, the gradient of $\nabla_\theta \mathcal{L}_{\text{off}}$ can be written as:*

$$\nabla_\theta \mathcal{L}_{\text{off}}(\theta) = \mathbb{E}_{c \sim \mathcal{D}, x^{1:K}} \nabla_\theta D_{KL}(p_{\text{gt}} || p_\theta). \tag{4}$$

*Here $p_\theta(x^{1:K}|c) = \prod_{k=1}^{K} p_\theta(x^k|c)$ represents the likelihood that policy model generates motions $x^{1:K}$ matching their rankings, where $p_\theta(x^k|c) = \frac{(\exp h_\theta(x^k, c))^\beta}{\sum_{j=k}^{K} (\exp h_\theta(x^j, c))^\beta}$.*

Theorem 1 shows that the gradient of offline DPO aligns with the gradient of the forward KL divergence, $D_{KL}(p_{\mathrm{gt}}||p_\theta)$. This suggests that the policy model $p_\theta$ (i.e., the trainable text-to-motion model) is optimized to match its distribution with the ground-truth motion preference distribution $p_{\mathrm{gt}}$.

**Discussion.** However, since training data is drawn from a fixed dataset $\mathcal{D}$, the model risks overfitting, particularly on unpreferred samples. Due to limited annotations, text-to-motion datasets typically contain only one preferred motion group $x_c^{1:K}$ per condition $c$, making $p_{\mathrm{gt}}(\cdot|c)$ resemble a one-point distribution, i.e., $p_{\mathrm{gt}}(x_c^{1:K}|c) = 1$. In this case, minimizing $D_{\mathrm{KL}}(p_{\mathrm{gt}}||p_\theta)$ reduces to maximizing likelihood: $\min D_{\mathrm{KL}}(p_{\mathrm{gt}}||p_\theta) \Leftrightarrow \min -\log p_\theta(x_c^{1:K}|c)$.

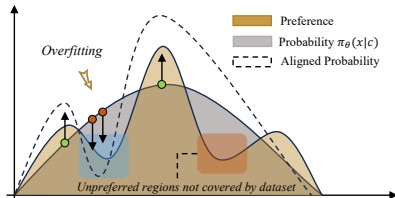

As a result, offline DPO progressively increases $p_\theta(x_c^{1:K}|c)$, widening the preference gap between preferred and unpreferred motions. As illustrated in Fig. 2, the model primarily learns from the fixed motion group $x_c^{1:K}$ for each $c$, causing the internal gap within $x_c^{1:K}$ to expand. This overfitting effect, also noted in [35], suggests that with limited unpreferred data, the model learns to avoid only specific patterns (e.g., red regions in Fig. 2) while ignoring many common unpreferred motions. Despite this limitation, the offline dataset is manually labeled and provides valuable preference information, where the gap between preferred and unpreferred motions is large, benefiting learning preferred motions.

Figure 2: Overfitting in offline DPO: green/red points are preferred/unpreferred motions; blue shows bias from fixed unpreferred data, red indicates uncovered unpreferred regions.

## 3.2 Online DPO

**Analysis.** In each online DPO training iteration, the current policy model $\pi_\theta$ generates $K$ samples for a given text $c$. A pretrained reward model $r$ ranks them by preference as $x_{\bar\pi_\theta}^1 \succ x_{\bar\pi_\theta}^2 \succ \cdots \succ x_{\bar\pi_\theta}^K$, where $x\bar\pi\theta^i$ is sampled from $\pi\theta$ without gradient backpropagation. Using the Plackett-Luce model [34], the probability of $x_{\bar\pi_\theta}^k$ being ranked $k$-th is given by:

$$p_r(x_{\bar\pi_\theta}^k|c) = \frac{\exp r(x_{\bar\pi_\theta}^k, c)}{\sum_{i=k}^K \exp r(x_{\bar\pi_\theta}^i, c)}. \tag{5}$$

Then we can analyze online DPO below.

**Theorem 2.** *Given a reward model $r$ and a reference model $\pi_{\mathrm{ref}}$, for the online DPO loss $\mathcal{L}_{\mathrm{on}}$, its gradient is:*

$$\nabla_\theta \mathcal{L}_{\mathrm{on}}(\theta) = \mathbb{E}_{c\sim\mathcal{D}, x^{1:K}} \nabla_\theta\, p_{\bar\pi_\theta}(x^{1:K}|c) D_{KL}(p_r||p_\theta), \tag{6}$$

*where $p_{\bar\pi_\theta}(x^{1:K}|c) = \prod_{k=1}^K p_{\bar\pi_\theta}(x^k|c)$ with $p_{\bar\pi_\theta}(x^k|c)$ being the generative probability of policy model to generate $x^k$ conditioned on c, and $p_\theta(x^k) = \frac{(\exp h_\theta(x_k,c))^\beta}{\sum_{j=k}^K (\exp h_\theta(x_j,c))^\beta)^\beta}$ denotes the likelihood that policy model generates motion $x_k$ with the $k$-th largest probability.*

See the proof in App. C.2. Theorem 2 indicates that online DPO minimizes the forward KL divergence $D_{KL}(p_r||p_\theta)$. Thus, online DPO trains the policy model $\pi_\theta$, i.e., the text-to-motion model, to align its text-to-motion distribution with the online preference distribution $p_r(x|c)$.

**Discussion.** We discuss the training bias and limitations of online DPO. Specifically, motions with high generative probability $p_{\bar\pi_\theta}(x_{\bar\pi_\theta}|c)$ are frequently synthesized and thus dominate the training of $\pi_\theta$. In contrast, motions with low generative probability—despite potentially high human preference—are rarely generated and scarcely contribute to training. Notably, when $p_{\bar\pi_\theta}(x_{\bar\pi_\theta}|c) \to 0$ but the reward $r(x_{\bar\pi_\theta}, c) \to 1$, the gradient still vanishes: $\lim_{p_{\pi_\theta}(x_{\bar\pi_\theta}|c)\to 0, r(x_{\bar\pi_\theta},c)\to 1} \nabla_\theta \mathcal{L}_{\mathrm{on}} = \mathbf{0}$ (see derivation in App. C.2). This highlights a key limitation: online DPO tends to ignore valuable but infrequent preferred motions, focusing instead on commonly generated ones regardless of their actual preference.

Additionally, online DPO aligns the generative probability $p_{\bar\pi_\theta}(x_{\bar\pi_\theta}|c)$ with the preference distribution $p_r(x_{\bar\pi_\theta}|c)$, leading to a positive correlation. Thus, motions with higher generative probabilities often exhibit higher preferences. However, since preference rankings are determined by a reward model, roughly half of these high-preference motions—those with lower rankings $k$ despite high scores $r(x_{\bar\pi_\theta}^k, c)$—are still treated as unpreferred. As a result, many unpreferred training motions retain considerable preference, reducing the preference gap compared to manually labeled offline datasets.

On the other hand, online DPO dynamically generates diverse motions, particularly unpreferred motions, in each iteration. This dynamic process enriches preference information and mitigates the overfitting observed in offline DPO, enabling the model to avoid the undesired patterns.

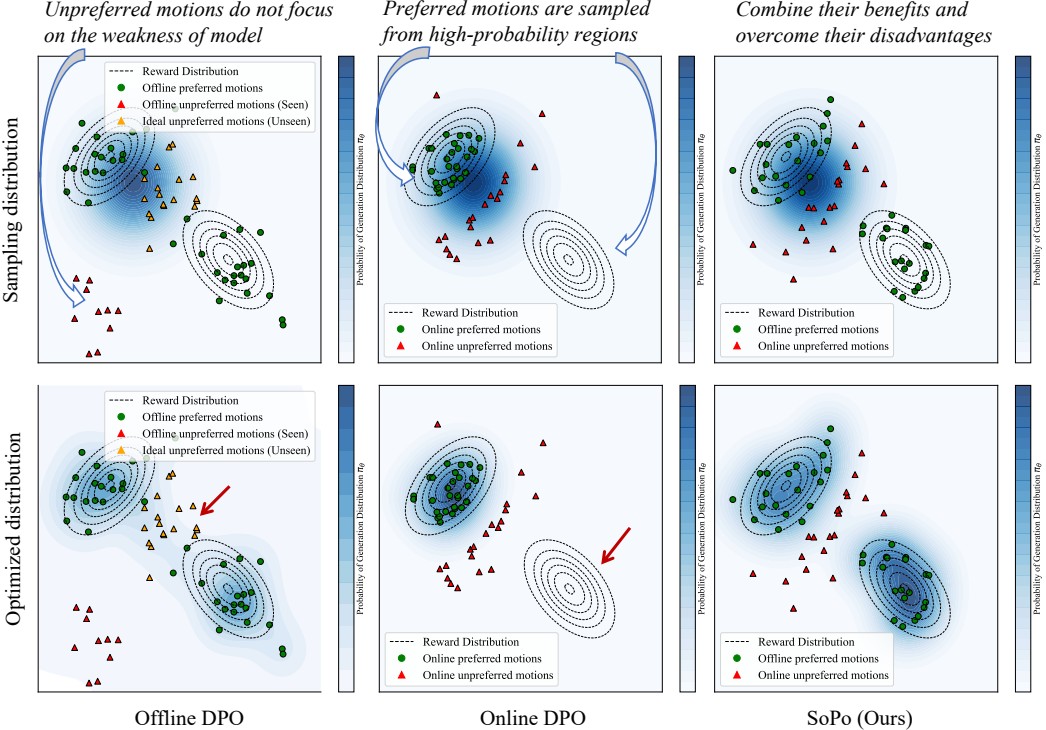

Figure 3: Comparison of offline, online DPO, and our SoPo on synthetic data. Offline DPO suffers from mining unpreferred motions with high probability, and online DPO is limited by biased sampling. Our SoPo utilizes the dynamic unpreferred motions and preferred motions from unbiased offline dataset, overcoming their advantage. Here, the blue region is the distribution of generative model.

## 3.3 DPO-based methods for Text-to-Motion

**Analysis.** DPO in MoDiPO [9] uses an offline dataset $\mathcal{D}$ that is indeed generated by a pre-trained model $\pi_p$, denoted as:

$$\begin{cases} x_{\pi_p}^w = \mathrm{argmax}_{x_{\pi_p}^{1:K} \in \bar{\pi}_p} \exp r(x_{\pi_p}^k, c), \\ x_{\pi_p}^l = \mathrm{argmin}_{x_{\pi_p}^{1:K} \in \bar{\pi}_p} \exp r(x_{\pi_p}^k, c), \end{cases} \quad \mathcal{D} = \{(x_{\pi_p}^w, x_{\pi_p}^l, c) | c \in \text{offline textural sets}\}. \tag{7}$$

For discussion, we formulate its sampled distribution as:

$$p_{\mathrm{gt}}^{Mo}(x_w, x_l | c) = \mathbb{I}((x_w, x_l, c) \in \mathcal{D}), \tag{8}$$

where the indication function $\mathbb{I}(\mathcal{E}) = 1$ if event $\mathcal{E}$ happens; otherwise, $\mathbb{I}(\mathcal{E}) = 0$.

From Eq. (7), we observe that, like online DPO, MoDiPO samples preference motions from the distribution $p_{\pi_p}(x|c)$ induced by the pre-trained model $\pi_p$. This leads to two main issues like online DPO. 1) Samples with low generative probability $p_{\pi_p}(x|c)$ but high preferences $r(x, c)$ are rarely generated by $\pi_p$ and thus seldom contribute to training, even though they are highly desirable motions. 2) As discussed in Sec. 3.2, the motions $x_{\pi_p}$ generated by $\pi_p$ typically exhibit both high generative probability and preference scores, which causes half of the preferred samples to be selected as unpreferred, skewing the model's learning process. See the detailed discussion in Sec. 3.2.

Additionally, from Eq. (8), we see that for a given condition $c$, MoDiPO trains on fixed preference data, similar to offline DPO. Consequently, MoDiPO is limited to avoiding only the unpreferred motions valued by the pre-trained model $\pi_p$, rather than those relevant to the policy model $\pi_\theta$. Thus, it inherits the limitations of both online and offline DPO, constraining the alignment performance.

# 4 Semi-Online Preference Optimization

## 4.1 Overview of SoPo

We introduce our Semi-Online Preference Optimization (SoPo) to address the limitations in both online and offline DPO for text-to-motion generation. Its core idea is to train the text-to-motion model on semi-online data pairs, where high-preference motions are from offline datasets, while low-preference and high-diversity unpreferred motions are generated online.

As discussed in Sec. 3, offline DPO provides high-preference motions with a clear preference gap from unpreferred ones but tends to overfit due to reliance on fixed, single-source unpreferred motions. In contrast, online DPO benefits from diverse, dynamically generated data but often lacks a sufficient preference gap and overlooks low-probability preferred motions. To leverage the strengths of both, SoPo samples diverse unpreferred motions $x_{\bar{\pi}_\theta}^l$ from online generation and high-preference motions $x_{\mathcal{D}}^w$ from offline datasets, ensuring a broad gap between them. Thus, SoPo mitigates the overfitting of offline DPO and the insufficient preference gaps in online DPO. Accordingly, we arrive at our SoPo:

$$\mathcal{L}_{\mathrm{DSoPo}}(\theta) = -\,\mathbb{E}_{(x^w,c)\sim\mathcal{D}}\mathbb{E}_{x^l\sim\bar{\pi}_\theta(x|c)}\log\sigma\Big(\beta\mathcal{H}_\theta(x^w,x^l,c)\Big), \tag{9}$$

where $\mathcal{H}_\theta(x^w,x^l,c)$ is defined below Eq. (2), $x^w$ is preferred motion from the offline dataset, and $x^l$ is unpreferred motion sampled from online DPO. To demonstrate the advantages of SoPo, we conduct experiments on synthetic data, as shown in Fig. 3 (Detailed experimental settings in App. A.2).

However, direct online generation of unpreferred motions from the policy model presents challenges, given the positive correlation between the generative distribution $p_{\bar{\pi}_\theta}$ and preference distribution $p_r$. Additionally, a large gap between preferred and unpreferred motions remains essential for effective SoPo. In Sec. 4.2 and 4.3, we receptively elaborate on SoPo's designs to address these challenges.

## 4.2 Online Generation for Unpreferred Motions

Here we introduce our generation pipeline for diverse unpreferred motions. Specifically, given a condition $c$, we first generate $K$ motions $\{x_{\bar{\pi}_\theta}^k\}_{k=1}^K$ from the policy model $\pi_\theta$, and select the one with the lowest preference value:

$$x_{\bar{\pi}_\theta}^l = \mathrm{argmin}_{\{x_{\bar{\pi}_\theta}^k\}_{k=1}^K\sim\pi_\theta}r(x_{\bar{\pi}_\theta}^k,c). \tag{10}$$

However, $x_{\bar{\pi}_\theta}^l$ could still exhibit a relatively high preference $r(x_{\bar{\pi}_\theta}^l,c)$ due to the positive correlation between the generative probability $p_{\bar{\pi}_\theta}$ and preference distribution $p_r$ (see Sec. 3.2 or 3.3). To identify genuinely unpreferred motions, we apply a threshold $\tau$ to the set $\{x_{\bar{\pi}_\theta}^k\}_{k=1}^K$ and check if any preference score is below it. This leads to two training strategies based on the result.

**Case 1:** The group $\{x_{\bar{\pi}_\theta}^k\}_{k=1}^K$ contains a low-preference unpreferred motion $x_{\bar{\pi}_\theta}^l$. Then we select these unpreferred motions iteratively which ensure diversity due to randomness of online generations and address the diversity lacking issue in offline DPO.

**Case 2:** The group contains no low-preference unpreferred motion $x_{\bar{\pi}_\theta}^l$, meaning all sampled motions are of high preference and should not be treated as unpreferred. This suggests the model performs well under condition $c$, so training should focus on high-quality preferred motions from offline data to further enhance generation quality.

To operationalize this, we apply: (1) distribution separation and (2) training loss amendment.

**(1) Distribution separation:** With a threshold $\tau$, we separate the distribution $p_{\bar{\pi}_\theta}(x_{\bar{\pi}_\theta}^{1:K}|c)$ into two sub-distributions:

$$p_{\bar{\pi}_\theta}(x_{\bar{\pi}_\theta}^{1:K}|c) = \underbrace{p_{\bar{\pi}_\theta}(x_{\bar{\pi}_\theta}^{1:K}|c)p_\tau(r(x_{\bar{\pi}_\theta}^l,c)\geq\tau)}_{\text{relatively high-preference unpreferred motions } \bar{\pi}_\theta^{hu}} + \underbrace{p_{\bar{\pi}_\theta}(x_{\bar{\pi}_\theta}^{1:K}|c)p_\tau(r(x_{\bar{\pi}_\theta}^l,c)<\tau)}_{\text{valuable unpreferred motions } \bar{\pi}_\theta^{vu}}, \tag{11}$$

where $p_{\bar{\pi}_\theta}(x^{1:K}|c) = \prod_{k=1}^K p_{\bar{\pi}_\theta}(x^k|c)$, $p_{\bar{\pi}_\theta}(x^k|c)$ is the generative probability of policy model $\pi_\theta$ to generate $x^k$ conditioned on $c$, $p_\tau(r(x_{\bar{\pi}_\theta}^l,c)\geq\tau)$ is the probability of the event $r(x_{\bar{\pi}_\theta}^l) \geq \tau$, and $p_\tau(r(x_{\bar{\pi}_\theta}^l,c)\leq\tau)$ has similar meaning.

Eq. (11) indicates that the online generative distribution $\bar{\pi}_\theta(x_{\bar{\pi}_\theta}^{1:K}|c)$ can be separated according to whether the sampled motion $x_{\bar{\pi}_\theta}^{1:K}$ group contains valuable unpreferred motions. Accordingly, our

objective loss in Eq. (9) can also be divided into two ones: $\mathcal{L}_{\mathrm{DSoPo}}(\theta) = \mathcal{L}_{\mathrm{vu}}(\theta) + \mathcal{L}_{\mathrm{hu}}(\theta)$, where $\mathcal{L}_{\mathrm{vu}}(\theta)$ targets valuable unpreferred motions and $\mathcal{L}_{\mathrm{hu}}(\theta)$ targets high-preference unpreferred motions:

$$
\begin{aligned}
\mathcal{L}_{\mathrm{vu}} &= -\,\mathbb{E}_{(x^w,c)\sim\mathcal{D}} Z_{vu}(c)\mathbb{E}_{x^{1:K}_{\bar{\pi}_\theta}\sim\bar{\pi}^{vu*}_\theta(\cdot|c)}\log\sigma\big(\beta\mathcal{H}_\theta(x^w, x^l_{\bar{\pi}_\theta}, c)\big), \\
\mathcal{L}_{\mathrm{hu}} &= -\,\mathbb{E}_{(x^w,c)\sim\mathcal{D}} Z_{hu}(c)\mathbb{E}_{x^{1:K}_{\bar{\pi}_\theta}\sim\bar{\pi}^{hu*}_\theta(\cdot|c)}\log\sigma\big(\beta\mathcal{H}_\theta(x^w, x^l_{\bar{\pi}_\theta}, c)\big),
\end{aligned}
\tag{12}
$$

where $\mathcal{H}_\theta(x^w, x^l_{\bar{\pi}_\theta}, c)$ is defined in Eq. (2), $p_{\bar{\pi}^{vu*}_\theta}(\cdot) = \frac{p^{vu}_{\bar{\pi}_\theta}(\cdot)}{Z_{vu}(c)}$ and $p^{hu*}_{\bar{\pi}_\theta}(\cdot) = \frac{p^{hu}_{\bar{\pi}_\theta}(\cdot)}{Z_{hu}(c)}$ respectively denote the distributions of valuable unpreferred and high-preference unpreferred motions. Here $Z_{vu}(c) = \int p_{\bar{\pi}^{vu}_\theta}(x)\mathrm{dx}$ and $Z_{hu}(c) = \int p_{\bar{\pi}^{hu}_\theta}(x)\mathrm{dx}$ are the partition functions, and are unnecessary to be computed in our implementation (More discussion are provided in App. C.3).

**(2) Training loss amendment:** As discussed above, unpreferred motions in case 2 have relatively high-preference (score $\geq \tau$), and thus should not be classified into unpreferred motions for training. Accordingly, we rewrite the loss $\mathcal{L}_{\mathrm{hu}}(\theta)$ into $\mathcal{L}_{\mathrm{USoPo-hu}}(\theta)$ for filtering them:

$$
\mathcal{L}_{\mathrm{USoPo-hu}}(\theta) = -\mathbb{E}_{(x^w,c)\sim\mathcal{D}} Z_{hu}(c)\log\sigma\Big(\beta h_\theta(x^w, c)\Big), \quad \mathcal{L}_{\mathrm{USoPo}}(\theta) = \mathcal{L}_{\mathrm{USoPo-hu}}(\theta) + \mathcal{L}_{\mathrm{vu}}(\theta). \tag{13}
$$

See more discussion on $\mathcal{L}_{\mathrm{USoPo}}/\mathcal{L}_{\mathrm{DSoPo}}$ in App. C.4.

## 4.3 Offline Sampling for Preferred Motions

As discussed, online DPO suffers from a limited preference gap between preferred and unpreferred motions. While high-quality motions from offline datasets can help mitigate this issue, they may not always differ significantly from generated motions—especially when the model is well-aligned with the dataset. Thus, motions with larger preference gaps (Sec. 4.2) are crucial and should be prioritized.

To utilize the generated unpreferred motion set $\mathcal{D}_c$ conditioned on $c$ from Sec. 4.2, we calculate its proximity with the unpreferred motions in $\mathcal{D}_c$ using cosine similarity:

$$
S(x^w) = \min_{x^k_{\bar{\pi}_\theta}\sim\mathcal{D}_c} \cos(x^w, x^k_{\bar{\pi}_\theta}).
$$

Then we reweight the loss using $\beta_w(x_w) = \beta(C - S(x^w))$ with a constant $C \geq 1$:

$$
\begin{aligned}
\mathcal{L}_{\mathrm{SoPo}}(\theta) = &-\mathbb{E}_{(x^w,c)\sim\mathcal{D}, x^{1:K}_{\bar{\pi}_\theta}\sim\bar{\pi}^{vu*}_\theta(\cdot|c)} Z_{vu}(c)\Big[\log\sigma\Big(\beta_w(x^w)h_\theta(x^w, c) - \beta h_\theta(x^l, c)\Big)\Big] \\
&-\mathbb{E}_{(x^w,c)\sim\mathcal{D}} Z_{hu}(c)\log\sigma\Big(\beta_w(x^w)h_\theta(x^w, c)\Big).
\end{aligned}
\tag{14}
$$

As similar samples have similar preferences, this reweighting strategy guides the model to prioritize preferred motions with a significant preference gap from unpreferred ones. Accordingly, this reweighting strategy relieves and even addresses the small preference gap issue in online DPO.

## 4.4 SoPo for Diffusion-Based Text-to-Motion

Recently, diffusion text-to-motion models have achieved remarkable success [2, 6, 11, 12], enabling the generation of diverse and realistic motion sequences. Inspired by [27], we derive the objective function of SoPo for diffusion-based text-to-image generation (See proof in App. C.5):

$$
\mathcal{L}^{\mathrm{diff}}_{\mathrm{SoPo}} = \mathcal{L}^{\mathrm{diff}}_{\mathrm{SoPo-vu}} + \mathcal{L}^{\mathrm{diff}}_{\mathrm{SoPo-hu}}, \tag{15}
$$

$$
\begin{aligned}
\mathcal{L}^{\mathrm{diff}}_{\mathrm{SoPo-vu}} &= -\mathbb{E}_{t\sim\mathcal{U}(0,T),(x^w,c)\sim\mathcal{D}, x^{1:K}_{\bar{\pi}_\theta}\sim\bar{\pi}^{vu*}_\theta(\cdot|c)} Z_{vu}(c)\Big[\log\sigma\Big(-T\omega_t\big(\beta_w(x_w)(\mathcal{L}(\theta, \mathrm{ref}, x^w_t) - \beta\mathcal{L}(\theta, \mathrm{ref}, x^l_t))\big)\Big)\Big] \\
\mathcal{L}^{\mathrm{diff}}_{\mathrm{SoPo-hu}} &= -\mathbb{E}_{t\sim\mathcal{U}(0,T),(x^w,c)\sim\mathcal{D}} Z_{hu}(c)\Big[\log\sigma\Big(-T\omega_t\beta_w(x_w)\mathcal{L}(\theta, \mathrm{ref}, x^w_t)\Big)\Big],
\end{aligned}
\tag{16}
$$

where $\mathcal{L}(\theta, \mathrm{ref}, x_t) = \mathcal{L}(\theta, x_t) - \mathcal{L}(\mathrm{ref}, x_t)$, and $\mathcal{L}(\theta/\mathrm{ref}, x_t) = \|\epsilon_{\theta/\mathrm{ref}}(x_t, t) - \epsilon\|^2_2$ denotes the loss of the policy or reference model. Equivalently, we optimize the following form

$$
\mathcal{L}^{\mathrm{diff}}_{\mathrm{SoPo}}(\theta) = -\mathbb{E}_{t\sim\mathcal{U}(0,T),(x^w,c)\sim\mathcal{D}, x^{1:K}_{\bar{\pi}_\theta}\sim\bar{\pi}_\theta(\cdot|c)}
\begin{cases}
\log\sigma\Big(-T\omega_t\big(\beta_w(x_w)\mathcal{L}(\theta, \mathrm{ref}, x^w_t) - \beta\mathcal{L}(\theta, \mathrm{ref}, x^l_t))\big)\Big), & \text{if } r(x^l, c) < \tau, \\
\log\sigma\Big(-T\omega_t\beta_w(x_w)\mathcal{L}(\theta, \mathrm{ref}, x^w_t)\Big), & \text{otherwise.}
\end{cases}
\tag{17}
$$

where $x^l = \operatorname{argmin}_{\{x^k_{\bar{\pi}_\theta}\}^K_{k=1}\sim\pi_\theta} r(x^k_{\bar{\pi}_\theta}, c)$. Proof and more details are provided in App. B.

Table 1: **Quantitative results of preference alignment methods for text-to-motion generation on the HumanML3D test set.** Results are borrowed from those reported in [9]. The subscripts in each cell denotes the relative performance change. Superscript "†" marks the largest improvement across all models; gray background highlights the largest improvement for each model. "Time*" denotes estimated online/offline motion generation time, with "1X" as the time for MLD [1] to generate all HumanML3D motions and "$K$" (unspecified in [9], typically 2∼6) as the number of motion pairs.

| Methods | Time* | R-Precision ↑ | | | MM Dist ↓ | Diversity → | FID ↓ |
|---|---|---|---|---|---|---|---|
| | | Top 1 | Top 2 | Top 3 | | | |
| Real | - | $0.511^{\pm0.003}$ | $0.703^{\pm0.003}$ | $0.797^{\pm.002}$ | $2.974^{\pm0.008}$ | $9.503^{\pm0.065}$ | $0.002^{\pm0.000}$ |
| MLD [1] | +0 X | $0.453^{\pm0.003}$ | $0.679^{\pm0.003}$ | $0.755^{\pm0.003}$ | $3.292^{\pm0.010}$ | $9.793^{\pm0.072}$ | $0.459^{\pm0.011}$ |
| + MoDiPO-T [9] | +121$K$ X | $0.455^{\pm0.002}$ | $0.682^{\pm0.003}$ | $0.758^{\pm0.002}_{+0.40\%}$ | $3.267^{\pm.010}_{+0.76\%}$ | $9.747^{\pm0.073}_{+0.046}$ | $0.303^{\pm0.031}_{+33.9\%}$ |
| + MoDiPO-G [9] | +121$K$ X | $0.452^{\pm0.003}$ | $0.678^{\pm0.003}$ | $0.753^{\pm0.003}_{-0.26\%}$ | $3.294^{\pm0.010}_{-0.01\%}$ | $9.702^{\pm.075}_{+0.091}$ | $0.281^{\pm0.031}_{+38.8\%}$ |
| + MoDiPO-O [9] | - | $0.406^{\pm0.003}$ | $0.609^{\pm0.003}$ | $0.677^{\pm0.003}_{-10.3\%}$ | $3.701^{\pm0.013}_{-12.4\%}$ | $9.241^{\pm.079}_{-0.018}$ | $0.276^{\pm0.007}_{+39.9\%}{}^{\dagger}$ |
| + SoPo (Ours) | +20 X | $0.463^{\pm0.003}_{+2.21\%}$ | $0.682^{\pm0.003}_{+2.23\%}$ | $0.763^{\pm0.003}_{+1.06\%}$ | $3.185^{\pm0.012}_{+3.25\%}{}^{\dagger}$ | $9.525^{\pm0.065}_{+0.268}{}^{\dagger}$ | $0.374^{\pm0.007}_{+18.5\%}$ |
| MDM [13] | +0 X | $0.418^{\pm0.005}$ | $0.604^{\pm0.005}$ | $0.703^{\pm0.005}$ | $3.658^{\pm0.025}$ | $9.546^{\pm0.066}$ | $0.501^{\pm0.037}$ |
| + MoDiPO-T [9] | +121$K$ X | $0.421^{\pm0.006}$ | $0.635^{\pm0.005}$ | $0.706^{\pm0.004}_{+0.42\%}$ | $3.634^{\pm.026}_{+0.66\%}$ | $9.531^{\pm0.073}_{+0.015}$ | $0.451^{\pm0.031}_{+9.98\%}$ |
| + MoDiPO-G [9] | +121$K$ X | $0.420^{\pm0.006}$ | $0.632^{\pm0.005}$ | $0.704^{\pm0.001}_{+0.14\%}$ | $3.641^{\pm0.025}_{+0.46\%}$ | $9.495^{\pm0.071}_{+0.035}$ | $0.486^{\pm0.031}_{+2.99\%}$ |
| MDM (fast) [13] | +0 X | $0.455^{\pm0.006}$ | $0.645^{\pm0.007}$ | $0.749^{\pm0.004}$ | $3.304^{\pm0.023}$ | $9.948^{\pm0.084}$ | $0.534^{\pm0.052}$ |
| + SoPo (Ours) | +60 X | $0.479^{\pm0.006}_{+5.27\%}{}^{\dagger}$ | $0.674^{\pm.005}_{+4.50\%}{}^{\dagger}$ | $0.770^{\pm0.006}_{+2.80\%}{}^{\dagger}$ | $3.208^{\pm0.025}_{+2.91\%}$ | $9.906^{\pm.083}_{+0.042}$ | $0.480^{\pm0.046}_{+10.1\%}$ |

# 5 Experiment

**Datasets & Evaluation Metrics.** For text-to-motion generation, we evaluate SoPo on two widely used datasets, HumanML3D [3] and KIT-ML [36], focusing on two key aspects: alignment and generation quality. Alignment is assessed using R-Precision and MM Dist, while generation quality is measured by Diversity and FID. For text-to-image generation, we utilize Flux-Dev [37] as the foundational model and employ HPSv2 [38] as the reward model. Further results and details are in App. A.1.

**Implementation Details.** Due to limited preference-labeled motion data, we use existing datasets (e.g., HumanML3D, KIT-ML) as offline preferred motions. For online generation of unpreferred motions, we use TMR, a text-to-motion retrieval model [39], as the reward model. Hyperparameters $K$ and $\tau$ are tuned through preliminary experiments to balance performance and efficiency, with $\tau = 0.45$, $C = 2$, and $\beta = 1$ in Eq. (14). We set $K = 4$ for MDM [40] and $K = 2$ for MLD [1]. All models are trained in 100 minutes on a single NVIDIA GeForce RTX 4090D GPU. Since MLD* [2] is tailored for HumanML3D, we use MLD [1] for KIT-ML. More details are in App. A.4.

## 5.1 Main Results on Text-to-Motion Generation

**Settings.** We evaluate SoPo for preference alignment and motion generation, comparing it with state-of-the-art preference alignment [9] and text-to-motion methods [1, 7]. For fairness, we fine-tune MLD [1] and MDM [13] with SoPo, using a fast diffusion variant [13] with 50 sampling steps. We also fine-tune MLD* [2] as a stronger baseline. Since MLD* is not adapted to KIT-ML, we use MLD [1] and MoMask [44] for diffusion-based and autoregressive methods, respectively.

Table 2: **Quantitative comparison of state-of-the-art text-to-motion generation on the HumanML3D test set.** 'MLD*' refers to the enhanced reproduction of MLD [1] from [2]. For a fair comparison, we selected the "LMM-T" [41] with a similar size to ours.

| Methods | Year | R-Precision ↑ | | | | MM Dist ↓ | Diversity → | Multimodal ↑ | FID ↓ |
|---|---|---|---|---|---|---|---|---|---|
| | | Top 1 | Top 2 | Top 3 | Avg. | | | | |
| Real | - | $0.511^{\pm0.003}$ | $0.703^{\pm0.003}$ | $0.797^{\pm0.002}$ | 0.670 | $2.794^{\pm0.008}$ | $9.503^{\pm0.065}$ | - | $0.002^{\pm0.000}$ |
| TEMOS [40] | 2022 | $0.424^{\pm0.002}$ | $0.612^{\pm0.002}$ | $0.722^{\pm0.002}$ | 0.586 | $3.703^{\pm0.008}$ | $8.973^{\pm0.071}$ | $0.368^{\pm0.018}$ | $3.734^{\pm0.028}$ |
| T2M [3] | 2022 | $0.457^{\pm0.002}$ | $0.639^{\pm0.003}$ | $0.740^{\pm0.003}$ | 0.612 | $3.340^{\pm0.008}$ | $9.188^{\pm0.002}$ | $2.090^{\pm0.083}$ | $1.067^{\pm0.002}$ |
| MDM [13] | 2022 | $0.418^{\pm0.005}$ | $0.604^{\pm0.005}$ | $0.703^{\pm0.005}$ | 0.575 | $3.658^{\pm0.025}$ | $9.546^{\pm0.066}$ | $\mathbf{2.799}^{\pm0.072}$ | $0.501^{\pm0.037}$ |
| MLD [1] | 2023 | $0.481^{\pm0.003}$ | $0.673^{\pm0.003}$ | $0.772^{\pm0.002}$ | 0.642 | $3.196^{\pm0.016}$ | $9.724^{\pm0.082}$ | $2.413^{\pm0.079}$ | $0.473^{\pm0.013}$ |
| MotionGPT [42] | 2023 | $0.492^{\pm0.003}$ | $0.681^{\pm0.003}$ | $0.778^{\pm0.002}$ | 0.650 | $3.096^{\pm0.008}$ | $\mathbf{9.528}^{\pm0.071}$ | $2.008^{\pm0.084}$ | $0.232^{\pm0.008}$ |
| MotionDiffuse [14] | 2024 | $0.491^{\pm0.004}$ | $0.681^{\pm0.002}$ | $0.782^{\pm0.001}$ | 0.651 | $3.113^{\pm0.018}$ | $9.410^{\pm0.049}$ | $1.553^{\pm0.042}$ | $0.630^{\pm0.011}$ |
| OMG [43] | 2024 | - | - | $0.784^{\pm0.002}$ | - | - | $9.657^{\pm0.085}$ | - | $0.381^{\pm0.008}$ |
| Wang et. al. [6] | 2024 | $0.433^{\pm0.007}$ | $0.629^{\pm0.007}$ | $0.733^{\pm0.006}$ | 0.598 | $3.430^{\pm0.061}$ | $9.825^{\pm0.159}$ | 2.835 | $0.352^{\pm0.109}$ |
| MoDiPO-T [9] | 2024 | $0.455^{\pm0.003}$ | $0.682^{\pm0.003}$ | $0.758^{\pm0.002}$ | - | $3.267^{\pm0.010}$ | $9.747^{\pm0.073}$ | $2.663^{\pm0.111}$ | $0.303^{\pm0.031}$ |
| PriorMDM [12] | 2024 | $0.481^{\pm0.002}$ | - | - | - | $5.610^{\pm0.023}$ | $9.620^{\pm0.074}$ | - | $0.600^{\pm0.053}$ |
| LMM-T[1] [41] | 2024 | $0.496^{\pm0.002}$ | $0.685^{\pm0.002}$ | $0.785^{\pm0.002}$ | 0.655 | $3.087^{\pm0.012}$ | $9.176^{\pm0.074}$ | $1.465^{\pm0.048}$ | $0.415^{\pm0.002}$ |
| CrossDiff[3] [11] | 2024 | - | - | $0.730^{\pm0.003}$ | - | $3.358^{\pm0.011}$ | $9.577^{\pm0.082}$ | - | $0.281^{\pm0.016}$ |
| Motion Mamba [7] | 2024 | $0.502^{\pm0.003}$ | $0.693^{\pm0.002}$ | $0.792^{\pm0.002}$ | 0.662 | $3.060^{\pm0.009}$ | $9.871^{\pm0.084}$ | $2.294^{\pm0.058}$ | $0.281^{\pm0.011}$ |
| MLD* [1, 2] | 2024 | $0.504^{\pm0.002}$ | $0.698^{\pm0.003}$ | $0.796^{\pm0.002}$ | 0.666 | $3.052^{\pm0.009}$ | $9.634^{\pm0.064}$ | $2.267^{\pm0.082}$ | $0.450^{\pm0.011}$ |
| **MLD* [2]$_{+ SoPo}$** | 2025 | $\mathbf{0.528}_{+4.76\%}$ | $\mathbf{0.722}_{+3.44\%}$ | $\mathbf{0.827}_{+3.89\%}$ | $\mathbf{0.692}_{+3.90\%}$ | $\mathbf{2.939}_{+3.70\%}$ | $9.584_{+38.1\%}$ | $2.301^{\pm0.076}$ | $\mathbf{0.174}_{+61.3\%}$ |

**Comparison with Preference Alignment Methods.** Table 1 compares preference alignment methods. MoDiPO, a DPO-based method for motion generation, faces overfitting and biased sampling issues [18]. Conversely, our SoPo method uses diverse high-probability unpreferred and high-quality preferred motions, improving generation quality and reducing unpreferred motions. SoPo excels in most metrics except FID, with R-Precision gains of 5.27%, 4.50%, and 2.80% (vs. baseline 0.42%) and a 3.25% MM Dist. improvement (vs. MoDiPO's $-12.4\%$ to $+0.76\%$). SoPo boosts Diversity by 0.268 (vs. MoDiPO's $-0.018$ to 0.091). Despite MoDiPO's slight FID edge, SoPo's results are comparable, owing to conservative training on low-probability, high-preference samples. SoPo also eliminates pairwise labels and cuts preference data generation time to $\sim 1/10$ of that MoDiPO.

**Comparison with Motion Generation Methods.** We evaluate SoPo on HumanML3D [3], with results in Table 2. Using preference alignment, SoPo surpasses state-of-the-art methods in R-Precision, MM Dist, and FID, achieving the **best performance**. Although MotionGPT [42] has slightly higher Diversity (9.584 vs. 9.528), SoPo improves R-Precision by 6.46%, FID by 33.5%, and MM Dist by 5.34%. Compared to Motion Mamba and CrossDiff, SoPo increases Diversity by 0.287 and reduces MM Dist by 12.5%. It also enhances

Table 3: **Comparison of text-to-motion generation performance on the KIT-ML dataset.**

| Methods | R Precision ↑ | | | FID ↓ | MM Dist ↓ | Diversity → |
|---|---|---|---|---|---|---|
| | Top 1 | Top 2 | Top 3 | | | |
| Real | 0.424 | 0.649 | 0.779 | 0.031 | 2.788 | 11.08 |
| TEMOS [40] | 0.370 | 0.569 | 0.693 | 2.770 | 3.401 | 10.91 |
| T2M [3] | 0.361 | 0.559 | 0.681 | 3.022 | 2.052 | 10.72 |
| MLD [1] | 0.390 | 0.609 | 0.734 | 0.404 | 3.204 | 10.80 |
| T2M-GPT [45] | 0.416 | 0.627 | 0.745 | 0.514 | 3.007 | 10.86 |
| MotionGPT [42] | 0.366 | 0.558 | 0.680 | 0.510 | 3.527 | 10.35 |
| MotionDiffuse[14] | 0.417 | 0.621 | 0.739 | 1.954 | 2.958 | 11.10 |
| Mo.Mamba [7] | 0.419 | 0.645 | 0.765 | 0.307 | 3.021 | **11.02** |
| MoMask [44] | 0.433 | 0.656 | 0.781 | 0.204 | **2.779** | 10.71 |
| **MLD [1]**$_{+\text{SoPo}}$ | 0.412 | 0.646 | 0.759 | 0.384 | 3.107 | 10.93 |
| **MoMask [44]**$_{+\text{SoPo}}$ | **0.446** | **0.673** | **0.797** | **0.176** | 2.783 | 10.96 |

MLD*'s FID by 61.3%. On KIT-ML (Table 3), SoPo with MoMask [44] achieves the **best results** across all metrics: Top-$k$ R-Precision (0.446, 0.673, 0.797), MM Dist (2.783), and FID (0.176). MLD w/ SoPo outperforms its original version, confirming its effectiveness across model architectures.

**Quantitative Evaluation of Spatial-Perception Motion Generation via SoPo.** We quantitatively analyze the efficacy of our SoPo in resolving issues related to *Spatial-Perception Motion Generation* shown in Fig.1. Experimental setting detailed in App. A.3. As exhibited in Fig. 4(a), these results confirm SoPo's effectiveness in enhancing spatial-perception capabilities.

## 5.2 Ablation Studies

**Impact of Sample Size $K$.** Due to computational and memory constraints, we recommend keeping $K < 8$. As shown in Table 4, increasing $K$ significantly improves generation quality. A larger sample pool allows the reward model to better evaluate and filter unpreferred motions, leading to more accurate guidance and higher-quality results.

**Impact of Objective Functions.** We fine-tune MDM [13] using four objectives: DSoPo (Eq. (12)), USoPo (Eq. (13)), SoPo without value-unpreferred (VU), and full SoPo (Eq. (14)). As shown in Table 4, DSoPo alleviates limitations of offline/online DPO (Sec. 4.1) and improves FID by 7.30%. Removing VU further boosts FID to 8.98% by emphasizing preferred motions that differ from unpreferred ones. USoPo, using a threshold $\tau$ to filter unpreferred motions, enhances R-Precision (+3.96%),

Table 4: **Ablation study on alignment methods, thresholds $\tau$, and sampled number $K$.**

| Methods | R Precision ↑ | | | FID ↓ | MM Dist ↓ | Diversity → |
|---|---|---|---|---|---|---|
| | Top 1 | Top 2 | Top 3 | | | |
| MDM (fast) [13] | .455 | .645 | .749 | 3.304 | 9.948 | .534 |
| +DSoPo | .460$_{+1.08\%}$ | .655$_{+1.55\%}$ | .756$_{+0.93\%}$ | 3.297$_{+0.02\%}$ | 9.925$_{+0.033}$ | .495$_{+7.30\%}$ |
| +SoPo w/o VU | .460$_{+1.08\%}$ | .656$_{+1.71\%}$ | .756$_{+0.93\%}$ | 3.295$_{+0.02\%}$ | 9.915$_{+0.033}$ | .486$_{+8.98\%}$ |
| +USoPo | .473$_{+3.96\%}$ | .668$_{+3.57\%}$ | .767$_{+2.40\%}$ | 3.226$_{+2.36\%}$ | **9.901**$_{+0.047}$ | .556$_{-4.12\%}$ |
| +SoPo | **.479**$_{+5.27\%}$ | **.674**$_{+4.50\%}$ | **.770**$_{+2.80\%}$ | **3.208**$_{+2.91\%}$ | 9.906$_{+0.042}$ | **.480**$_{+10.1\%}$ |
| +SoPo ($\tau = 0.40$) | .475$_{+4.40\%}$ | .661$_{+2.48\%}$ | .768$_{+2.53\%}$ | 3.272$_{+0.97\%}$ | 10.04$_{-0.088}$ | .600$_{-12.4\%}$ |
| +SoPo ($\tau = 0.45$) | .479$_{+5.27\%}$ | .674$_{+4.50\%}$ | .770$_{+2.80\%}$ | 3.208$_{+2.91\%}$ | 9.906$_{+0.042}$ | .480$_{+10.1\%}$ |
| +SoPo ($\tau = 0.50$) | .468$_{+2.86\%}$ | .663$_{+2.79\%}$ | .764$_{+2.01\%}$ | 3.256$_{+1.45\%}$ | 9.900$_{+0.048}$ | .491$_{+8.05\%}$ |
| +SoPo ($\tau = 0.55$) | .466$_{+2.41\%}$ | .660$_{+1.86\%}$ | .763$_{+1.87\%}$ | 3.263$_{+1.24\%}$ | 9.896$_{+0.041}$ | .430$_{+19.5\%}$ |
| +SoPo ($\tau = 0.60$) | .461$_{+1.31\%}$ | .656$_{+1.71\%}$ | .758$_{+1.20\%}$ | 3.288$_{+0.48\%}$ | **9.803**$_{+0.145}$ | **.399**$_{+25.3\%}$ |
| +SoPo ($K = 2$) | **.480**$_{+5.50\%}$ | .671$_{+4.03\%}$ | **.771**$_{+2.94\%}$ | 3.212$_{+2.78\%}$ | 9.907$_{+0.041}$ | .502$_{+5.99\%}$ |
| +SoPo ($K = 4$) | .479$_{+5.27\%}$ | **.674**$_{+4.50\%}$ | .770$_{+2.80\%}$ | **3.208**$_{+2.91\%}$ | 9.906$_{+0.042}$ | **.480**$_{+10.1\%}$ |

MM Dist (+2.36%), and Diversity (+0.047), though FID slightly drops (–4.12%). Combining all advantages, SoPo achieves the best results: +5.27% R-Precision and +10.1% FID.

**Impact of Cut-Off Thresholds $\tau$.** Table 4 reports results with $\tau$ ranging from 0.40 to 0.60. A lower $\tau$ leads to stricter filtering, yielding more reliable unpreferred motions. As $\tau$ decreases, R-Precision and MM Dist improve, indicating better alignment. In contrast, higher $\tau$ values improve FID and Diversity, suggesting enhanced generative quality due to exposure to more diverse samples. More

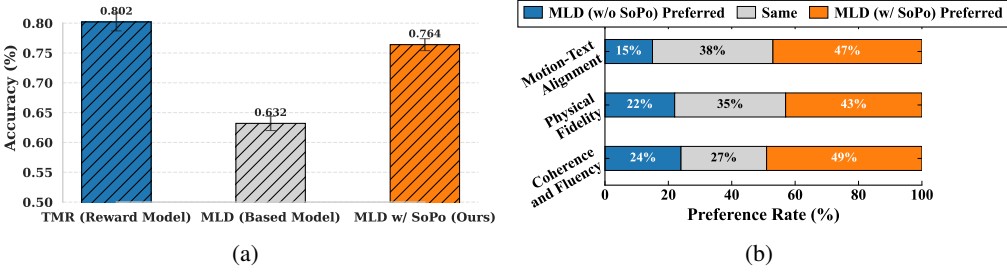

Figure 4: Quantitative results on (a) spatial-preception motion generation, and (b) user study.

experimental results, including ablation of training strategy and DPO hyper-parameter are shown in App. A.5.

**Impact of Training Strategy.** To compare different training strategy, we conducted new experiments comparing online DPO (ON. DPO), offline DPO (Off. DPO), their naive combination (Com. DPO), and combination with our proposed strategies (SoPo), as shown in Table 5. These results highlight that SoPo's hybrid semi-online design provides more effective and data-efficient alignment, avoiding the limitations of both pure online and offline DPO.

Table 5: **Ablation study on training strategy.**

| Methods | R Precision ↑ | | | FID ↓ | MM Dist ↓ | Diversity → |
|---------|---------------|---------|---------|-------|-----------|-------------|
| | Top 1 | Top 2 | Top 3 | | | |
| MLD* [2] | 0.504 | 0.698 | 0.796 | 3.052 | 9.634 | 0.450 |
| Off.DPO | 0.498 | 0.692 | 0.791 | 3.080 | 9.620 | 0.470 |
| On.DPO | 0.514 | 0.709 | 0.808 | 3.010 | 9.610 | 0.410 |
| Com.DPO | 0.517 | 0.712 | 0.811 | 2.985 | 9.605 | 0.340 |
| SoPo | **0.528** | **0.722** | **0.827** | **2.939** | **9.584** | **0.174** |

### 5.3 Discussion on Reward Hacking.

**User Study & Visualization.** To assess whether our fine-tuned model exhibits reward hacking, we conducted a user study and visualized the corresponding motions, as shown in Fig. 4(b). Additionlly, we visualize results of our SoPo and existing methods, provided in App. A.6. These results confirm that our SoPo can avoid reward hacking by KL-Divergence in Eq.(1).

## 6 Conclusion

In this study, we introduce a semi-online preference optimization method: a DPO-based fine-tune method for the text-to-motion model to directly align preference on "Semi-online data" consisting of high-quality preferred and diverse unpreferred motions. Our SoPo leverages the advantages both of online DPO and offline DPO, to overcome their own limitations. Furthermore, to ensure the validity of SoPo, we present a simple yet effective online generation method along with an offline reweighing strategy. Extensive experimental results show the effectiveness of our SoPo.

**Limitation discussion.** SoPo relies on a reward model to motion quality evaluation and identify usable unpreferred samples. However, research on reward models in the motion domain remains scarce, and current models, trained on specific datasets, exhibit limited generalization. Consequently, SoPo inherits these limitations, facing challenges in seamlessly fine-tuning diffusion models with reward models across diverse, open-domain scenarios.

## Acknowledgements

This work was supported by the Jiangsu Science Foundation (BK20230833, BG2024036, BK20243012), the National Science Foundation of China (62302093, 52441503, 62125602, U24A20324, 92464301), the Fundamental Research Funds for the Central Universities (2242025K30024), the Open Research Fund of the State Key Laboratory of Multimodal Artificial Intelligence Systems (E5SP060116), the Big Data Computing Center of Southeast University, and the Singapore Ministry of Education (MOE) Academic Research Fund (AcRF) Tier 1 grant (Proposal ID: 23-SIS-SMU-070). Any opinions, findings and conclusions or recommendations expressed in this material are those of the author(s) and do not reflect the views of the Ministry of Education, Singapore..

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

# SoPo: Text-to-Motion Generation Using Semi-Online Preference Optimization

## Supplementary Material

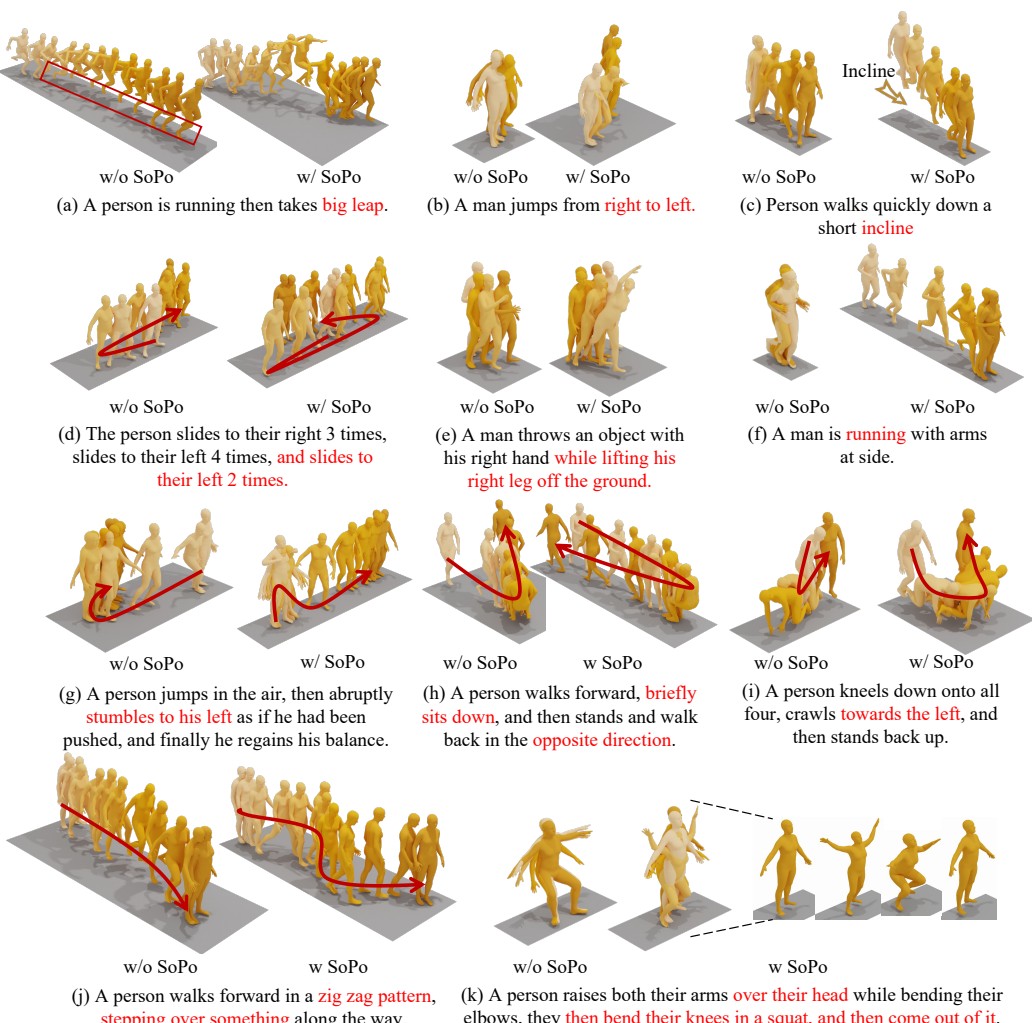

Figure S1: Visual results on HumanML3D dataset. We integrate our SoPo into MDM [13] and MLD [1], respectively. Our SoPo improves the alignment between text and motion preferences. Here, the red text denotes descriptions inconsistent with the generated motion.

This supplementary document contains the technical proofs of results and some additional experimental results. It is structured as follows. Sec. A presents the additional experiment information, including additional experimental details (Sec. A.2 and A.4) and results (Sec. A.6). Sec. B provides the implementation and theoretical analysis of our SoPo. Sec. C gives the proofs of the main results, including Theorem 1, Theorem 2, the objective function of DSoPo, the objective function of USoPo, and theorem of SoPo for text-to-motion generation. Sec. D provides more related works.

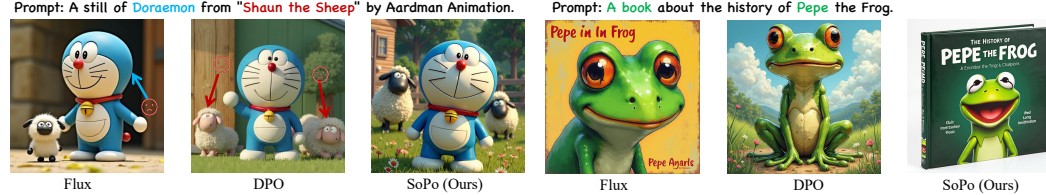

Figure S2: Visualization of text-to-image generation on the HPD dataset.

# A  Experiment

## A.1  Details of Experiments on Text-to-Motion Generation

For text-to-image generation, we utilize Flux-Dev [37] as the foundational generation model and employ HPSv2 [38] as the reward model. To construct the offline training pairs, we first sample data from the HPDv2 dataset. However, due to the inferior image quality in HPDv2, compared to that produced by Flux-Dev, we generated 20,000 high-fidelity image pairs using Flux-Dev to create the final offline dataset. We evaluate text-to-image the performance on Pick Score [46], CLIP [47], Image Reward [48] and Unified Reward [49]. The text-to-image model was trained for 330 GPU hours across 8 NVIDIA GPUs using LoRA, configured with a rank of $r = 128$ and a scaling factor $\alpha = 256$.

**Results on Text-to-Image Generation.** As shown in Table S1, the proposed SoPo consistently achieves superior performance across all evaluated text-to-image metrics, including HPS$(0.321)$ and IR$(1.194)$, outperforming the base FLUX model and standard DPO variants. Visualization is shown in Fig.S2.

Table S1: **Comparison of text-to-image generation on HPD dataset.**

| Method | HPS [38] | CLIP [47] | PS [46] | IR [48] | UR[49] | GPU Hours |
|---|---|---|---|---|---|---|
| FLUX | 0.313 | 0.388 | 0.227 | 1.088 | 3.370 | - |
| + On.DPO | 0.317 | 0.390 | 0.228 | 1.154 | 3.421 | 316 |
| + Off.DPO | 0.318 | 0.392 | 0.230 | 1.177 | 3.402 | 41 |
| + SoPo | **0.321** | **0.396** | **0.232** | **1.194** | **3.439** | **32** |

## A.2  Details of Experiments on Synthetic Data

To simulate our preference optimization framework, we design a 2D synthetic setup with predefined generation and reward distributions. The generator distribution $\pi_\theta$ is modeled as a Gaussian with mean $[-2, 1]$ and covariance matrix $\mathrm{diag}(2.0, 2.0)$. The reward model is defined as a mixture of two Gaussians with means $[-3, 2]$ and $[2, -2]$, covariances $\begin{bmatrix} 1 & \pm0.5 \\ \pm0.5 & 1 \end{bmatrix}$, and equal weights of 0.5.

For the **offline** dataset, preferred samples are randomly drawn from the reward distribution, while unpreferred samples are sampled from a manually specified distribution dissimilar to the reward model. These are used to fine-tune the generator via offline preference optimization. For the **online** setting, we draw samples from the reference model and assign preference labels using the reward model to distinguish preferred and unpreferred motions. This process is repeated iteratively to optimize the model online.In **SoPo**, we combine offline preferred samples with online-generated unpreferred ones to perform semi-online preference optimization, thereby leveraging the strengths of both offline and online data.

## A.3  Details of Spatial-Perception Motion Generation via SoPo

The core insight to solve this issue is that reward models (discriminators) are better at judging spatial semantics than generative models (generator), and SoPo leverages reward feedback to improve alignment.

We divide this issue into three sub-issues:

1. *Can the reward model distinguish left/right correctly?*
2. *Can the diffusion model generate motions consistent with left/right prompts?*

3. *Can SoPo improve generation via preference optimization?*

**Reward Model Discrimination Ability of Spatial Misalignment.** From the HumanML3D test set (2,192 prompts), 783 prompts (35.72%) contain spatial information (e.g., "left" or "right"), highlighting the prevalence of this issue. For a text-motion pair $(x, t)$, we computed the reward score $r(x, t)$. We then created a misaligned text $t'$ by swapping "left" with "right" and computed $r(x, t')$. The reward model is considered successful if $r(x, t) > r(x, t')$.

**Diffusion Model Generative Ability of Spatial Alignment Generation.** We randomly selected 100 spatial prompts from the 783 and generated 5 motions per prompt (500 total). Human annotators judged whether motions matched the spatial constraints.

These results in Fig. 4 (a) demonstrate that:

1. *The reward model is capable of detecting spatial misalignments;*

2. *The original diffusion model struggles with spatial understanding;*

3. *SoPo effectively enhances spatial alignment in generated motions.*

Thus, SoPo offers a practical solution to address spatial misalignment by integrating spatial semantic information from the text-motion-aligned reward model.

## A.4 Additional Experimental Datails

**Datasets & Evaluation.** HumanML3D is derived from the AMASS [50] and HumanAct12 [51] datasets and contains 14,616 motions, each described by three textual annotations. All motion is split into train, test, and evaluate sets, composed of 23384, 1460, and 4380 motions, respectively. For both HumanML3D and KIT-ML datasets, we follow the official split and report the evaluated performance on the test set.

We evaluate our experimental results on two main aspects: alignment quality and generation quality. Following prior research [2, 7, 11], we use motion retrieval precision (R-Precision) and multi-modal distance (MM Dist) to evaluate alignment quality, while diversity and Fréchet Inception Distance (FID) are employed to assess generation quality. (1) R-Precision evaluates the similarity between generated motion and their corresponding text descriptions. Higher values indicate better alignment quality. (2) MM Dist represents the average distance between the generated motion features and their corresponding text embedding. (3) Diversity calculates the variation in generated samples. A diversity close to real motions ensures that the model produces rich patterns rather than repetitive motions. (4) FID measures the distribution proximity between the generated and real samples in latent space. Lower FID scores indicate higher generation quality.

**Implementation Details.** For the preference alignment of MDM [13], we largely adopt the original implementation's settings. The model is trained using the AdamW optimizer [52] with a cosine decay learning rate scheduler and linear warm-up over the initial steps. We use a batch size of 64, with a guidance parameter of 2.5 during testing. Diffusion employs a cosine noise schedule with 50 steps, and an evaluation batch size of 32 ensures consistent metric computation. For fine-tuning MLD [1], we similarly follow its original parameter settings.

Table S2: **Hyperparameters analysis of our SoPo.**

| Methods | R Precision ↑ | | | FID ↓ | MM Dist ↓ |
|---|---|---|---|---|---|
| | Top 1 | Top 2 | Top 3 | | |
| SoPo(C=1,$\beta$=0.25) | 0.523 | 0.717 | 0.823 | 2.941 | 0.176 |
| SoPo(C=1,$\beta$=0.5) | 0.524 | 0.718 | 0.824 | 2.940 | 0.175 |
| SoPo(C=1,$\beta$=1) | 0.525 | 0.719 | 0.825 | 2.939 | 0.174 |
| SoPo(C=2,$\beta$=0.25) | 0.527 | 0.721 | 0.826 | 2.938 | 0.173 |
| SoPo(C=2,$\beta$=0.5) | 0.528 | 0.722 | 0.827 | 2.937 | 0.172 |
| SoPo(C=2,$\beta$=1) | 0.528 | 0.722 | 0.827 | 2.939 | 0.174 |
| SoPo(C=3,$\beta$=0.5) | **0.532** | **0.726** | **0.831** | 2.935 | 0.170 |
| SoPo(C=3,$\beta$=1) | 0.530 | 0.724 | 0.829 | **2.934** | **0.169** |
| SoPo(C=3,$\beta$=2) | 0.529 | 0.723 | 0.828 | 2.936 | 0.171 |

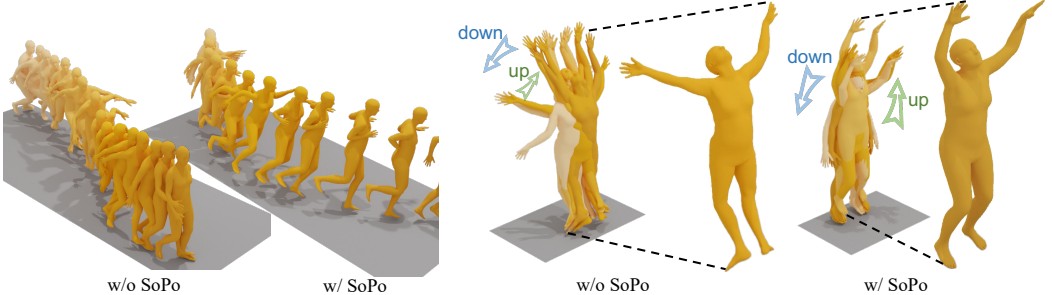

(a) A person runs to their right and then curves to the left and continues to run then stops.

(b) A man jumps and brings both arms above his head as … and then moves them back into the original position.

Figure S3: Visual results on HumanML3D dataset.

## A.5 Additional Ablation Results

**Impact of Hyperparameters Setting.** The hyperparameters of our SoPo can be divided into two types: (1). From SoPo: filtering threshold $\tau$, candidate number $K$, weight $C$; (2). From DPO: temperature $\beta$. For SoPo-specific hyperparameters, Table. S2 shows they have minor influence. Below, we report results on MLD* to analyze the sensitivity to $\beta$ and $C$:

## A.6 Additional Experimental Results

We visualize the generated motion for our SoPo. As shown in Fig. S3, our proposed approach helps text-to-motion models avoid frequent mistakes, such as incorrect movement direction and specific semantics. Additionally, we also present additional results generated by text-to-motion models with SoPo, as illustrated in Fig. S1. Our proposed SoPo significantly enhances the ability of text-to-motion models to comprehend text semantics. For instance, in Fig. S1 (j), a model integrated with SoPo can successfully interpret the semantics of "zig-zag pattern", whereas a model without SoPo struggles to do so.

## B    Details of SoPo for Text-to-Motion Generation

In this section, we first examine the objective function of SoPo and argue that it presents significant challenges for optimization. Fortunately, we then discover and derive an equivalent form that is easier to optimize (Sec. B.1). Finally, we design an algorithm to optimize it and finish discussing their correspondence (Sec. B.2).

### B.1    Equivalent form of SoPo

In Eq. (15) and (16), the objective function of SoPo is defined as:

$$\mathcal{L}_{\text{SoPo}}^{\text{diff}} = \mathcal{L}_{\text{SoPo-vu}}^{\text{diff}} + \mathcal{L}_{\text{SoPo-hu}}^{\text{diff}}, \tag{S1}$$

$$
\begin{aligned}
\mathcal{L}_{\text{SoPo-vu}}^{\text{diff}} &= -\mathbb{E}_{t\sim\mathcal{U}(0,T),(x^w,c)\sim\mathcal{D},x_{\bar{\pi}_\theta}^{1:K}\sim\bar{\pi}_\theta^{vu*}(\cdot|c)} Z_{vu}(c) \\
&\qquad \left[ \log\sigma\Big( -T\omega_t\big(\beta_w(x_w)\big(\mathcal{L}(\theta,\text{ref},x_t^w) - \beta\mathcal{L}(\theta,\text{ref},x_t^l)\big)\big)\Big) \right], \\
\mathcal{L}_{\text{SoPo-hu}}^{\text{diff}} &= -\mathbb{E}_{t\sim\mathcal{U}(0,T),(x^w,c)\sim\mathcal{D}} Z_{hu}(c) \\
&\qquad \left[ \log\sigma\Big( -T\omega_t\beta_w(x_w)\mathcal{L}(\theta,\text{ref},x_t^w)\Big) \right],
\end{aligned}
\tag{S2}
$$

However, these objectives can not be directly optimized, since the distribution $\bar{\pi}_\theta^{vu*}$ and $\bar{\pi}_\theta^{hu*}$ are not defined explicitly. To this end, we begin by inducing its equivalent form:

$$\mathcal{L}_{\text{SoPo}}^{\text{diff}}(\theta) = -\mathbb{E}_{t\sim\mathcal{U}(0,T),(x^w,c)\sim\mathcal{D},x_{\bar{\pi}_\theta}^{1:K}\sim\bar{\pi}_\theta(\cdot|c)}$$

$$\begin{cases} \log\sigma\Big(-T\omega_t\big(\beta_w(x_w)\big(\mathcal{L}(\theta,\text{ref},x_t^w)-\beta\mathcal{L}(\theta,\text{ref},x_t^l)\big)\big)\Big), & \text{if } r(x^l,c)<\tau, \\ \log\sigma\Big(-T\omega_t\beta_w(x_w)\mathcal{L}(\theta,\text{ref},x_t^w)\Big), & \text{otherwise.} \end{cases} \quad \text{(S3)}$$

where $x^l = \operatorname{argmin}_{\{x_{\bar{\pi}_\theta}^k\}_{k=1}^K\sim\pi_\theta} r(x_{\bar{\pi}_\theta}^k,c)$.

*Proof.* Recall our definition of $\mathcal{L}_{\text{SoPo}}^{\text{diff}}(\theta)$ in Eq. (15) and (16). Through algebraic maneuvers, we have:

$$\mathcal{L}_{\text{SoPo}}^{\text{diff}} = \mathcal{L}_{\text{SoPo}-\text{vu}}^{\text{diff}} + \mathcal{L}_{\text{SoPo}-\text{hu}}^{\text{diff}}$$

$$= -\mathbb{E}_{t\sim\mathcal{U}(0,T),(x^w,c)\sim\mathcal{D},x_{\bar{\pi}_\theta}^{1:K}\sim\bar{\pi}_\theta^{vu*}(\cdot|c)} Z_{vu}(c)$$
$$\Big[\log\sigma\Big(-T\omega_t\big(\beta_w(x_w)\big(\mathcal{L}(\theta,\text{ref},x_t^w)-\beta\mathcal{L}(\theta,\text{ref},x_t^l)\big)\big)\Big)\Big]$$
$$- \mathbb{E}_{t\sim\mathcal{U}(0,T),(x^w,c)\sim\mathcal{D}} Z_{hu}(c)\Big[\log\sigma\Big(-T\omega_t\beta_w(x_w)\mathcal{L}(\theta,\text{ref},x_t^w)\Big)\Big]$$

$$= -\mathbb{E}_{t\sim\mathcal{U}(0,T),(x^w,c)\sim\mathcal{D}}\mathbb{E}_{x_{\bar{\pi}_\theta}^{1:K}\sim\bar{\pi}_\theta^{vu*}(\cdot|c)} Z_{vu}(c)$$
$$\Big[\log\sigma\Big(-T\omega_t\big(\beta_w(x_w)\big(\mathcal{L}(\theta,\text{ref},x_t^w)-\beta\mathcal{L}(\theta,\text{ref},x_t^l)\big)\big)\Big)\Big]$$
$$- \mathbb{E}_{t\sim\mathcal{U}(0,T),(x^w,c)\sim\mathcal{D}}\mathbb{E}_{x_{\bar{\pi}_\theta}^{1:K}\sim\bar{\pi}_\theta^{hu*}(\cdot|c)} Z_{hu}(c)\Big[\log\sigma\Big(-T\omega_t\beta_w(x_w)\mathcal{L}(\theta,\text{ref},x_t^w)\Big)\Big]$$

$$= -\mathbb{E}_{t\sim\mathcal{U}(0,T),(x^w,c)\sim\mathcal{D}}\mathbb{E}_{x^{1:K}} p_{\bar{\pi}_\theta}^{vu}(x_{\bar{\pi}_\theta}^{1:K}|c)$$
$$\Big[\log\sigma\Big(-T\omega_t\big(\beta_w(x_w)\big(\mathcal{L}(\theta,\text{ref},x_t^w)-\beta\mathcal{L}(\theta,\text{ref},x_t^l)\big)\big)\Big)\Big]$$
$$- \mathbb{E}_{t\sim\mathcal{U}(0,T),(x^w,c)\sim\mathcal{D}}\mathbb{E}_{x^{1:K}} p_{\bar{\pi}_\theta}^{hu}(x_{\bar{\pi}_\theta}^{1:K}|c)\Big[\log\sigma\Big(-T\omega_t\beta_w(x_w)\mathcal{L}(\theta,\text{ref},x_t^w)\Big)\Big]$$

$$\overset{\text{①}}{=} -\mathbb{E}_{t\sim\mathcal{U}(0,T),(x^w,c)\sim\mathcal{D}}\mathbb{E}_{x_{\bar{\pi}_\theta}^{1:K}\sim\bar{\pi}_\theta(\cdot|c)} p_\tau(r(x_{\bar{\pi}_\theta}^l,c)<\tau)$$
$$\Big[\log\sigma\Big(-T\omega_t\big(\beta_w(x_w)\big(\mathcal{L}(\theta,\text{ref},x_t^w)-\beta\mathcal{L}(\theta,\text{ref},x_t^l)\big)\big)\Big)\Big]$$
$$- \mathbb{E}_{t\sim\mathcal{U}(0,T),(x^w,c)\sim\mathcal{D}}\mathbb{E}_{x_{\bar{\pi}_\theta}^{1:K}\sim\bar{\pi}_\theta(\cdot|c)} p_\tau(r(x_{\bar{\pi}_\theta}^l,c)\geq\tau)\Big[\log\sigma\Big(-T\omega_t\beta_w(x_w)\mathcal{L}(\theta,\text{ref},x_t^w)\Big)\Big]$$

$$= -\mathbb{E}_{t\sim\mathcal{U}(0,T),(x^w,c)\sim\mathcal{D},x_{\bar{\pi}_\theta}^{1:K}\sim\bar{\pi}_\theta(\cdot|c)}$$
$$\begin{cases} \log\sigma\Big(-T\omega_t\big(\beta_w(x_w)\big(\mathcal{L}(\theta,\text{ref},x_t^w)-\beta\mathcal{L}(\theta,\text{ref},x_t^l)\big)\big)\Big), & \text{if } r(x^l,c)<\tau, \\ \log\sigma\Big(-T\omega_t\beta_w(x_w)\mathcal{L}(\theta,\text{ref},x_t^w)\Big), & \text{otherwise.} \end{cases}$$

where $\overset{\text{①}}{=}$ holds since $p_{\bar{\pi}_\theta^{vu*}}(\cdot) = \frac{p_{\bar{\pi}_\theta}^{vu}(\cdot)}{Z_{vu}(c)}$ and $p_{\bar{\pi}_\theta}^{vu}(x_{\bar{\pi}_\theta}^{1:K}|c) = p_{\bar{\pi}_\theta}(x_{\bar{\pi}_\theta}^{1:K}|c)\cdot p_\tau(r(x_{\bar{\pi}_\theta}^l,c)\geq\tau)$. The proof is completed. □

## B.2 The process of SoPo for text-to-motion generation

Based on the equivalent form of SoPo in Eq. (S3), we can design an algorithm to directly optimize it, as shown in **Algorithm 1**.

The SoPo optimizes a policy model $\pi_\theta$ for text-to-motion generation through an iterative process guided by a reward model. In each iteration, given a preferred motion $x^w$ and a conditional code $c$, a random diffusion step $t$ is selected, and $K$ candidate motions are generated by $\pi_\theta$. The motion with the lowest preference score is then treated as the unpreferred motion. To determine the weight of the preferred motion $x^w$, the similarities between all generated motions are computed, and the lowest cosine similarity value is used to calculate its weight. Finally, the loss is calculated in two ways, determined based on the preference scores of the unpreferred motion. If the preference score of the selected unpreferred motion falls below a threshold $\tau$, it is identified as a valuable unpreferred motion and used for training. Otherwise, it indicates that the motions generated by the policy model $\pi_\theta$ are satisfactory. In such cases, the policy model is trained exclusively on high-quality preferred motions, rather than on both preferred motions and relatively high-preference unpreferred motions.

**Algorithm 1** SoPo for text-to-motion generation

**Input:** Preference dataset $\mathcal{D}$; diffusion steps $T$; iterations $I$; samples $K$; ref model $\pi_{\text{ref}}$; policy $\pi_\theta$; threshold $\tau$
**Output:** Aligned model $\pi_\theta$
1: **for** $i = 1$ to $I$ **do**
2:     **for** each $(x^w, c) \in \mathcal{D}$ **do**
3:         Sample $t \sim \mathcal{U}(0, T)$
4:         Sample $x_{\bar{\pi}_\theta}^{1:K} \sim \bar{\pi}_\theta(\cdot|c)$
5:         Compute $S(x^w) = \min_k \cos(x^w, x_{\bar{\pi}_\theta}^k)$
6:         $x^l = \arg\min_k r(x_{\pi_\theta}^k, c)$
7:         **if** $r(x^l, c) < \tau$ **then**
8:             $\mathcal{L} = \log \sigma(-T\omega_t \beta_w(x^w)(\mathcal{L}(\theta, \text{ref}, x_t^w) - \beta\mathcal{L}(\theta, \text{ref}, x_t^l)))$
9:         **else**
10:            $\mathcal{L} = \log \sigma(-T\omega_t \beta_w(x^w)\mathcal{L}(\theta, \text{ref}, x_t^w))$
11:         **end if**
12:         Accumulate loss: $\mathcal{L}_{\text{SoPo}}^{\text{diff}} + = \mathcal{L}$
13:     **end for**
14:     Update $\pi_\theta$ using $\nabla_\theta \mathcal{L}_{\text{SoPo}}^{\text{diff}}$
15: **end for**
16: **return** $\pi_\theta$

To further understand the objective function, we analyze the correspondence between the objective function in Eq. (S3) and Algorithm 1:

$$\mathcal{L}_{\text{SoPo}}^{\text{diff}}(\theta) = - \mathbb{E}_{\underbrace{(x^w, c) \sim \mathcal{D}}_{\text{Line 2}}, \underbrace{t \sim \mathcal{U}(0,T)}_{\text{Line 3}}, \underbrace{x_{\bar{\pi}_\theta}^{1:K} \sim \bar{\pi}_\theta(\cdot|c)}_{\text{Line 4}}}$$

$$\begin{cases} \underbrace{\log \sigma\Big( - T\omega_t \big(\beta_w(x_w)(\mathcal{L}(\theta, \text{ref}, x_t^w) - \beta\mathcal{L}(\theta, \text{ref}, x_t^l))\big) \Big)}_{\text{Line 8}}, & \underbrace{\text{If } r(x^l, c) < \tau}_{\text{Line 7}}, \\ \underbrace{\log \sigma\Big( - T\omega_t \beta_w(x_w)\mathcal{L}(\theta, \text{ref}, x_t^w)\Big)}_{\text{Line 10}}, & \underbrace{\text{Otherwise}}_{\text{Line 9}} . \end{cases} \quad \text{(S4)}$$

## C   Theories

### C.1   Proof of Theorem 1

*Proof.* The offline DPO based on Plackett-Luce model [34] can be denoted as:

$$\mathcal{L}_{\text{off}}(\theta) = -\mathbb{E}_{(x^{1:K}, c) \sim \mathcal{D}}\Big[ \log \prod_{k=1}^{K} \frac{\exp(\beta h_\theta(x^k, c))}{\sum_{j=k}^{K} \exp(\beta h_\theta(x^j, c))} \Big], \quad \text{(S5)}$$

where $h_\theta(x, c) = \log \frac{\pi_\theta(x|c)}{\pi_{\text{ref}}(x|c)}$. Then we have:

$$
\begin{aligned}
\mathcal{L}_{\text{off}}(\theta) &= -\mathbb{E}_{(x^{1:K}, c) \sim \mathcal{D}} \left[ \log \prod_{k=1}^{K} \frac{\exp(\beta h_\theta(x^k, c))}{\sum_{j=k}^{K} \exp(\beta h_\theta(x^j, c))} \right] \\
&= -\mathbb{E}_{c \sim \mathcal{D}, x^{1:K}} \, p_{\text{gt}}(x^{1:K}|c) \left[ \log \prod_{k=1}^{K} \frac{\exp(\beta h_\theta(x^k, c))}{\sum_{j=k}^{K} \exp(\beta h_\theta(x^j, c))} \right] \\
&= -\mathbb{E}_{c \sim \mathcal{D}, x^{1:K}} \, p_{\text{gt}}(x^{1:K}|c) \left[ \log \prod_{k=1}^{K} \frac{\exp(\beta \log \frac{\pi_\theta(x^k|c)}{\pi_{\text{ref}}(x^k|c)})}{\sum_{j=k}^{K} \exp(\beta \log \frac{\pi_\theta(x^j|c)}{\pi_{\text{ref}}(x^j|c)})} \right] \\
&= -\mathbb{E}_{c \sim \mathcal{D}, x^{1:K}} \, p_{\text{gt}}(x^{1:K}|c) \left[ \log \prod_{k=1}^{K} \frac{\exp \log(\frac{\pi_\theta(x^k|c)}{\pi_{\text{ref}}(x^k|c)})^\beta)]}{\sum_{j=k}^{K} \exp \log(\frac{\pi_\theta(x^j|c)}{\pi_{\text{ref}}(x^j|c)})^\beta} \right] \\
&= -\mathbb{E}_{c \sim \mathcal{D}, x^{1:K}} \, p_{\text{gt}}(x^{1:K}|c) \left[ \log \prod_{k=1}^{K} \underbrace{\frac{(\frac{\pi_\theta(x^k|c)}{\pi_{\text{ref}}(x^k|c)})^\beta}{\sum_{j=k}^{K} (\frac{\pi_\theta(x^j|c)}{\pi_{\text{ref}}(x^j|c)})^\beta}}_{p_\theta(x^k|c)} \right] \quad (\text{S6}) \\
&= -\mathbb{E}_{c \sim \mathcal{D}, x^{1:K}} \, p_{\text{gt}}(x^{1:K}|c) \left[ \log \underbrace{\prod_{k=1}^{K} p_\theta(x^k|c)}_{p_\theta(x^{1:K}|c)} \right] \\
&= -\mathbb{E}_{c \sim \mathcal{D}, x^{1:K}} \, p_{\text{gt}}(x^{1:K}|c) \left[ \log p_\theta(x^{1:K}|c) - \log p_{\text{gt}}(x^{1:K}|c) + \log p_{\text{gt}}(x^{1:K}|c) \right] \\
&= \mathbb{E}_{c \sim \mathcal{D}, x^{1:K}} \, p_{\text{gt}}(x^{1:K}|c) \left[ \log \frac{p_{\text{gt}}(x^{1:K}|c)}{p_\theta(x^{1:K}|c)} - \log p_{\text{gt}}(x^{1:K}|c) \right] \\
&= \mathbb{E}_{c \sim \mathcal{D}, x^{1:K}} \, D_{KL}(p_{\text{gt}}|p_\theta) - p_{\text{gt}}(x^{1:K}|c) \log p_{\text{gt}}(x^{1:K}|c)
\end{aligned}
$$

Therefore, we have:

$$
\nabla_\theta \mathcal{L}_{\text{off}}(\theta) = \mathbb{E}_{c \sim \mathcal{D}, x^{1:K}} \nabla_\theta D_{KL}(p_{\text{gt}}||p_\theta). \quad (\text{S7})
$$

The proof is completed. $\qquad \square$

### C.2 Proof of Theorem 2

*Proof.* Inspired by [53], we replace the one-hot vector in DPO with Plackett-Luce model [34], and then the online DPO can be expressed as

$$
\mathcal{L}_{\text{DPO-On}}(\theta) = -\mathbb{E}_{c \sim \mathcal{D}, x^{1:K} \sim \bar{\pi}_\theta(\cdot|c)} \left[ \sum_{k=1}^{K} p_r(x_k|c) \log \frac{(\frac{\pi_\theta(x^k|c)}{\pi_{\text{ref}}(x^k|c)})^\beta}{\sum_{j=k}^{K} (\frac{\pi_\theta(x^j|c)}{\pi_{\text{ref}}(x^j|c)})^\beta} \right], \quad (\text{S8})
$$

where $p_r(x^k_{\bar{\pi}_\theta}|c) = \frac{\exp r(x^k_{\bar{\pi}_\theta}, c)}{\sum_{i=k}^K \exp r(x^i_{\bar{\pi}_\theta}, c)}$. Then we have:

$$\mathcal{L}_{\text{on}}(\theta) = -\mathbb{E}_{c\sim\mathcal{D}, x^{1:K}\sim\bar{\pi}_\theta(\cdot|c)}\Big[\sum_{k=1}^K p_r(x_k|c) \log \frac{(\frac{\pi_\theta(x^k|c)}{\pi_{\text{ref}}(x^k|c)})^\beta}{\sum_{j=k}^K (\frac{\pi_\theta(x^j|c)}{\pi_{\text{ref}}(x^j|c)})^\beta}\Big]$$

$$= -\mathbb{E}_{c\sim\mathcal{D}}\, p_{\bar{\pi}_\theta}(x^{1:K}|c)\Big[\sum_{k=1}^K p_r(x_k|c) \log \frac{(\frac{\pi_\theta(x^k|c)}{\pi_{\text{ref}}(x^k|c)})^\beta}{\sum_{j=k}^K (\frac{\pi_\theta(x^j|c)}{\pi_{\text{ref}}(x^j|c)})^\beta}\Big]$$

$$= -\mathbb{E}_{c\sim\mathcal{D}}\, p_{\bar{\pi}_\theta}(x^{1:K}|c)\Big[\sum_{k=1}^K p_r(x^k|c) \log \underbrace{\frac{(\frac{\pi_\theta(x^k|c)}{\pi_{\text{ref}}(x^k|c)})^\beta}{\sum_{j=k}^K (\frac{\pi_\theta(x^j|c)}{\pi_{\text{ref}}(x^j|c)})^\beta}}_{p_\theta(x^k|c)}\Big] \quad\quad (S9)$$

$$= -\mathbb{E}_{c\sim\mathcal{D}}\, p_{\bar{\pi}_\theta}(x^{1:K}|c)\Big[\sum_{k=1}^K p_r(x^k|c) \log p_\theta(x^k|c)\Big]$$

$$= -\mathbb{E}_{c\sim\mathcal{D}}\, p_{\bar{\pi}_\theta}(x^{1:K}|c)\Big[\sum_{k=1}^K p_r(x^k|c)(\log p_\theta(x^k|c) - \log p_r(x^k|c) + \log p_r(x^k|c))\Big]$$

$$= \mathbb{E}_{c\sim\mathcal{D}}\, p_{\bar{\pi}_\theta}(x^{1:K}|c)\Big[D_{KL}(p_r|p_\theta) - p_r(x^k|c) \log p_r(x^k|c)\Big]$$

Therefore, we have:

$$\nabla_\theta \mathcal{L}_{\text{on}}(\theta) = \mathbb{E}_{c\sim\mathcal{D}}\nabla_\theta\, p_{\bar{\pi}_\theta}(x^{1:K}|c)D_{KL}(p_r||p_\theta). \quad\quad (S10)$$

The proof is completed. $\square$

Given a sample $x$ with a tiny generative probability $p_{\bar{\pi}_\theta|c}(x) \to 0$, and large reward value $r(x, c) \to 1$, we have $\lim_{p_{\pi_\theta}(x|c)\to 0, r(x,c)\to 1} \nabla_\theta \mathcal{L}_{\text{on}} = \mathbf{0}$.

*Proof.* Since $x$ is contained in the sampled motion group $x^{1:K}$, we have:

$$\lim_{p_{\pi_\theta}(x|c)\to 0, r(x,c)\to 1} \nabla_\theta \mathcal{L}_{\text{on}}$$

$$= \lim_{p_{\pi_\theta}(x|c)\to 0, r(x,c)\to 1} \nabla_\theta\, p_{\bar{\pi}_\theta}(x^{1:K}|c)D_{KL}(p_r||p_\theta)$$

$$\overset{\textcircled{1}}{=} \lim_{p_{\pi_\theta}(x^{1:K}|c)\to 0, r(x,c)\to 1} \nabla_\theta\, p_{\bar{\pi}_\theta}(x^{1:K}|c)D_{KL}(p_r||p_\theta) \quad\quad (S11)$$

$$= \mathbf{0},$$

where $\textcircled{1}$ holds since $p_{\pi_\theta}(x^{1:K}|c) = p_{\pi_\theta}(x|c)p_{\pi_\theta}(x^M|c) \le p_{\pi_\theta}(x|c)$, and $x^M$ denotes a motion group obtained by removing the given motion $x$ from the group $x^{1:K}$, i.e., satisfying that $x^M = x^{1:K} - \{x\}$. The proof is completed. $\square$

## C.3 Proof of DSoPo

*Proof.* Eq. (10) suggests that DSoPo samples multiple unpreferred motion candidates instead of a single unpreferred motion. Thus, we should first extend Eq. (9) as:

$$\mathcal{L}_{\text{DSoPo}}(\theta) = -\mathbb{E}_{(x^w, c)\sim\mathcal{D}}\mathbb{E}_{x^{1:K}\sim\bar{\pi}_\theta(x|c)} \log \sigma\Big(\beta\mathcal{H}_\theta(x^w, x^l, c)\Big), \quad\quad (S12)$$

where $x^l = \arg\min_{\{x_{\bar\pi_\theta}^k\}_{k=1}^K \sim \pi_\theta} r(x_{\pi_\theta}^k, c)$. Then, we have:

$$
\begin{aligned}
\mathcal{L}_{\mathrm{DSoPo}}(\theta) &= -\,\mathbb{E}_{(x^w,c)\sim\mathcal{D}}\mathbb{E}_{x^{1:K}\sim\bar\pi_\theta(x|c)}\log\sigma\Big(\beta\mathcal{H}_\theta(x^w,x^l,c)\Big) \\
&= -\,\mathbb{E}_{(x^w,c)\sim\mathcal{D}}\mathbb{E}_{x^{1:K}}\underbrace{p_{\bar\pi_\theta}(x^{1:K}|c)}_{\text{Substituting with (11)}}\log\sigma\Big(\beta\mathcal{H}_\theta(x^w,x^l,c)\Big) \\
&= -\,\mathbb{E}_{(x^w,c)\sim\mathcal{D}}\mathbb{E}_{x^{1:K}}\Big(p_{\bar\pi_\theta}(x_{\pi_\theta}^{1:K}|c)p_\tau(r(x^l,c)\geq\tau)+p_{\bar\pi_\theta}(x_{\pi_\theta}^{1:K}|c)p_\tau(r(x^l,c)<\tau)\Big)\log\sigma\Big(\beta\mathcal{H}_\theta(x^w,x^l,c)\Big) \\
&= -\,\mathbb{E}_{(x^w,c)\sim\mathcal{D}}\mathbb{E}_{x^{1:K}}\underbrace{p_{\bar\pi_\theta}(x_{\pi_\theta}^{1:K}|c)p_\tau(r(x^l,c)\geq\tau)}_{p_{\pi_\theta}^{hu}(x^{1:K}|c)}\log\sigma\Big(\beta\mathcal{H}_\theta(x^w,x^l,c)\Big) \\
&\quad -\,\mathbb{E}_{(x^w,c)\sim\mathcal{D}}\mathbb{E}_{x^{1:K}}\underbrace{p_{\bar\pi_\theta}(x_{\pi_\theta}^{1:K}|c)p_\tau(r(x^l,c)<\tau)}_{p_{\pi_\theta}^{vu}(x^{1:K}|c)}\log\sigma\Big(\beta\mathcal{H}_\theta(x^w,x^l,c)\Big) \\
&= -\,\mathbb{E}_{(x^w,c)\sim\mathcal{D}}\mathbb{E}_{x^{1:K}}Z_{hu}(c)p_{\pi_\theta}^{hu}(x^{1:K}|c)\log\sigma\Big(\beta\mathcal{H}_\theta(x^w,x^l,c)\Big) \\
&\quad -\,\mathbb{E}_{(x^w,c)\sim\mathcal{D}}\mathbb{E}_{x^{1:K}}Z_{vu}(c)p_{\pi_\theta}^{vu*}(x^{1:K}|c)\log\sigma\Big(\beta\mathcal{H}_\theta(x^w,x^l,c)\Big) \\
&= -\,\mathbb{E}_{(x^w,c)\sim\mathcal{D}}Z_{hu}(c)\mathbb{E}_{x^{1:K}}p_{\pi_\theta}^{hu*}(x^{1:K}|c)\log\sigma\Big(\beta\mathcal{H}_\theta(x^w,x^l,c)\Big) \\
&\quad -\,\mathbb{E}_{(x^w,c)\sim\mathcal{D}}Z_{vu}(c)\mathbb{E}_{x^{1:K}}p_{\pi_\theta}^{vu*}(x^{1:K}|c)\log\sigma\Big(\beta\mathcal{H}_\theta(x^w,x^l,c)\Big) \\
&= -\,\mathbb{E}_{(x^w,c)\sim\mathcal{D}}Z_{hu}(c)\mathbb{E}_{x^{1:K}\sim\bar\pi_\theta^{hu*}}\log\sigma\Big(\beta\mathcal{H}_\theta(x^w,x^l,c)\Big) \\
&\quad -\,\mathbb{E}_{(x^w,c)\sim\mathcal{D}}Z_{vu}(c)\mathbb{E}_{x^{1:K}\sim\bar\pi_\theta^{vu*}}\log\sigma\Big(\beta\mathcal{H}_\theta(x^w,x^l,c)\Big) \\
&= \mathcal{L}_{\mathrm{vu}}(\theta) + \mathcal{L}_{\mathrm{hu}}(\theta),
\end{aligned}
\tag{S13}
$$

where $p_{\bar\pi_\theta^{vu*}}(\cdot) = \frac{p_{\bar\pi_\theta}^{vu}(\cdot)}{Z_{vu}(c)}$ and $p_{\bar\pi_\theta^{hu*}}(\cdot) = \frac{p_{\bar\pi_\theta}^{hu}(\cdot)}{Z_{hu}(c)}$ respectively denote the distributions of valuable unpreferred and high-preference unpreferred motions. The proof is completed. $\square$

Accordingly, we rewrite $\mathcal{L}_{\mathrm{hu}}(\theta)$ and obtain the objective function of USoPo:

$$
\begin{aligned}
\mathcal{L}_{\mathrm{USoPo-hu}}(\theta) &= -\mathbb{E}_{(x^w,c)\sim\mathcal{D}}Z_{hu}(c)\log\sigma\Big(\beta h_\theta(x^w,c)\Big), \\
\mathcal{L}_{\mathrm{USoPo}}(\theta) &= \mathcal{L}_{\mathrm{USoPo-hu}}(\theta) + \mathcal{L}_{\mathrm{vu}}(\theta).
\end{aligned}
\tag{S14}
$$

**Implementation**  Now, we discuss how to deal with the computation of $Z_{vu}(c)$ and $Z_{hu}(c)$ in our implementation. As discussed in Sec. B, directly optimizing the objective function $\mathcal{L}_{\mathrm{SoPo}}^{\mathrm{diff}}(\theta)$ is challenging, and we used **Algorithm 1** optimized its equivalent form:

$$
\mathcal{L}_{\mathrm{SoPo}}^{\mathrm{diff}}(\theta) = -\mathbb{E}_{t\sim\mathcal{U}(0,T),(x^w,c)\sim\mathcal{D},x_{\bar\pi_\theta}^{1:K}\sim\bar\pi_\theta(\cdot|c)}
$$
$$
\begin{cases}
\log\sigma\Big(-T\omega_t\big(\beta_w(x_w)\big(\mathcal{L}(\theta,\mathrm{ref},x_t^w)-\beta\mathcal{L}(\theta,\mathrm{ref},x_t^l))\big)\Big), & \text{if } r(x^l,c)<\tau, \\
\log\sigma\Big(-T\omega_t\beta_w(x_w)\mathcal{L}(\theta,\mathrm{ref},x_t^w)\Big), & \text{otherwise.}
\end{cases}
\tag{S15}
$$

Similarly, we can optimize the equivalent form of UDoPo to avoid the computation of $Z_{vu}(c)$ and $Z_{hu}(c)$:

$$
\mathcal{L}_{\mathrm{USoPo}}(\theta) = -\mathbb{E}_{(x^w,c)\sim\mathcal{D},x_{\bar\pi_\theta}^{1:K}\sim\bar\pi_\theta(\cdot|c)}
\begin{cases}
\log\sigma\Big(\beta\mathcal{H}_\theta(x^w,x^l,c)\Big), & \text{If } r(x^l,c)<\tau, \\
\log\sigma\Big(\beta h_\theta(x^w,c)\Big), & \text{Otherwise.}
\end{cases}
\tag{S16}
$$

The proof of Eq. (S16) follows the same steps as the proof of Eq. (S15) in Sec. B.

### C.4  Discussion of USoPo and DSoPo

In this section, we discuss the relationship between USoPo and DSoPo and the difference between their optimization. Here, USoPo and DSoPo are defined as:

$$
\mathcal{L}_{\mathrm{USoPo}}(\theta) = -\mathbb{E}_{(x^w,c)\sim\mathcal{D}}Z_{hu}(c)\log\sigma\Big(\beta h_\theta(x^w,c)\Big) + \mathcal{L}_{\mathrm{vu}}(\theta).
\tag{S17}
$$

$$\mathcal{L}_{\text{DSoPo}}(\theta) = \mathcal{L}_{\text{vu}}(\theta) + \mathcal{L}_{\text{hu}}(\theta), \tag{S18}$$

**Relationship between USoPo and DSoPo**   We begin by analyzing the size relationship between USoPo and DSoPo:

$$\mathcal{L}_{\text{DSoPo}}(\theta) - \mathcal{L}_{\text{USoPo}}(\theta)$$

$$= \mathcal{L}_{\text{hu}}(\theta) + \mathbb{E}_{(x^w,c)\sim\mathcal{D}} Z_{hu}(c) \log \sigma\Big(\beta h_\theta(x^w,c)\Big)$$

$$= -\mathbb{E}_{(x^w,c)\sim\mathcal{D}} Z_{hu}(c) \mathbb{E}_{x^{1:K}\sim\bar{\pi}_\theta^{hu*}} \log \sigma\Big(\beta\mathcal{H}_\theta(x^w,x^l,c)\Big) + \mathbb{E}_{(x^w,c)\sim\mathcal{D}} Z_{hu}(c) \log \sigma\Big(\beta h_\theta(x^w,c)\Big)$$

$$= -\mathbb{E}_{(x^w,c)\sim\mathcal{D}} Z_{hu}(c) \mathbb{E}_{x^{1:K}\sim\bar{\pi}_\theta^{hu*}} \Big[ \log \sigma\Big(\beta\mathcal{H}_\theta(x^w,x^l,c)\Big) - \log \sigma\Big(\beta h_\theta(x^w,c)\Big) \Big]. \tag{S19}$$

Considering that $\mathcal{H}_\theta(x^w,x^l,c) = h_\theta(x^w,c) - h_\theta(x^l,c)$ and $h_\theta(x,c) = \log \frac{\pi_\theta(x|c)}{\pi_{\text{ref}}(x|c)}$, we have:

$$\mathcal{L}_{\text{DSoPo}}(\theta) - \mathcal{L}_{\text{USoPo}}(\theta)$$

$$= -\mathbb{E}_{(x^w,c)\sim\mathcal{D}} Z_{hu}(c) \mathbb{E}_{x^{1:K}\sim\bar{\pi}_\theta^{hu*}} \Big[ \log \sigma\Big(\beta\mathcal{H}_\theta(x^w,x^l,c)\Big) - \log \sigma\Big(\beta h_\theta(x^w,c)\Big) \Big]$$

$$= -\mathbb{E}_{(x^w,c)\sim\mathcal{D}} Z_{hu}(c) \mathbb{E}_{x^{1:K}\sim\bar{\pi}_\theta^{hu*}} \Big[ \log \frac{\exp \beta h_\theta(x^w,c)}{\exp \beta h_\theta(x^w,c) + \exp \beta h_\theta(x^l,c)} - \log \frac{\exp \beta h_\theta(x^w,c)}{\exp \beta h_\theta(x^w,c) + 1} \Big]$$

$$= -\mathbb{E}_{(x^w,c)\sim\mathcal{D}} Z_{hu}(c) \mathbb{E}_{x^{1:K}\sim\bar{\pi}_\theta^{hu*}} \Big[ \log \frac{\exp \beta h_\theta(x^w,c) + 1}{\exp \beta h_\theta(x^w,c) + \exp \beta h_\theta(x^l,c)} \Big]$$

$$= -\mathbb{E}_{(x^w,c)\sim\mathcal{D}} Z_{hu}(c) \mathbb{E}_{x^{1:K}\sim\bar{\pi}_\theta^{hu*}} \Big[ \log \frac{(\frac{\pi_\theta(x^w|c)}{\pi_{\text{ref}}(x^w|c)})^\beta + 1}{(\frac{\pi_\theta(x^w|c)}{\pi_{\text{ref}}(x^w|c)})^\beta + (\frac{\pi_\theta(x^l|c)}{\pi_{\text{ref}}(x^l|c)})^\beta} \Big]. \tag{S20}$$

In general, DPO focuses on reducing the generative probability of loss samples (unpreferred motions). Consequently, the generative probability of the policy model $\pi_\theta(x^l|c)$ will be lower than that of the reference model $\pi_{\text{ref}}(x^l|c)$, i.e., $\pi_\theta(x^l|c) \le \pi_{\text{ref}}(x^l|c)$, resulting in $\frac{\pi_\theta(x^l|c)}{\pi_{\text{ref}}(x^l|c)} \le 1$. Hence, the following relationship holds:

$$\frac{\pi_\theta(x^l|c)}{\pi_{\text{ref}}(x^l|c)} \le 1$$

$$\Rightarrow (\frac{\pi_\theta(x^w|c)}{\pi_{\text{ref}}(x^w|c)})^\beta + 1 \ge (\frac{\pi_\theta(x^w|c)}{\pi_{\text{ref}}(x^w|c)})^\beta + (\frac{\pi_\theta(x^l|c)}{\pi_{\text{ref}}(x^l|c)})^\beta$$

$$\Rightarrow \frac{(\frac{\pi_\theta(x^w|c)}{\pi_{\text{ref}}(x^w|c)})^\beta + 1}{(\frac{\pi_\theta(x^w|c)}{\pi_{\text{ref}}(x^w|c)})^\beta + (\frac{\pi_\theta(x^l|c)}{\pi_{\text{ref}}(x^l|c)})^\beta} \ge 1$$

$$\Rightarrow \log \frac{(\frac{\pi_\theta(x^w|c)}{\pi_{\text{ref}}(x^w|c)})^\beta + 1}{(\frac{\pi_\theta(x^w|c)}{\pi_{\text{ref}}(x^w|c)})^\beta + (\frac{\pi_\theta(x^l|c)}{\pi_{\text{ref}}(x^l|c)})^\beta} \ge 0 \tag{S21}$$

$$\Rightarrow \underbrace{-\mathbb{E}_{(x^w,c)\sim\mathcal{D}} Z_{hu}(c) \mathbb{E}_{x^{1:K}\sim\bar{\pi}_\theta^{hu*}} \Big[ \log \frac{(\frac{\pi_\theta(x^w|c)}{\pi_{\text{ref}}(x^w|c)})^\beta + 1}{(\frac{\pi_\theta(x^w|c)}{\pi_{\text{ref}}(x^w|c)})^\beta + (\frac{\pi_\theta(x^l|c)}{\pi_{\text{ref}}(x^l|c)})^\beta} \Big]}_{\mathcal{L}_{\text{DSoPo}}(\theta) - \mathcal{L}_{\text{USoPo}}(\theta)} \le 0$$

$$\Rightarrow \mathcal{L}_{\text{DSoPo}}(\theta) \le \mathcal{L}_{\text{USoPo}}(\theta).$$

Eq. (S21) indicates that $\mathcal{L}_{\text{USoPo}}$ is one of upper bounds of $\mathcal{L}_{\text{DSoPo}}$.

**Difference between the optimization of USoPo and DSoPo**   The difference between the optimization of USoPo and DSoPo can be measured by that between their objective function. Let $\mathcal{L}_{\text{d}}(\theta) = \mathcal{L}_{\text{USoPo}}(\theta) - \mathcal{L}_{\text{DSoPo}}(\theta)$, the difference between their objective function can be denoted as:

$$\mathcal{L}_{\text{d}}(\theta) = \mathcal{L}_{\text{USoPo}}(\theta) - \mathcal{L}_{\text{DSoPo}}(\theta)$$

$$= \mathbb{E}_{(x^w,c)\sim\mathcal{D}} Z_{hu}(c) \mathbb{E}_{x^{1:K}\sim\bar{\pi}_\theta^{hu*}} \Big[ \log \frac{(\frac{\pi_\theta(x^w|c)}{\pi_{\text{ref}}(x^w|c)})^\beta + 1}{(\frac{\pi_\theta(x^w|c)}{\pi_{\text{ref}}(x^w|c)})^\beta + (\frac{\pi_\theta(x^l|c)}{\pi_{\text{ref}}(x^l|c)})^\beta} \Big] \overset{\text{①}}{\ge} 0 \tag{S22}$$

where ① holds due to Eq. (S21). As discussed above, the generative probability of the policy model $\pi_\theta(x^l|c)$ will be lower than that of the reference model $\pi_{\text{ref}}(x^l|c)$, and thus $\pi_\theta(x^l|c)$ falls in the range between 0 and $\pi_{\text{ref}}(x^l|c)$, i.e., $0 \le \pi_\theta(x^l|c) \le \pi_{\text{ref}}(x^l|c)$.

Assuming that the value of $\pi_\theta(x^w|c)$ is fixed, the value of $\mathcal{L}_d(\theta)$ is negatively correlated with $\pi_\theta(x^l|c)$, since we have:

$$
\begin{aligned}
\nabla_\theta \mathcal{L}_d(\theta) =& \nabla_\theta - \mathbb{E}_{(x^w,c)\sim\mathcal{D}} Z_{hu}(c) \mathbb{E}_{x^{1:K}\sim\bar{\pi}_\theta^{hu*}} \Big[ \log \frac{(\frac{\pi_\theta(x^w|c)}{\pi_{\text{ref}}(x^w|c)})^\beta + 1}{(\frac{\pi_\theta(x^w|c)}{\pi_{\text{ref}}(x^w|c)})^\beta + (\frac{\pi_\theta(x^l|c)}{\pi_{\text{ref}}(x^l|c)})^\beta} \Big] \\
=& \mathbb{E}_{(x^w,c)\sim\mathcal{D}} Z_{hu}(c) \mathbb{E}_{x^{1:K}\sim\bar{\pi}_\theta^{hu*}} \nabla_\theta - \log \Big[ (\frac{\pi_\theta(x^w|c)}{\pi_{\text{ref}}(x^w|c)})^\beta + (\frac{\pi_\theta(x^l|c)}{\pi_{\text{ref}}(x^l|c)})^\beta \Big] \\
=& \mathbb{E}_{(x^w,c)\sim\mathcal{D}} Z_{hu}(c) \mathbb{E}_{x^{1:K}\sim\bar{\pi}_\theta^{hu*}} \frac{1}{(\frac{\pi_\theta(x^w|c)}{\pi_{\text{ref}}(x^w|c)})^\beta + (\frac{\pi_\theta(x^l|c)}{\pi_{\text{ref}}(x^l|c)})^\beta} - \nabla_\theta (\frac{\pi_\theta(x^l|c)}{\pi_{\text{ref}}(x^l|c)})^\beta \\
\overset{①}{\sim}& - \nabla_\theta (\frac{\pi_\theta(x^l|c)}{\pi_{\text{ref}}(x^l|c)})^\beta .
\end{aligned}
\tag{S23}
$$

where ① holds since $\frac{1}{(\frac{\pi_\theta(x^w|c)}{\pi_{\text{ref}}(x^w|c)})^\beta + (\frac{\pi_\theta(x^l|c)}{\pi_{\text{ref}}(x^l|c)})^\beta} > 0$.

Hence, when the generative probability of unpreferred motions $\pi_\theta(x^l|c)$ is lower, the difference between the optimization of USoPo and DSoPo is larger. However, the unpreferred motions are sampled from the relatively high-preference distribution $\pi_\theta^{hu*}$, and thus should not be treated as unpreferred motions. Using $\mathcal{L}_{\text{USoPo}}(\theta)$ to optimize policy model $\pi_\theta$ instead of $\mathcal{L}_{\text{DSoPo}}(\theta)$ can avoid unnecessary optimization of these relatively high-preference unpreferred motion $\mathcal{L}_d(\theta)$.

### C.5 Proof of Eq. (16)

Before proving Eq. (16), we first present some useful lemmas from [27].

**Lemma 1.** *[27] Given a winning sample $x_w$ and a losing sample $x_l$, the DPO denoted as*

$$
\mathcal{L}_{\text{DPO}}(\theta) = \mathbb{E}_{(x^w,x^l,c)\sim\mathcal{D}} \Big[ -\log \sigma \Big( \beta \log \frac{\pi_\theta(x^w|c)}{\pi_{ref}(x^w|c)} - \beta \log \frac{\pi_\theta(x^l|c)}{\pi_{\text{ref}}(x^l|c)} \Big) \Big].
\tag{S24}
$$

*Then the objective function for diffusion models can be denoted as:*

$$
\begin{aligned}
\mathcal{L}_{\text{DPO-Diffusion}}(\theta) = & -\mathbb{E}_{(x_0^w,x_0^l)\sim\mathcal{D}} \log \sigma \big( \beta \mathbb{E}_{x_{1:T}^w\sim\pi_\theta(x_{1:T}^w|x_0^w), x_{1:T}^l\sim\pi_\theta(x_{1:T}^l|x_0^l)} \\
& [\log \frac{\pi_\theta(x_{0:T}^w)}{\pi_{\text{ref}}(x_{0:T}^w)} - \log \frac{\pi_\theta(x_{0:T}^l)}{\pi_{\text{ref}}(x_{0:T}^l)}] ),
\end{aligned}
\tag{S25}
$$

*where $x_t^*$ denoted the noised sample $x^*$ for the $t$-th step.*

**Lemma 2.** *[27] Given the objective function of diffusion-based DPO denoted as Eq. (S25), it has an upper bound $\mathcal{L}_{\text{UB}}(\theta)$:*

$$
\begin{aligned}
\mathcal{L}_{\text{DPO-Diffusion}}(\theta) \le & -\mathbb{E}_{(x_0^w,x_0^l)\sim\mathcal{D}, t\sim\mathcal{U}(0,T), x_{t-1,t}^w\sim\pi_\theta(x_{t-1,t}^w|x_0^w), x_{t-1,t}^l\sim\pi_\theta(x_{t-1,t}^l|x_0^l)} \log \sigma \\
& \underbrace{\Big( \beta T \log \frac{\pi_\theta(x_{t-1}^w|x_t^w)}{\pi_{\text{ref}}(x_{t-1}^w|x_t^w)} - \beta T \log \frac{\pi_\theta(x_{t-1}^l|x_t^l)}{\pi_{\text{ref}}(x_{t-1}^l|x_t^l)} \Big)}_{\mathcal{L}_{\text{UB}}(\theta)},
\end{aligned}
\tag{S26}
$$

*where $T$ denotes the number of diffusion steps.*

**Lemma 3.** *[27] Given the objective function for diffusion model denoted as Eq. (S26), it can be rewritten as :*

$$
\begin{aligned}
\mathcal{L}_{\text{UB}}(\theta) = & -\mathbb{E}_{(x_0^w,x_0^l)\sim\mathcal{D}, t\sim\mathcal{U}(0,T), x_t^w\sim q(x_t^w|x_0^w), x_t^l\sim q(x_t^l|x_0^l)} \log \sigma(-\beta T \omega_t \\
& (\|\epsilon - \epsilon_\theta(x_t^w,t)\|_2^2 - \|\epsilon - \epsilon_{\text{ref}}(x_t^w,t)\|_2^2 - (\|\epsilon - \epsilon_\theta(x_t^l,t)\|_2^2 - \|\epsilon - \epsilon_{\text{ref}}(x_t^l,t)\|_2^2))),
\end{aligned}
\tag{S27}
$$

*where $x_t^* = \alpha_t x_0^* + \sigma_t \epsilon$, $\epsilon \sim \mathcal{N}(0,\mathbb{I})$ is a draw from the distribution of forward process $q(x_t^*|x_0^*)$.*

Now, we proof Eq. (16) based on these lemmas.

*Proof.* This proof has three steps. In each step, we apply the three lemmas introduced above in succession. We begin with the loss function of SoPo for probability models:

$$\mathcal{L}_{\text{SoPo}}(\theta) = \underbrace{-\mathbb{E}_{(x^w,c)\sim\mathcal{D},x_{\bar{\pi}_\theta}^{1:K}\sim\bar{\pi}_\theta^{vu*}(\cdot|c)} Z_{vu}(c)\Big[\log\sigma\Big(\beta_w(x^w)h_\theta(x^w,c) - \beta h_\theta(x^l,c)\Big)\Big]}_{\mathcal{L}_{\text{SoPo}-vu}(\theta)}$$

$$\underbrace{-\mathbb{E}_{(x^w,c)\sim\mathcal{D}} Z_{hu}(c)\log\sigma\Big(\beta_w(x^w)h_\theta(x^w,c)\Big)}_{\mathcal{L}_{\text{SoPo}-hu}(\theta)}.$$

(S28)

Based on **Lemma 1**, we can rewrite the objective function for diffusion models:

$$\mathcal{L}_{\text{SoPo}-\text{Diffusion}}(\theta) = \mathcal{L}_{\text{SoPo}-vu}^{\text{diff}-\text{ori}}(\theta) + \mathcal{L}_{\text{SoPo}-hu}^{\text{diff}-\text{ori}}(\theta)$$

$$\mathcal{L}_{\text{SoPo}-vu}^{\text{diff}-\text{ori}}(\theta) = -\mathbb{E}_{(x_0^w,c)\sim\mathcal{D},x_0^{1:K}\sim\bar{\pi}_\theta^{vu*}(\cdot|c)} Z_{vu}(c)$$

$$\log\sigma\big(\mathbb{E}_{x_{1:T}^w\sim\pi_\theta(x_{1:T}^w|x_0^w),x_{1:T}^l\sim\pi_\theta(x_{1:T}^l|x_0^l)}[\beta_w(x_0^w)\log\frac{\pi_\theta(x_{0:T}^w)}{\pi_{\text{ref}}(x_{0:T}^w)} - \beta\log\frac{\pi_\theta(x_{0:T}^l)}{\pi_{\text{ref}}(x_{0:T}^l)}]\big),$$

$$\mathcal{L}_{\text{SoPo}-hu}^{\text{diff}-\text{ori}}(\theta) = -\mathbb{E}_{(x_0^w,c)\sim\mathcal{D}} Z_{hu}(c)\log\sigma\big(\mathbb{E}_{x_{1:T}^w\sim\pi_\theta(x_{1:T}^w|x_0^w)}[\beta_w(x^w)\log\frac{\pi_\theta(x_{0:T}^w)}{\pi_{\text{ref}}(x_{0:T}^w)}]\big),$$

(S29)

where $x_t^*$ denoted the noised sample $x^*$ for the $t$-th step. According to **Lemma 2**, the upper bound of $\mathcal{L}_{\text{SoPo}-vu}^{\text{diff}-\text{ori}}(\theta)$ and $\mathcal{L}_{\text{SoPo}-hu}^{\text{diff}-\text{ori}}(\theta)$ can be denoted as:

$$\mathcal{L}_{\text{SoPo}-vu}^{\text{diff}-\text{ori}}(\theta) \leq -\mathbb{E}_{(x_0^w,c)\sim\mathcal{D},x_0^{1:K}\sim\bar{\pi}_\theta^{vu*}(\cdot|c),t\sim\mathcal{U}(0,T),x_{t-1,t}^w\sim\pi_\theta(x_{t-1,t}^w|x_0^w),x_{t-1,t}^l\sim\pi_\theta(x_{t-1,t}^l|x_0^l)}$$

$$\underbrace{\log\sigma\left(\beta_w(x_0^w)T\log\frac{\pi_\theta(x_{t-1}^w|x_t^w)}{\pi_{\text{ref}}(x_{t-1}^w|x_t^w)} - \beta T\log\frac{\pi_\theta(x_{t-1}^l|x_t^l)}{\pi_{\text{ref}}(x_{t-1}^l|x_t^l)}\right)}_{\mathcal{L}_{\text{SoPo}-vu}^{\text{diff}}(\theta)},$$

$$\mathcal{L}_{\text{SoPo}-hu}^{\text{diff}-\text{ori}}(\theta) \leq -\mathbb{E}_{(x_0^w,c)\sim\mathcal{D},t\sim\mathcal{U}(0,T),x_{t-1,t}^w\sim\pi_\theta(x_{t-1,t}^w|x_0^w)}\underbrace{\log\sigma\left(\beta_w(x_0^w)T\log\frac{\pi_\theta(x_{t-1}^w|x_t^w)}{\pi_{\text{ref}}(x_{t-1}^w|x_t^w)}\right)}_{\mathcal{L}_{\text{SoPo}-hu}^{\text{diff}}(\theta)},$$

$$\mathcal{L}_{\text{SoPo}-\text{Diffusion}}(\theta) = \mathcal{L}_{\text{SoPo}-vu}^{\text{diff}-\text{ori}}(\theta) + \mathcal{L}_{\text{SoPo}-hu}^{\text{diff}-\text{ori}}(\theta) \leq \mathcal{L}_{\text{SoPo}-vu}^{\text{diff}}(\theta) + \mathcal{L}_{\text{SoPo}-hu}^{\text{diff}}(\theta) = \mathcal{L}_{\text{SoPo}}^{\text{diff}}(\theta).$$

(S30)

Applying **Lemma 3** to $\mathcal{L}_{\text{SoPo}-vu}^{\text{diff}}(\theta)$ and $\mathcal{L}_{\text{SoPo}-hu}^{\text{diff}}(\theta)$ , we have

$$\mathcal{L}_{\text{SoPo}-vu}^{\text{diff}}(\theta) = -\mathbb{E}_{(x_0^w,c)\sim\mathcal{D},x_0^{1:K}\sim\bar{\pi}_\theta^{vu*}(\cdot|c),t\sim\mathcal{U}(0,T),x_t^w\sim q(x_t^w|x_0^w),x_t^l\sim q(x_t^l|x_0^l)}$$

$$\log\sigma\bigg(-T\omega_t\Big(\beta_w(x_0^w)\big(\|\epsilon - \epsilon_\theta(x_t^w,t)\|_2^2 - \|\epsilon - \epsilon_{\text{ref}}(x_t^w,t)\|_2^2\big)$$

$$-\beta\big(\|\epsilon - \epsilon_\theta(x_t^l,t)\|_2^2 - \|\epsilon - \epsilon_{\text{ref}}(x_t^l,t)\|_2^2\big)\Big)\bigg),$$

(S31)

$$\mathcal{L}_{\text{SoPo}-hu}^{\text{diff}}(\theta) = -\mathbb{E}_{(x_0^w,c)\sim\mathcal{D},t\sim\mathcal{U}(0,T),x_{t-1,t}^w\sim\pi_\theta(x_{t-1,t}^w|x_0^w)}$$

$$\log\sigma\bigg(-T\omega_t\beta_w(x_0^w)\big(\|\epsilon - \epsilon_\theta(x_t^w,t)\|_2^2 - \|\epsilon - \epsilon_{\text{ref}}(x_t^w,t)\|_2^2\big)\bigg),$$

$$\mathcal{L}_{\text{SoPo}}^{\text{diff}}(\theta) = \mathcal{L}_{\text{SoPo}-vu}^{\text{diff}}(\theta) + \mathcal{L}_{\text{SoPo}-hu}^{\text{diff}}(\theta)$$

To simplify the symbolism, the objective functions can be rewritten as:

$$\mathcal{L}_{\text{SoPo}-vu}^{\text{diff}} = -\mathbb{E}_{t\sim\mathcal{U}(0,T),(x^w,c)\sim\mathcal{D},x_{\bar{\pi}_\theta}^{1:K}\sim\bar{\pi}_\theta^{vu*}(\cdot|c)} Z_{vu}(c)$$

$$\Big[\log\sigma\Big(-T\omega_t\big(\beta_w(x_w)\big(\mathcal{L}(\theta,\text{ref},x_t^w) - \beta\mathcal{L}(\theta,\text{ref},x_t^l)\big)\big)\Big)\Big],$$

$$\mathcal{L}_{\text{SoPo}-hu}^{\text{diff}} = -\mathbb{E}_{t\sim\mathcal{U}(0,T),(x^w,c)\sim\mathcal{D}} Z_{hu}(c)$$

$$\Big[\log\sigma\Big(-T\omega_t\beta_w(x_w)\mathcal{L}(\theta,\text{ref},x_t^w)\Big)\Big],$$

(S32)

where $\mathcal{L}(\theta,\text{ref},x_t) = \mathcal{L}(\theta,x_t) - \mathcal{L}(\text{ref},x_t)$, and $\mathcal{L}(\theta/\text{ref},x_t) = \|\epsilon_{\theta/\text{ref}}(x_t,t) - \epsilon\|_2^2$ denotes the loss of the policy or reference model. The proof is completed. □

# D  More Related Works

Fine-tuning pre-trained diffusion models [54–57] using task-specific reward functions [46, 48] is a widely adopted approach for adapting models to specific downstream tasks. Current approaches are broadly classified into three mechanisms: those relying on differentiable rewards [54, 57], conventional reinforcement learning algorithms [58, 59], and Direct Preference Optimization (DPO) [25]. Our work is most closely related to methods based on DPO, which provides a remarkably straightforward path to align the model with specific downstream objectives by directly utilizing pairs of motions reflecting human judgments. Recently, some research focus on the issues of fine-grained human preference [60], visually consistence [61], and personalized preference [62] for image generation. As a powerful alignment method, DPO is also extended to video generation [63], 3D generation [64], and audio generation [65].

