# OpenReview forum: "SoPo: Text-to-Motion Generation Using Semi-Online Preference Optimization"
_NeurIPS.cc/2025/Conference — NeurIPS 2025 poster_

### Official Review · Reviewer_7xd7 · 2025-06-04

**Clarity:** 3
**Significance:** 1
**Originality:** 2
**Rating:** 4
**Confidence:** 4

**Summary:**

In SoPo, the authors introduce a novel preference optimization approach.
Online-generated samples are generated by the model, filtered by a reward function to serve as unpreferred data.
Offline samples are designated as preferred data.
This combination addresses the issues with existing pure offline DPO (overfitting) and pure online DPO methods (biased sampling).

**Questions:**

1. Appendix (s6): The 5th line of the equation in line 44 has unmatched “)]”.
2. line 102: Here pθ … with represents -> which represents
3. Is the reward function just the TMR similarity score? What happens if TMR is overfitting?
I think it will be necessary to test novel or out of distribution text and motion samples requested by the users.

4. There’re a lot of engineering hyperparameters, which are “tuned through preliminary experiments”. I wonder how sensitive they are?

**Ethical Concerns:**

["NO or VERY MINOR ethics concerns only"]

**Final Justification:**

Please refer to my below comments.

Despite some concerns remain, the overall quality and novelty of the submission out shine the drawbacks.

Therefore for the final rating, I recommend acceptance for this paper.

**Limitations:**

The authors address some limitations in the reward model, but as mentioned above, there are other limitations unaddressed in the paper.

**Paper Formatting Concerns:**

I didn't notice any formatting concerns

**Quality:**

2

**Strengths And Weaknesses:**

Pros:
1. Clarity and writing quality: The paper is well written with clear explanations and a logical flow of information.
2. Theoretical rigor and practicality: The algorithm has a comprehensive theoretical analysis, but is also easy to implement engineeringly. The paper also provides an intuitive toy case to explain the motivation behind the algorithm.
3. The paper contains a wide spectrum of motion experiments, and contains major variants of text to motion algorithms such as motion diffusion models and MMM.

Cons:
1. The proposed algorithm is a general DPO improvement, but the experiments are only conducted in text to motion generation. It raises concerns on how generalized the solution is and how well the claim is supported. That is, if the claim is true, why don’t we apply it to other domains?

2. The visual quality of the presented motion results is not entirely convincing. Specifically, the baseline performance without SoPo appears significantly inferior to what would be expected from a well-trained Motion Diffusion Model (as shown in the supplementary).
This observation suggests that the baseline model may be either undertrained or not optimally configured, potentially impacting the comparative efficacy demonstrated by SoPo.

3. While the paper presents improved numerical results across metrics such as R-precision and FID, it is not clear that these quantitative gains consistently translate into a better alignment with human preferences.
This is particularly questionable since the “preferred samples” are directly chosen from the target dataset. With additional training it’s expected that those metrics will improve (unless I misunderstand how training works and the proposed algorithm actually trains from scratch?)

To address this critical ambiguity and to validate the practical efficacy of the approach, we need a comprehensive human evaluation study to support the claim of the paper.

4. A more fundamental concern lies in the core assumption that 'preferred data' is exclusively offline and directly equates to the target dataset.
The application of text to motion authoring is an interactive process where preferred motions are chosen among generations or even directly authored by artists. By presuming preferred data is always pre-generated, the model's ultimate capacity to adapt to dynamic user preferences and its applicability in interactive creative workflows may be severely limited.
The fact that the model is always trying to give high scores to existing samples in the dataset also raises very big concerns on how well or if the model can generalize to novel text prompts from the users.

---

> ### Author Rebuttal · Authors · 2025-07-31
>
> Thank you for your encouraging feedback. **We’ve addressed your concerns on overfitting, generalization, and preference diversity with added experiments and clarifications.** We also included an illustrative example to highlight our method’s effectiveness. We appreciate your insights and look forward to further discussion.
>
> ---
>
> ## [Q1/Q2] Typo
>
> Thank you for pointing out the typo. We will correct it in the next version.
>
> ---
>
> ## [Q3/W3] Is the reward function just the TMR similarity score? What happens if TMR is overfitting? I think it will be necessary to test novel or out of distribution text and motion samples requested by the users.
>
> Thank you for your insightful question about **the reward function and the potential overfitting of the TMR model**, particularly its impact on **out-of-distribution text and motion generation**.
>
> To address your concerns, we conducted an OOD generation experiment.
>
> ### [Q3-1] Is the reward function just the TMR similarity score?
>
> Yes, our reward function currently relies on the TMR similarity score. However, designing a reward function may be not the core focus of our work. **Our method is fully extensible to multi-reward scenarios, allowing integration of additional reward signals as needed.**
>
> ### [Q3-2] What happens if TMR is overfitting?
>
> This is an important concern. To clarify, we use a **pretrained TMR model** rather than training one ourselves. As is standard in prior work [2,3,4,5], reward model training and policy fine-tuning are treated as independent stages. **It is thus critical that the reward model is trained properly to avoid overfitting [2,3].**
>
> In fact, our method is **explicitly designed to mitigate this risk**. **If TMR exhibits some degree of overfitting, our generative model is not necessarily affected.**  As shown in Lines 88–89, we introduce a KL divergence constraint to regularize training:
> $$
> \max\_{\pi\_\theta} \mathbb{E}\_{c\sim\mathcal{D}, x \sim \pi_\theta(\cdot|c)} \left[ r(x,c) - \beta D_{\text{KL}} \left( \pi_\theta(x|c)\,\|\,\pi_{\text{ref}}(x|c) \right) \right].
> $$
> DPO-based methods inherently prevent overfitting by encouraging proximity to the reference model, acting as implicit regularization.
>
> ### [Q3-3] Out-of-distribution Evaluation
>
> Thanks for the suggestion. We clarify that for each benchmark, the preference data, pretrained generative model, and reward model are all trained on the training set of benchmark. **All evaluations are done on a held-out test set, which is not seen during training and thus nearly out-of-distribution.**
>
> **To further assess generalization, we conduct an OOD test by sampling 100 text prompts from the KIT-ML dataset and generating motions using a model fine-tuned on HumanML3D.** As a baseline, we also evaluate the original HumanML3D pretrained model. The results are:
>
> |Model|MotionCritic↑|
> |-|-|
> |Pretrained model|0.68|
> |+SoPo|1.36|
>
> Note: Since KIT and HumanML3D differ in their motion representation formats, ground-truth motions are incompatibility, making metrics like FID infeasible. Instead, we report MotionCritic scores for evaluation.
>
> ---
>
> ## [Q4] Sensitivity to engineering hyperparameters
>
> Thank you for the question. Our method uses a **unified setting** across experiments: $C = 2$, $\beta = 1$. The hyperparameters fall into two groups:
> 1. **From SoPo**: filtering threshold $\tau$, candidate number $K$, weight $C$
> 2. **From DPO**: temperature $\beta$
>
> For SoPo-specific hyperparameters, **Tab.4**(in paper) shows they have minor influence. Below, we report results on MLD* to analyze the sensitivity to $\beta$ and $C$:
>
> |Method|R@1↑|R@2↑|R@3↑|MMDist↓|FID↓|
> |-|-|-|-|-|-|
> |SoPo(C=1,$\beta$=0.25)|0.523|0.717|0.823|2.941|0.176|
> |SoPo(C=1,$\beta$=0.5)|0.524|0.718|0.824|2.940|0.175|
> |SoPo(C=1,$\beta$=1)|0.525|0.719|0.825|2.939|0.174|
> |SoPo(C=2,$\beta$=0.25)|0.527|0.721|0.826|2.938|0.173|
> |SoPo(C=2,$\beta$=0.5)|0.528|0.722|0.827|2.937|0.172|
> |SoPo(C=2,$\beta$=1)|0.528|0.722|0.827|2.939|0.174|
> |SoPo(C=3,$\beta$=0.5)|**0.532**|**0.726**|**0.831**|2.935|0.170|
> |SoPo(C=3,$\beta$=1)|0.530|0.724|0.829|**2.934**|**0.169**|
> |SoPo(C=3,$\beta$=2)|0.529|0.723|0.828|2.936|0.171|
>
> The results show our method is robust to changes in $\beta$ and $C$.
>
> ---
>
> ## [W1] Concern on Generalizability
>
> Thank you for raising this point — it clarifies SoPo’s generalizability.
>
> To test beyond text-to-motion, we follow [1] and apply SoPo to image generation. We use Pick-a-Pic preferences as offline data, PickScore as reward, and SD 1.5 as the policy. Evaluation on PartiPrompts uses PickScore, HPSv2.1, ImReward, Aesthetic, and CLIP. Results:
>
> |Model|PickScore↑|HPSv2.1↑|ImReward↑|Aesthetic↑|CLIP↑|
> |-|-|-|-|-|-|
> |SD1.5|21.40|24.97|0.121|5.355|33.11|
> |SFT|21.77|28.17|0.702|5.592|34.07|
> |DPO|21.62|25.79|0.386|5.442|33.54|
> |KTO|21.77|28.05|0.677|5.547|34.11|
> |SmPO [1] (ICML25)|22.00|**28.81**|0.911|5.636|34.72|
> |SoPo(Ours)|**22.23**|28.70|**0.958**|**5.749**|**34.80**|
>
> We will include these results in the appendix of the final version.
>
> [1] Smoothed Preference Optimization via ReNoise Inversion for Aligning Diffusion Models with Varied Human Preferences
>
> ---
>
> ## [W2] Concern on baseline quality (whether either undertrained or not optimally configured) and the reason of SoPo’s improvement
>
> ### [W2-1] May the selected diffusion model be either undertrained or not optimally configured
>
> Thank you for the question. We acknowledge that our qualitative results use classic MDM and MLD models (ICLR 23 & CVPR 23). **However, all pre-trained models are official open-source implementations with pretrained weights.** We selected these two classic because prior work (MoDiPO) only reported results on these models without releasing code. **Importantly, Tab. 2 and 3 use SoTA MLD\* and MoMask as base models, showing consistent SoPo improvements on both classic and SoTA models.**
>
> ### [W2-2] Visual improvement and quantitative analysis
>
> This is a valuable question. **To further evaluate visual improvement, we now quantify the representative case in our paper, spatial alignment.**
>
> Among all test prompts, 783 (35.7%) contain spatial cues (e.g., “left”, “right”). This raises an important question: **Why does SoPo help the motion diffusion model better understand left-right spatial information?**
>
> #### [W2-2.1] Reward model’s spatial understanding
>
> Given a prompt $t$ and motion $x$, we compute reward $r(x,t)$. We then swap “left” and “right” to get $t'$ and compare $r(x,t)$ vs. $r(x,t')$. If $r(x,t) > r(x,t')$, it’s considered a success.
>
> |Model|Count|Success|Failure|Accuracy|
> |-|-|-|-|-|
> |Random|-|-|-|0.500|
> |Reward model|783|628|155|0.802|
>
> #### [W2-2.2] Diffusion model generation
>
> We sampled 100 prompts with spatial terms. For each, 5 motions were generated (500 total). Human evaluation counted how many reflected correct spatial meaning.
>
> |Model|Count|Success|Failure|Accuracy|
> |-|-|-|-|-|
> |Diffusion model|500|316|184|0.632|
> |+ SoPo|500|382|118|0.764|
>
> Results suggest that the reward model captures spatial distinctions more effectively than the diffusion model. **SoPo leverages this supervision signal from the reward model to improve spatial understanding of generative model.**
>
> ---
>
> ## [W3] Concern: It's unclear whether improvements in metrics like R-precision and FID truly reflect better human preference alignment, especially since the "preferred samples" are drawn from the training set. Reviewer also questions whether improvements come simply from more training.
>
> ### [W3-1] Relationship Between Metrics and Human Preference
>
> We appreciate the reviewer’s concern. In our setting, **human preference is defined as a combination of text-motion alignment and motion quality**. R-precision and FID respectively quantify these aspects (in Line 242-243), and are therefore aligned with this definition.
>
> ### [W3-2] Clarification on Training Data
>
> We appreciate the concern regarding potential data leakage. **To clarify, the “preferred samples” used as training data correspond strictly to the benchmark’s original training set.** Both the pretrained generative model and the reward model are trained solely on this set **without access to any additional data**.
>
> Thus, the reported improvements reflect better alignment within the benchmark distribution rather than from exposure to extra or out-of-distribution data. **Additionally, the reason for performance improvement is clarified in [W2].**
>
>
> ## [W4] Concerns on Preference Diversity and Generalization
>
> Thank you for raising these points. The concerns are:
> 1. **Real-world preferences are diverse and dynamic**, but we use fixed preferred data and static reward models, limiting adaptability.
> 2. **Offline-learned preferences may not generalize to out-of-distribution prompts.**
>
> These are common challenges in offline post-training/RLHF and DPO approaches, which, while not the main focus of our work, are important for the broader community:
> - We focus on preferences like **motion realism and text-motion alignment**, which are relatively objective. In the image domain, common criteria include alignment and safety. **Handling subjective, diverse preferences with limited data remains a key open problem.**
> - Learning from fixed datasets is standard in offline post-training. **Prior work (e.g., SD3 [5]) shows strong generalization despite being trained on static data**.  In my opinion, this may be since post-training reshapes the pretrained distribution without compromising its continuity or semantic structure.
>
> We hope this respond addresses your concerns well.
>
> [1] TMR: Text-to-Motion Retrieval Using Contrastive 3D Human Motion Synthesis
>
> [2] ImageReward: Learning and Evaluating Human Preferences for Text-to-Image Generation
>
> [3] Pick-a-Pic: An Open Dataset of User Preferences for Text-to-Image Generation
>
> [4] Diffusion Model Alignment Using Direct Preference Optimization
>
> [5] Scaling Rectified Flow Transformers for High-Resolution Image Synthesis

---

> > ### Comment · Reviewer_7xd7 · 2025-08-08
> > **thank you for your rebuttal**
> >
> > I would like to thank the authors for this very extensive rebuttal.
> >
> >
> > Q1, Q2, Q3: This clears my confusion and improves the quality & clarity of the paper.
> >
> > Q4: The engineering hyper-parameter study here is very informative and provides a good reference for the robustness of the algorithm.
> >
> > W1: I appreciate the authors on performing the exps on the image generation domain. This clears my concerns about generalization and show that the proposed methods could potentially help the image generation domain, broadening the impact of this work.
> >
> > W2: The authors address my concerns on the baseline performance.

---

> > > ### Comment · Reviewer_7xd7 · 2025-08-08
> > > **thank you for your rebuttal (2)**
> > >
> > > 1. For W3, I am still not entirely convinced that R-precision and FID truly reflects human preference.
> > >
> > > The human preference can only be evaluated convincingly with human in the loop.
> > >
> > > R-precision and FID are just proxy metrics and there's no guarantee.
> > >
> > > A user study will be the mostly convincing method to show that it actually align with human preference.
> > >
> > > 2. I think the issues of W3-1 and W3-2 and W4 are somewhat related.
> > >
> > > The authors didn't have an actual "preference data" bandwidth. There's no real human preference, and there's no real human evaluation.
> > >
> > > Overall this is my biggest concern. But the quality of the work outweigh this concern and I truly appreciate the detailed high quality rebuttal from the authors.

---

> ### Comment · Reviewer_7xd7 · 2025-08-08
> **thank you for your rebuttal (3)**
>
> Based on my previous comments, I tend to recommend for acceptance for this paper.
>
> Some further suggestions for the final draft that I believe will contribute to the quality of the paper:
>
> 1) adding human evaluation on the results (actually asking people to rate the quality not using FID / R-precision)
> 2) discuss future work on collecting real human preference data to train the algorithm instead of using groundtruth data.

---

> ### Author Response · Authors · 2025-08-09
>
> Thank you for your positive feedback. We sincerely appreciate the time and effort you put into your thorough review, which has greatly improved our manuscript. Thank you again for your support.

---

### Official Review · Reviewer_GBbd · 2025-06-29

**Clarity:** 3
**Significance:** 2
**Originality:** 2
**Rating:** 3
**Confidence:** 4

**Summary:**

This paper proposes a preference-learning-based generation method for text-to-motion generation. This paper attempt to utilize DPO in motion synthesis via text-to-motion model fine-tuning, and propose a Semi-online Preference Optimization. The training sample consisting of unpreferred motion from online distribution and preferred motion in offline datasets. The experimental results indicate better R-precision for motion retrieval.

**Questions:**

1. In the ablation studies, comparisons are made for different K and \tau, it will be more important to see different sampling or weighting strategies.
2. SoPo relies on a reward model to assess motion preferences. However, the current simple reward model struggles to evaluate human motion in a fine-grained manner. A more effective reward model should take into account richer physical and semantic information.

**Ethical Concerns:**

["NO or VERY MINOR ethics concerns only"]

**Final Justification:**

The rebuttal provides the missing results. However, there are still concerns on the contribution of integration of off-line and on-line preference optimization built upon existing Preference Optimization. It is not enough. Meanwhile, the accuracy of the reward model is not convincing. So, I will keep my original rating.

**Limitations:**

There is no obvious negative societal impact of their work.

**Paper Formatting Concerns:**

There is no paper formatting concerns.

**Quality:**

2

**Strengths And Weaknesses:**

Strengths:
1. By combining high-quality offline "preferred samples" with dynamically generated online "non-preferred samples", the method improves the preference gap and prevents overfitting.
2. The use of a threshold strategy can make the truly "non-preferred" samples be selected.
3. The method is evaluated on different datasets and models, consistently outperforming baseline methods.
Weakness:
1 Preference Optimization for Motion Synthesis in MoDiPO [1] and combination of on-line and off-line preference optimization has been proposed in [2]. So, the contribution should be more clearly stated.
2. According to Table 2, the R-precision increases, but the MM Dist and Diversity degrade, the proposed method may scarify some metrics for others.
3. Lots of results are missing in Table 1 which make it seems to be unfinished.

[1]Pappa, M., Collorone, L., Ficarra, G., Spinelli, I. and Galasso, F., 2024. MoDiPO: text-to-motion alignment via AI-feedback-driven Direct Preference Optimization. CoRR. 2024
[2] Cen, S., Mei, J., Goshvadi, K., Dai, H., Yang, T., Yang, S., Schuurmans, D., Chi, Y. and Dai, B., Value-Incentivized Preference Optimization: A Unified Approach to Online and Offline RLHF. ICLR 2025.

---

> ### Author Rebuttal · Authors · 2025-07-31
>
> Thank you for your valuable feedback and recognition of our method. Your comments on related work, ablations, and physical-semantic settings were very helpful. We reviewed the suggested references, clarified key differences, extended ablation studies, and added a new experiment to evaluate our method in physically and semantically grounded scenarios.
>
> ---
>
> ## [Q1/W1] Ablation studies about different sampling or weighting strategies.
>
> Thank you for suggesting more comprehensive ablation studies on sampling and weighting strategies.
>
> SoPo builds upon on-/offline DPO, combining their strengths into a semi-online approach, termed Combination DPO, as noted by [GBbd]. However, this naive combination has limitations (in Sec. 4). To address these, we introduced Offline Sampling for Preferred Motions (Sec. 4.3) and Online Generation for Unpreferred Motions (Sec. 4.4). For offline dataset, we generate the preference dataset following MoDiPO. Results are shown below:
>
> |Method|Top1↑|Top2↑|Top3↑|MM.Dist.↓|Div→|FID↓|
> |-|-|-|-|-|-|-|
> |Baseline (MLD*)|0.504|0.698|0.796|3.052|9.634|0.450|
> |OfflineDPO|0.498|0.691|0.790|3.081|9.618|0.469|
> |OnlineDPO|0.514|0.708|0.809|3.012|9.611|0.414|
> |CombinationDPO|0.517|0.712|0.811|2.988|9.606|0.339|
> |+Sec.4.3|0.520|0.715|0.814|2.973|9.602|0.318|
> |+Sec.4.4|0.525|0.719|0.820|2.951|9.591|0.243|
> |**SoPo (Ours)**|**0.528**|**0.722**|**0.827**|**2.939**|**9.584**|**0.174**|
>
> The results highlights the effectiveness of each SoPo component.
>
> ---
>
> ## [Q2/W2] Limitation of the Current Reward Model in Capturing Fine-Grained Human Motion Preferences and the Need for Richer Physical and Semantic Awareness
>
> Thank you for raising this important point. Recent research in motion generation has indeed highlighted the need for **physical plausibility and semantic awareness**. We believe your concern touches on two aspects:
> 1. Whether our simple reward model (text-motion alignment) can effectively **capture semantic information** in generated motions.
> 2. Whether SoPo is extensible to **physically simulated reward environments**.
>
> Our work primarily focuses on how to leverage the reward model to improve motion generation quality, such as realism and text-motion alignment. Designing a more fine-grained reward model remains an important and open issue for future work. For these question, we think:
> 1. Our reward model can capture fine-grained semantic imformation and improving the generative model’s spatial and physical awareness.
> 2. Compared to online/offline DPO, SoPo has more potential to integrate with physically grounded environments.
>
> ### [Q2-1] Experiment: The spatial semantic perception ability of Reward Model/SoPo
>
> Our key idea for improving text-motion alignment is to use feedback from a discriminator (reward model) to guide the diffusion model. Since discriminators better capture spatial semantics than generators, this enhances the generator’s spatial perception.
>
> We focus on three questions:
> 1. Can the reward model distinguish spatial semantic misalignment?
> 2. Can the diffusion model generate spatially semantic aligned motions?
> 3. Can SoPo improve the generative model’s spatial semantic perception?
>
>
> **[Q2-1.1] Can the reward model detect spatial semantic misalignment?**
>
> We analyzed the HumanML3D test set with 2192 prompts and found that 783 (35.72%) contain spatial expressions such as “left” or “right”, showing this issue is common.
>
> For each motion-text pair $(x, t)$, we calculate a reward score $r(x, t)$. Then, we flip the spatial keyword in the text prompt (e.g., “left” to “right”) to get $t’$, and compute $r(x, t’)$. If $r(x, t) > r(x, t’)$, we count this as a successful detection. The results are as the 2-nd row of the table.
>
> **[Q2-1.2] Can the diffusion model generate spatially semantic aligned motions?**
>
> We randomly sampled 100 of the 783 spatial prompts and generated five motions per prompt (500 total) using the pre-trained diffusion model. Human evaluators then judged whether each motion aligned with its prompt. The results are shown in the 3-rd row of the table.
>
> **[Q2-1.3] Can SoPo improve the generative model’s spatial semantic understanding?**
>
> As shown in the 4-th row of table, fine-tuning the diffusion model using SoPo significantly improved its spatial perception.
>
> |Model|Samples|Correct|Wrong|Accuracy|
> |-|-|-|-|-|
> |Random|-|-|-|0.500|
> |Reward Model|783|628|155|0.802|
> |Diffusion Model|500|316|184|0.632|
> |**+ SoPo**|500|382|118|0.764|
>
> This highlights that  **SoPo effectively aligns the spatial semantics of generative model with the discriminator.**
>
> ### [Q2-2] Discussion: SoPo for Richer Physically Simulated Optimization
>
> Existing works in physical simulation — such as TokenHSI [1], PhysHOI [2], and SkillMimic [3] — typically rely on imitation learning from physically simulated motion data.
>
> For preference optimization setting, the motions used for imitation learning can be viewed as positive preference data, while failures during environment interaction serve as negative preference data. **This naturally aligns with SoPo’s semi-online optimization: combining offline motions with online failures for joint preference learning. It enables the model to both imitate high-quality motions and avoid undesirable ones during interaction.**
>
> Compared to pure online or offline DPO, SoPo is better suited for integration with simulated environments. Recent Embodied AI work applying Online DPO to tasks like grasping [4] further supports the extensibility of DPO-based frameworks.
>
> Thank you for the insightful suggestion. We agree that stronger reward models with physical/semantic priors are valuable.  **However, our main contribution lies in proposing a novel preference alignment framework (SoPo),** rather than in designing new reward models per se.
>
> [1] *Tokenhsi: Unified synthesis of physical human-scene interactions through task tokenization.*
>
> [2] *Physhoi: Physics-based imitation of dynamic human-object interaction.*
>
> [3] *Skillmimic: Learning basketball interaction skills from demonstrations.*
>
> [4] *Evolvinggrasp: Evolutionary grasp generation via efficient preference alignment.*
>
> ---
>
> ## [W1] Related work Discussion
>
> ### [W1-1] Preference Optimization for Motion Synthesis
>
> Thank you very much for your valuable comment—this is indeed a critical and insightful point that we are eager to clarify.
>
> In our paper (Lines 41–60), **we explicitly acknowledge and build upon the existing work of MoDiPO** [1], which first applies preference optimization to motion synthesis. **Rather than claiming to be the first to explore this direction, we carefully analyze why MoDiPO fails to deliver promising results** (Lines 41–50) and **propose SoPo as a new method to better unlock the potential of preference optimization** for this task (Lines 51–60).
>
> ### [W1-2] Differences from VPO [2]
>
> Thank you again for pointing out [2]—it is an excellent work. Below, we elaborate on the key differences between [2] and our method to more clearly state our contributions.
>
> In [2], the authors formulate a unified preference optimization objective that incorporates both online and offline signals through reward regularization. **However, they design separate algorithms for the online and offline settings (see Algorithms 1 & 2) and report results for each setting independently, rather than integrating both within a unified algorithm.** In contrast, our work analyzes the theoretical strengths and limitations of online and offline preference optimization and proposes SoPo to combine their respective advantages. For experiment, **we jointly use both online and offline preference signals in a unified training pipeline, rather than treating them separately.**
>
> Additionally, concurrent work in the LLM domain [3] (arXiv in 2025.06) tries to explore semi-online preference optimization, supporting the novelty and relevance of our approach.
>
> Thank you again for your thoughtful feedback. We will revise our related work section. **Therefore, we believe our proposal of SoPo, which integrates online-DPO and offline-DPO into a unified semi-online optimization framework, represents a justified and novel contribution, as stated in Lines 41–60.**
>
> [1] *MoDiPO: text-to-motion alignment via AI-feedback-driven Direct Preference Optimization.*
>
> [2] *Value-Incentivized Preference Optimization: A Unified Approach to Online and Offline RLHF.*
>
> [3] *Bridging Offline and Online Reinforcement Learning for LLMs.*
>
> ---
>
> ## [W2 (4-th point in "Strengths And Weaknesses")] According to Table 2, the R-precision increases, but the MM Dist and Diversity degrade, the proposed method may scarify some metrics for others.
>
> We sincerely thank you for highlighting the potential trade-offs in Tab. 2. **To clarify, MMDist is better when lower, and Diversity is better when closer to ground truth.**
>
> In Tab. 2, SoPo consistently improves R-precision. For MM Dist and Diversity: MLD benefits in both metrics, and MoMask improves in Diversity while maintaining comparable MM Dist (with only a slight drop). Only in one case does MM Dist degrade slightly, but it remains competitive.
>
> **Hence, while SoPo improves R-precision, it does not necessarily sacrifice MM Dist or Diversity.** We will make this clearer in the revised version to avoid misunderstanding.
>
>
> ## [W3 (5-th point in "Strengths And Weaknesses")]  Missing Results in Table 1
>
> Thank you for pointing out the missing entries in Table 1. These omissions were due to MoDiPO not reporting certain metrics. (Note: the MLD and MDM results are from MoDiPO, which differ from those in the original papers.) We have reproduced the missing values to the best of our ability, as shown below.
>
> ||Top 1|Top 2|
> |-|:-:|:-:|
> |MLD|0.469|0.660|
> |+MoDiPO-T|0.471|0.662|
> |+MoDiPO-G|0.468|0.659|
> |+MoDiPO-O|0.416|0.581|
> |+SoPo|0.476|0.671|
> |MDM|0.418|0.604|
> |+MoDiPO-T|0.421|0.607|
> |+MoDiPO-G|0.416|0.605|
>
> We will update the manuscript to include these values.

---

> > ### Comment · Reviewer_GBbd · 2025-08-05
> >
> > Thanks for your response. In Q2/W2, the reward model is better than a random justification. However, the accuracy is far from enough. This may give a lower upper bound for your method. Meanwhile, the contribution of integration of off-line and on-line preference optimization built upon existing Preference Optimization is not enough. The W3 part give the missing results. The utilization of SoPo in simulated environments is not evidenced.

---

> ### Author Response · Authors · 2025-08-05
>
> Thank you for raising this question. We would like to provide further clarification regarding your concern and any possible misunderstanding.
>
> ---
>
> ### 1. Reward model may limit the upper bound of performance
>
> This is indeed a very central and important point.
>
> Theoretically, it is true that the reward model could potentially limit the upper bound of performance. In general, **judging whether a motion is well-aligned is an easier task than generating such an aligned motion.** That is, given the same amount of data, a discriminative model (i.e., the reward model) is typically capable of achieving better performance than a generative model.
>
> SoPo, at its core, serves as an alignment algorithm. **Its goal is to align the generative model’s capabilities with the stronger discriminative power of the reward model, rather than to completely solve the semantic misalignment problem.** In fact, thoroughly solving semantic misalignment would indeed require training with large-scale data. However, **under a fixed data budget,** SoPo is able to improve generation performance by leveraging a stronger reward model.
>
> We thank the reviewer for pointing out this important concern. **Completely resolving semantic misalignment remains a challenging task—especially in the motion domain unless the dataset is large enough.** SoPo offers a practical and effective way to mitigate this issue and further push the performance boundary.
>
> ---
>
> ### 2. Meanwhile, the contribution of integration of off-line and on-line preference optimization built upon existing Preference Optimization is not enough.
>
> Thank you for pointing out this issue. As stated in the contributions section(Line 41-60), our work does not lie in simply combining the two strategies. Rather, **we provide a theoretical explanation of the respective limitations of offline and online preference optimization, and why their integration can effectively address these issues.** Our approach also differs significantly from the related work mentioned (see “[W1] Related Work Discussion”). **This is intuitively illustrated in Fig. 3 of the paper.**
>
> ---
>
> ### 3. The W3 part give the missing results.
>
> We sincerely thank the reviewer for the positive recognition.
>
> ---
>
> ### 4. The utilization of SoPo in simulated environments is not evidenced.
>
> We sincerely thank the reviewer for raising this important point.
>
> SoPo is a DPO-based (or RL-based) method. To the best of our knowledge, there is indeed no direct literature demonstrating the use of DPO specifically in physically simulated environments within the motion domain.
>
> However, **prior work [1,2,3] has shown that RL-based approaches have been successfully applied in such environments**. Additionally, some **DPO-based methods have been explored in simulated robotic tasks** (e.g., grasp generation [4]), and even extended to **real-world scenarios**.
>
> From a theoretical perspective, although empirical evidence in the filed human motion is still limited, the DPO-based nature of SoPo makes its application to physically simulated environments a promising direction (see “[Q2-2] Discussion: SoPo for Richer Physically Simulated Optimization”).
>
> ---
>
> [1] *Tokenhsi: Unified synthesis of physical human-scene interactions through task tokenization.*
>
> [2] *Physhoi: Physics-based imitation of dynamic human-object interaction.*
>
> [3] *Skillmimic: Learning basketball interaction skills from demonstrations.*
>
> [4] *Evolvinggrasp: Evolutionary grasp generation via efficient preference alignment.*
>
>
> ---
>
> Lastly, we are truly grateful to the reviewer for raising this valuable concern. We wonder whether our response has adequately addressed your question. Even with the limited time remaining, we are more than willing to engage in further discussions should you have any remaining concerns, and would greatly appreciate it if you could consider revisiting your evaluation of our paper.

---

### Official Review · Reviewer_tJvD · 2025-06-30

**Clarity:** 3
**Significance:** 2
**Originality:** 3
**Rating:** 4
**Confidence:** 4

**Summary:**

This work introduces Semi-online Preference Optimization to train text-to-motion models that better align with textual inputs. The key idea is to use unpreferred motion from online distribution and preferred motion in offline datasets for DPO training. The authors provide theoretical analysis for this choice and show in experiments that it boosts performance on motion generation models like MLD and MDM.

**Questions:**

Following weaknesses part, I have several questions.

1. Could the authors provide results of applying the proposed method to other domains and include them in the appendix?

2. Could the authors provide clearer comparisons with online DPO, offline DPO and their naive combination?

3. How well does the preference optimization solve spatial constraint misalignment for given texts? Could the authors design some experiments and show quantitative results?

**Ethical Concerns:**

["NO or VERY MINOR ethics concerns only"]

**Final Justification:**

The rebuttal addresses my concerns. I appreciate the authors' effort to validate the generalizability beyond text-to-motion tasks and the capability of solving spatial misalignment. It would help if the authors include these points in the final version. I keep my score as 4: borderline accept based on the overall contribution of this paper.

**Limitations:**

Limitations are discussed in the paper.

**Paper Formatting Concerns:**

No.

**Quality:**

3

**Strengths And Weaknesses:**

Strengths:

1. The research scope of this work i.e. preference alignment for text-to-motion models is important yet under-explored in the community.

1. Figuring out the respective limitation of online and offline DPO and combining them is technically sound and valuable. The core insight of the proposed method is concise yet interesting enough.

2. The paper is well-written and of high quality. There are nice figures (especially Fig. 2 and Fig. 3) that make this work intuitive to understand.

Weaknesses:

My concerns are mainly about the evaluation.

1. Although applied to the text-to-motion task, the proposed method is essentially a general loss that is not dependent on specific datasets, diffusion models and retrieval models. This work lacks evaluation on generation tasks in other domains to prove its generalizability, such as for image generation.

2. In order to better evaluate the merits of such combination, the paper should more clearly present the baseline results of only using online DPO, offline DPO and their naive combination. Currently, the ablation study in Table 4 is a bit vague and needs better clarity. Also from Table 4, it seems that the improvement of each design is not very significant.

3. It is not clear what type of problem in motion generation is addressed by preference optimization. From Fig. 1, I assume that spatial constraints in text inputs are better aligned, such as left/right, back, etc. Consequently, besides qualitative examples, some quantitative statistics reflecting ratio of successful/failure samples can be added to show how well the preference optimization solves these issues.

4. As for the related work, more discussion is required for the recent advances in preference alignment for generation tasks.

---

> ### Author Rebuttal · Authors · 2025-07-31
>
> Thank you very much for your recognition of the novelty, writing quality, and significance of our work. You also raised several insightful points — including the **generality of SoPo to other domains**, **more complete ablations**, and **how SoPo addresses spatial perception through semantic feedback**. These are all valuable suggestions. In response, we have conducted additional in-depth experiments, including a new analysis inspired by your comments to **illustrate how preference alignment addresses spatial consistency in motion generation**. We sincerely appreciate your feedback — these experiments significantly improve the completeness of our work.
>
> ---
>
> ## [Q1/W1] Could the authors provide results of applying the proposed method to other domains and include them in the appendix?
>
> Thank you for the suggestion — it helps clarify SoPo’s generalizability.
>
> We demonstrate SoPo's transferability to the **text-to-image generation domain**, following the setup in [1]. We use Pick-a-Pic’s preferred image as the offline dataset, PickScore as the reward model, and Stable Diffusion 1.5 as the policy model. Evaluation is conducted on the Parti-Prompts (1632 prompts) using PickScore, HPSv2.1, ImReward, Aesthetic, and CLIP.
>
> |Model|PickScore↑|HPSv2.1↑|ImReward↑|Aesthetic↑|CLIP↑|
> |-|-|-|-|-|-|
> |SD1.5|21.40|24.97|0.121|5.355|33.11|
> |SFT|21.77|28.17|0.702|5.592|34.07|
> |DPO|21.62|25.79|0.386|5.442|33.54|
> |KTO|21.77|28.05|0.677|5.547|34.11|
> |SmPO (ICML'25)|22.00|**28.81**|0.911|5.636|34.72|
> |**SoPo (Ours)**|**22.23**|28.70|**0.958**|**5.749**|**34.80**|
>
> These results show that **SoPo consistently improves multiple preference metrics across domains**. We will include the full results in the appendix.
>
> [1] Yunhong, Lu, et al. "*Smoothed Preference Optimization via ReNoise Inversion for Aligning Diffusion Models with Varied Human Preferences.*" arXiv preprint arXiv:2506.02698 (2025).
>
> ---
>
> ## [Q2/W2] Could the authors provide clearer comparisons with online DPO, offline DPO and their naive combination?
>
> We appreciate your valuable suggestion to provide clearer comparisons.
>
> In Tab.1 (in paper), we treat MoDiPO-G and MoDiPO-T as approximate online/offline DPO baselines, since MoDiPO only introduces minor technical changes to the standard DPO framework.
>
> **For a clearer comparisons, we conducted new experiments comparing online DPO, offline DPO, their naive combination, and combination with our proposed strategies** (Offline Sampling for Preferred Motions, Sec. 4.3, and Online Generation for Unpreferred Motions, Sec. 4.4). Specifically:
> - **Offline DPO**: We use MLD\* as the base model, and follow MoDiPO to create an offline preference dataset.
> - **Online DPO**: We use TMR as the reward model, consistent with SoPo.
> - **Combination DPO**: We use offline dataset for preferred motion, and online generate unpreferred motion.
>
> Results are shown below:
>
> |Method|Top1↑|Top2↑|Top3↑|MM.Dist.↓|Div→|FID↓|
> |-|-|-|-|-|-|-|
> |Baseline (MLD*)|0.504|0.698|0.796|3.052|9.634|0.450|
> |OfflineDPO|0.498|0.691|0.790|3.081|9.618|0.469|
> |OnlineDPO|0.514|0.708|0.809|3.012|9.611|0.414|
> |CombinationDPO|0.517|0.712|0.811|2.988|9.606|0.339|
> |+Sec.4.3|0.520|0.715|0.814|2.973|9.602|0.318|
> |+Sec.4.4|0.525|0.719|0.820|2.951|9.591|0.243|
> |**SoPo (Ours)**|**0.528**|**0.722**|**0.827**|**2.939**|**9.584**|**0.174**|
>
> These results highlight that **SoPo's hybrid semi-online design provides more effective and data-efficient alignment**, avoiding the limitations of both pure online and offline DPO.
>
> ---
>
> ## [Q3/W3] How well does the preference optimization solve spatial constraint misalignment for given texts?
>
> This is a critical and valuable question — spatial misalignment is a common failure case in text-to-motion generation.
>
> The core insight to solve this issue is that **reward models (discriminators)** are better at judging spatial semantics than **generative models** (generator), and SoPo leverages reward feedback to improve alignment.
>
> We divide this issue into three sub-issues:
> 1. Can the reward model distinguish left/right correctly?
> 2. Can the diffusion model generate motions consistent with left/right prompts?
> 3. Can SoPo improve generation via preference optimization?
>
> ### [Q3-1] Reward Model Discrimination Ability of Spatial Misalignment
>
> From the HumanML3D test set (2,192 prompts), 783 prompts (35.72%) contain spatial information (e.g., “left” or “right”), highlighting the prevalence of this issue. For a text-motion pair $(x, t)$, we computed the reward score $r(x, t)$. We then created a misaligned text $t'$ by swapping “left” with “right” and computed $r(x, t')$. The reward model is considered successful if $r(x, t) > r(x, t')$. Results are shown below:
>
> |Model|Samples|Correct|Wrong|Accuracy|
> |-|-|-|-|-|
> |Random|-|-|-|0.500|
> |Reward Model|783|628|155|0.802|
>
> ### [Q3-2] Diffusion Model Generative Ability of Spatial Alignment Generation
> We randomly selected 100 spatial prompts from the 783 and generated 5 motions per prompt (500 total). Human annotators judged whether motions matched the spatial constraints:
>
> |Model|Samples|Correct|Wrong|Accuracy|
> |-|-|-|-|-|
> |Random|-|-|-|0.500|
> |Reward Model|783|628|155|0.802|
> |Diffusion Model|500|316|184|0.632|
>
> ### [Q3-3] The Alignment Quality of SoPo
>
> After fine-tuning the diffusion model using SoPo, we evaluated the same 500 prompts:
>
> |Model|Samples|Correct|Wrong|Accuracy|
> |-|-|-|-|-|
> |Random|-|-|-|0.500|
> |Reward Model|783|628|155|0.802|
> |Diffusion Model|500|316|184|0.632|
> |**+ SoPo**|500|382|118|0.764|
>
> These results demonstrate that:
> 1. The reward model is capable of detecting spatial misalignments;
> 2. The original diffusion model struggles with spatial understanding;
> 3. **SoPo effectively enhances spatial alignment in generated motions.**
>
> Hence, SoPo offers a practical solution to address spatial misalignment by integrating spatial semantic information from the text-motion-aligned reward model.
>
> Sincerely, thank your insightful suggestion!
>
> ---
>
> ## [W4] As for the related work, more discussion is required for the recent advances in preference alignment for generation tasks.
>
> We sincerely thank the reviewer for this suggestion. We will enhance the related work section to include recent advances in preference alignment, including **RL-based**, **differentiable-based**, **text-time alignment**, and **recent DPO-based** advancements, as well as applications in domains like **LLMs**, **video**, **3D**, and **audio**.

---

> > ### Comment · Reviewer_tJvD · 2025-08-05
> >
> > Thanks for the detailed response. The rebuttal addresses my concerns. I  appreciate the authors' effort to validate the generalizability beyond text-to-motion tasks and the capability of solving spatial misalignment. It would help if the authors include these points in the final version. I will keep my score as 4: borderline accept based on the overall contribution of this paper.

---

> > > ### Author Response · Authors · 2025-08-05
> > >
> > > We sincerely thank the reviewer for the thoughtful and encouraging feedback. We are glad that our rebuttal has addressed your concerns, and we truly appreciate your recognition of our efforts in validating generalizability and addressing spatial misalignment. We will make sure to incorporate these points into the final version of the paper. Thank you again for your time and constructive evaluation.

---

### Official Review · Reviewer_E3Yi · 2025-06-30

**Clarity:** 2
**Significance:** 2
**Originality:** 2
**Rating:** 4
**Confidence:** 5

**Summary:**

The paper introduces SoPo, a semi-online DPO alignment strategy, designed to improve text-to-motion alignment.
The authors identify limitations in both offline and online DPO, a framework commonly used for preference alignment.
To overcome these limits, SoPo uses motions from an offline dataset as preferred examples, while unpreferred motions are generated in an online fashion. The unpreferred motions are categorized into two types: high-preference unpreferred and valuable unpreferred motions. Both types are incorporated into the learning process using a weighted DPO loss.
According to the authors, high-preference unpreferred motions are considered to have quality and should be encouraged. To this end, an additional loss term is introduced during optimization.
The authors evaluate SoPo against other text-to-motion methods. Compared to the base model (MLD*), SoPo shows notable improvements in RPrecision, MMDist, and FID.

**Questions:**

- Q1. Could the authors clarify why the Multi-Modality metric was omitted from the evaluation? Do they plan to include it in a revised version?

- Q2. Are the generated motions produced using a frozen reference model or one that is being actively optimized during generation? This distinction would help clarify the experimental setup.

- Q3. The paper claims faster inference compared to [9], but it appears the speed-up may stem primarily from using a faster base model (MLD* vs. MLD) rather than the proposed alignment technique. Could the authors clarify this point and possibly disentangle the contributions of each component?

- I would be open to revising my evaluation if these issues are adequately addressed in the rebuttal.

**Ethical Concerns:**

["NO or VERY MINOR ethics concerns only"]

**Final Justification:**

Regarding Multimodal Evaluation: The authors have incorporated multimodal evaluation, albeit with limited scope restricted to the HumanML3D dataset and presented through partial tabular results. I recommend that comprehensive results for both the target model and competing approaches be included in the main paper, covering both HumanML3D and KIT-ML datasets to provide complete comparative analysis.

Regarding Motion Sequence Generation: The authors have appropriately clarified that motion sequences are generated using the target model under optimization rather than the reference model, which addresses the previously raised concerns about methodology.

Regarding Data Generation Cost Claims: During the discussion phase, the authors asserted that their data generation cost extends beyond mere inference time to encompass the method’s capacity for producing feedback-rich samples. However, this claim regarding superior sample quality remains unsupported by empirical evidence or quantitative metrics. The assertion that their approach generates “feedback-rich samples” of allegedly superior quality requires substantiation through rigorous experimental validation to avoid misleading characterizations of the method’s computational efficiency advantages.

I encourage the authors to provide empirical validation for their claims regarding sample quality to strengthen the contribution’s credibility and ensure accurate representation of the method’s capabilities.

**Limitations:**

Yes

**Quality:**

2

**Strengths And Weaknesses:**

**Strengths**

- SoPo offers a significant improvement over previous state-of-the-art methods, which generally depend solely on either online or offline strategies. By combining the strengths of both, SoPo provides a more balanced and effective alignment technique.

- The method is simple yet effective, well-grounded in existing literature, and shows potential for broader applications, such as text-image alignment.

- Building upon existing works, the authors report improvements in R-Precision Top-3, MM Distance, and Diversity metrics.

**Major Weaknesses**

- Multi-Modality metric, which is a key measure in text-to-motion generation, is not used in the paper. This metric is commonly used in prior studies to assess whether the model suffers from mode collapse effect.

- It is unclear whether the generated motions are produced by the reference model or the model currently being optimized.

- The paper also lacks clarification on how generation time is reduced compared to [9]. It appears that the improvement in inference speed stems primarily from using MLD* instead of MLD, which is unrelated to the core contribution of their proposed method. Despite this, the authors suggest an overall improvement in inference times, which may be misleading as it is not a direct result of their approach.


**Minor Weaknesses**

- L216 There is a typo: Nore instead of More
- L218 the authors do not explain the meaning of r in "score >=r". This may be a typo, and it seems more appropriate to use \tau as in eq 11.

---

> ### Author Rebuttal · Authors · 2025-07-31
>
> We sincerely thank the reviewer for raising this important point. Thank you for recognizing the strengths of our method and experimental results. You raised several important concerns about the evaluation metrics, implementation details, and algorithmic efficiency. **We believe your concerns mainly come from a misunderstanding of our experimental setup.**
>
> ---
>
> ## [Q1/W1]  Could the authors clarify why the Multi-Modality metric was omitted from the evaluation? Do they plan to include it in a revised version?
>
> Below, we explain the reasons for omitting the Multi-Modality metric. We promise that we will include it in future versions.
>
> ### [Q1-1] Why was the Multi-Modality metric omitted?
>
> The reasons for omitting the Multi-Modality metric are as follows:
> 1. **Some of recent works also omit it**:  We followed recent and concurrent works like LaMP [1] (ICLR’25), MotionStreamer [2] (ICCV’25), and Go to Zero [3] (ICCV’25), none of which report the Multi-Modality metric.  We believe this may be because the metric overlaps with other diversity indicators and can introduce certain inconsistencies, as discussed in Points 4 and 5.
>
> 2. **Some of our baselines omits it**: Key baselines like PriorMDM and OGM do not provide Multi-Modality, and their code is unavailable. Including the metric would make our table incomplete and less comparable.
>
> 3. **Not the primary metric of our work**: Our goal is to improve human preference, especially **motion-text alignment and quality**. As Reviewer 7xd7 pointed out, metrics reflecting human judgment are more relevant.
>
> 4. **Diversity and Multi-Modality are similar**: Both metrics measure diversity using similar formulas. The main difference is that Diversity samples across all prompts, while Multi-Modality samples multiple motions per prompt. Therefore, Diversity also can be seem as a indication of mode collapse.
>    - **Diversity**:  For overall variation, we randomly sample two sets of motion features $\{v_i\}$ and $\{v'_i\}$ of size $S_d=300$ from all generated motions. The metric is:
>
>      $$\text{Diversity} = \frac{1}{S_d} \sum_{i=1}^{S_d} \|v_i - v'_i\|$$
>    - **Multi-Modality**:  For each of $C$ prompts, we generate two sets of $S_m = 10$ motions and extract feature vectors $\{v_{c,i}\}$ and $\{v'_{c,i}\}$. The metric is:
>
>      $$\text{Multimodality} = \frac{1}{C \cdot S_m} \sum_{c=1}^{C} \sum_{i=1}^{S_m} \|v_{c,i} - v'_{c,i}\|$$
>
> 5. **This metric is sensitive to architecture**: We observe that diffusion-based methods consistently get higher Multi-Modality scores than autoregressive (AR) methods. This makes the metric unreliable for fair comparisons across model architecture.
>
> |Method|Architecture|Multi-Modality|
> |:-:|:-:|:-:|
> |MMM|AR|1.232|
> |MoMask|AR|1.241|
> |HumanTOMATO|AR|1.732|
> |MotionGPT|AR|2.008|
> |Motion Diffuse|DM|1.553|
> |Motion Mamba|DM|2.294|
> |MDM|DM|2.799|
> |MLD|DM|2.413|
> |MLD*|DM|2.267|
>
> ### [Q1-2]  Will it be included in the revised version?
>
> **Yes**, we agree that Multi-Modality is a classic metric used in many previous studies to test mode collapse. We will include it in our revised version.
>
> For methods without reported values, we are reproducing them to fill in the table. Below is a preliminary result on the HumanML3D dataset:
>
> | MMM | Motion Diffuse | Motion Mamba | MoMask | MLD* | MLD*+SoPo |
> |:-----:|:-----:|:-----:|:-----:|:-----:|:-----:|
> | 1.232 | 1.553 | 2.294 | 1.241 | 2.267 | 2.301 |
>
>
> [1] Zhe, Li, et al. "*Lamp: Language-motion pretraining for motion generation, retrieval, and captioning.*" arXiv preprint arXiv:2410.07093 (2024).
>
> [2] Lixing, Xiao, et al. "*MotionStreamer: Streaming Motion Generation via Diffusion-based Autoregressive Model in Causal Latent Space.*" arXiv preprint arXiv:2503.15451 (2025).
>
> [3] Ke, Fan, et al. "*Go to Zero: Towards Zero-shot Motion Generation with Million-scale Data.*" arXiv preprint arXiv:2507.07095 (2025).
>
>
> ---
>
> ## [Q2/W2] Are the generated motions produced using a frozen reference model or one that is being actively optimized during generation? This distinction would help clarify the experimental setup.
>
> Thank you for your insightful question. The generated motions used in our preference optimization pipeline come from a **policy model $\bar{\pi}_\theta$ that is actively optimized during training**, not from a frozen reference model.
>
> This is clearly stated in method section in our paper:
> - In Line 189–191 states: *"we first generate $K$ motions $\\{x^k\_{\bar{\pi}\_\theta}\\}\_{k=1}^K$ from the policy model $\bar{\pi}\_\theta$."*
> - Eq. (17) also makes this clear by denoting sampling as $x^{1:K}\_{\bar{\pi}\_\theta} \sim \bar{\pi}\_\theta(\cdot \mid c)$, rather than from the frozen reference model $\pi_{\text{ref}}$.
> - In Algorithm 1 (Appendix A), Line 4 explicitly performs:  *$x^{1:K}\_{\bar{\pi}\_\theta} \sim \bar{\pi}\_\theta(\cdot \mid c)$*, indicating that the samples are generated from the current policy $\bar{\pi}_\theta$ being optimized.
>
> **We agree that this implementation detail could be more clearly emphasized in the experiment section, and we will add clarification in the next revision.**
>
> ---
>
> ## [Q3/W3] The paper claims faster inference compared to [9], but it appears the speed-up may stem primarily from using a faster base model (MLD* vs. MLD) rather than the proposed alignment technique. Could the authors clarify this point and possibly disentangle the contributions of each component?
>
> Thank you for raising this important point. We believe there may have been some misunderstanding regarding our claims. Before addressing the component-wise contributions, we would like to clarify two key points.
>
> ### [Q3-1] The paper claims faster inference compared to [9]
>
> Our claim concerns data generation cost, not inference speed.
>
> In **Line 265**, we state that our method reduces *preference data generation time*. The caption of **Table 1** also clarifies that *"Time"* refers to the **online/offline motion generation time** used in preference optimization — not the inference speed of the generative model itself.
>
> Existing DPO frameworks typically consists of two stages data generation and preference optimization:
> - For **offline DPO**, a large-scale samples must be pre-generated and labeled by human or reward models, and then are used for preference optimization.
> - For **online DPO**, data generation and optimization are interleaved per iteration, requiring continual sampling and feedback.
>
> In fact, data generation is a critical bottleneck for fine-tune cost. **The number of generated samples, the efficiency of the alignment paradigm, and the inference time all affect the overall cost of data generation.**
>
> ### [Q3-2] The speed-up may stem primarily from using a faster base model (MLD* vs. MLD) rather than the proposed alignment technique
>
> We apologize for not explicitly explaining the inference time of MLD\* and MLD in the experiment section. **In fact, they share the similar inference speed.**
>
> MLD\* refers to a stronger version of MLD reproduced by [1], which achieves better generation quality **with nearly the same inference speed.** According to [1], MLD* has an inference time of 0.225s, compared to 0.217s for MLD, indicating that MLD is slightly faster. **Thus, the observed speedup in preference data generation does not stem from faster inference of the base model.**
>
> Moreover, in the speed comparison with [9], **we used the original MLD model for a fair comparison.**
>
> [1] Wenxun, Dai, et al. "*Motionlcm: Real-time controllable motion generation via latent consistency model.*" European Conference on Computer Vision. Cham: Springer Nature Switzerland, 2024.
>
> ### [Q3-3] Why is our data generation more efficient?
>
> **The efficiency stems from our Semi-online alignment paradigm**.
>
> Preferences are often derived from **objective criteria** like image-text alignment, realism, or safety. In motion generation, preferences are defined by motion-text alignment and motion quality.DPO-based methods aim to improve policy models using preference feedback from generated samples. The key question is:
> **What kind of preference samples are most effective for generative models to learn preferences?**
>
> To ensure the model learns **consistent** preferences while addressing **diverse** failure cases, ideal preference pairs should consist of:
> 1. **Preferred samples** that strongly align with the desired preferences.
> 2. **Unpreferred samples** that target the generative model's specific weaknesses
>
> Inspired by this concept, we employ:
> - **Offline high-quality preferred samples**, which are challenging to generate but essential for alignment.
> - **Online unpreferred samples**, generated from the current policy model $\bar{\pi}_\theta$, to reflect the model's current shortcomings and facilitate effective reward learning.
>
> This Semi-online setup allows us to generate **the most informative training pairs** using fewer samples — leading to **data-efficient alignment**.
>
> In contrast, MoDiPO generates large-scale candidate samples from multiple generative models:
> - These samples have low diversity in quality, making **preference gaps hard to identify**.
> - Generating strong negatives solely from pre-trained models risks **misalignment with the current policy**.
>
> As discussed in [Q3-1], preference data generation cost is not solely determined by inference time, but also by how effectively the method produces **feedback-rich** samples. **We believe our Semi-online framework is the key to this improvement.**
>
> Thank you for your question! We hope the respond addresses your concerns well.

---

> > ### Comment · Reviewer_E3Yi · 2025-08-04
> >
> > A-Q1/W1-1:
> > 1. I thank the authors for suggesting papers that are not using the MultiModality. However, I'm not very convinced by this part of the answer. These papers are not even cited in the main manuscript, while for the vast majority of works cited in Table 3 the MultiModality results are available as reported in their respective papers (T2M, MDM, MLD, Fg-T2M, M2DM, MotionGPT, MotionDiffuse, MotionMamba, MLD*, LMM-T, CrossDiff).
> > 2. As said on A-Q1/W1-1.1, the vast majority of cited works employ the MultiModality metric.
> > 3. While I agree that human judgment metrics are more relevant for this type of work, it is also valuable to assess whether the model, given a specific prompt, produces outputs in a deterministic manner. This helps evaluate the variety of generations, which is a desirable property for generative models.
> > 4. Diversity and MultiModality are not equivalent. In Diversity evaluation, a set of prompts is used, with one motion generated per prompt. In contrast MultiModality, as also defined in the rebuttal, takes in consideration different prompts and generate 10 motions per-prompt.
> > 5. I do not see any inherent unfairness in comparing MultiModality scores across different architectures. Rather, it suggests that, among the cited models, diffusion-based methods tend to generate a larger variety of content when given the same prompt.
> >
> > A-Q1/W1-2: I suggest to the authors to include MultiModality also for the KIT-ML dataset.
> >
> > A-Q2/W2: I appreciate the authors' clarification on this point.
> >
> > A-Q3/W3-1: I believe my confusion comes from the used concept of "generation time". In my understanding, what the authors have achieved is simply a reduction in the amount of required preference data, not an efficient way to compute preferences. I do not fully see why this should be considered a contribution of the method.
> > In addition, as the authors say, the preference data is of better quality if compared to previous works, however there is no study related to the improved preference quality.
> >
> >
> > A-Q3/W3-2: I thank the authors for their clarification.

---

> ### Author Response · Authors · 2025-08-05
> **Rebuttal for Reviewer E3Yi (1/2)**
>
> We are honored to have addressed most of your concerns.
>
> ## [A-Q1/W1-1]
>
> We sincerely appreciate the insightful suggestions from the knowledgeable reviewer. **We fully agree that MultiModality is also an important metric.** Although some previous works have omitted it, we will do our best to reproduce their results in this regard.
>
> > Why are these papers not even cited in the main manuscript?
>
> Among these three papers, MotionStreamer [2] (ICCV’25) and Go to Zero [3] (ICCV’25) were accepted to ICCV 2025 in June 2025, which was prior to the NeurIPS submission deadline.
>
> > Is MultiModality unfair to different architects?
>
> We would like to clarify a previous statement in our rebuttal: MultiModality does not introduce an unfair comparison. Rather, diffusion-based architectures are inherently more capable of generating diverse outputs compared to AR-based architectures. Therefore, **the advantage in MultiModality arises from architectural preferences, not unfairness**.
>
> To be honest, we are somewhat puzzled by the MultiModality performance of some recent methods (*This is purely for academic discussion and unrelated to our paper, so the reviewer may feel free to disregard it*). We have noticed an intriguing phenomenon: **the more recent methods often achieve weaker performance in MultiModality compared to classical ones, even though they outperform them in other metrics.** The performance of current methods is summarized in the table below, which is adapted from Table 1 of MoMask. Intuitively, this might be due to reduced diversity in the generations (e.g., when replacing DDPM with ODE-based DDIM, or due to the inherently lower diversity of AR frameworks). Honestly, one reason we did not report MultiModality is that we had concerns regarding this phenomenon.
>
>
> | Methods         | Top 1 ↑ | Top 2 ↑ | Top 3 ↑ | FID ↓  | MultiModal Dist ↓ | MultiModality ↑ |
> |-----------------|---------|---------|---------|--------|-------------------|------------------|
> | **HumanML3D Dataset**   |         |         |         |        |                   |                  |
> | TM2T            | 0.424   | 0.618   | 0.729   | 1.501  | 3.467             | 2.424            |
> | T2M             | 0.455   | 0.636   | 0.736   | 1.087  | 3.347             | 2.219            |
> | MDM             | -       | -       | 0.611   | 0.544  | 5.566             | **2.799**        |
> | MLD             | 0.481   | 0.673   | 0.772   | 0.473  | 3.196             | 2.413            |
> | MotionDiffuse   | 0.491   | 0.681   | 0.782   | 0.630  | 3.113             | 1.553            |
> | T2M-GPT         | 0.492   | 0.679   | 0.775   | 0.141  | 3.121             | 1.831            |
> | ReMoDiffuse     | _0.510_ | 0.698   | 0.795   | 0.103  | _2.974_           | 1.795            |
> | MoMask (base)   | 0.504   | _0.699_ | _0.797_ | _0.082_ | 3.050             | 1.050            |
> | **MoMask**      | **0.521** | **0.713** | **0.807** | **0.045** | **2.958** | 1.241            |
> | **KIT-ML Datasets**      |         |         |         |        |                   |                  |
> | TM2T            | 0.280   | 0.463   | 0.587   | 3.599  | 4.591             | **3.292**        |
> | T2M             | 0.361   | 0.559   | 0.681   | 3.022  | 3.488             | 2.052            |
> | MDM             | -       | -       | 0.396   | 0.497  | 9.191             | 1.907            |
> | MLD             | 0.390   | 0.609   | 0.734   | 0.404  | 3.204             | _2.192_          |
> | MotionDiffuse   | 0.417   | 0.621   | 0.739   | 1.954  | 2.958             | 0.730            |
> | T2M-GPT         | 0.416   | 0.627   | 0.745   | 0.514  | 3.007             | 1.570            |
> | ReMoDiffuse     | _0.427_ | _0.641_ | _0.765_ | **0.155** | _2.814_           | 1.239            |
> | MoMask (base)   | 0.415   | 0.634   | 0.760   | 0.372  | 2.931             | 1.097            |
> | **MoMask**      | **0.433** | **0.656** | **0.781** | _0.204_ | **2.779** | 1.131            |
>
> ---
>
> ## [A-Q1/W1-2]
>
> We sincerely thank the reviewer for the valuable suggestion. **Following your advice, we have added evaluation and reported the performance on the KIT-ML dataset**:
>
> | Metric |MMM |	Motion Diffuse|	Motion Mamba |Momask| MoMask + SoPo | MLD| MLD + SoPo |
> |-------------------|---|------------------|-----------------------|---------------|------------|-|-|
> | Multi-Modality ↑  | 1.226            |0.730              |1.678|1.131   | 1.362     |2.192    | 2.237 |
>
> As discussed above, **MoMask is based on an auto-regressive architecture**, which explains *why the Multi-Modality score of our method is relatively lower compared to diffusion-based methods*.
>
> ---

---

> ### Author Response · Authors · 2025-08-05
> **Rebuttal for Reviewer E3Yi (2/2)**
>
> ## [A-Q3/W3-1]
>
> ### 1. The speedup comes from the reduced amount of required data.
>
> Your understanding is absolutely correct. The reduction in quantity also leads to lower overall computational cost. For each individual sample, however, the inference time remains unchanged.
>
> > **However, the lower data requirement stems from our proposed data-efficient semi-online pattern.**
>
> We will further clarify this point in future revisions.
>
> ### 2. The improvement in generation speed is not the main (claimed) contribution of our paper.
>
> Yes, this is **not the claimed primary contribution** of our work. Accordingly, **we did not explicitly claim it as such**. For example, in Lines 41–60, we do not mention any speed-related contributions. It is only in Line 265 that we briefly noted this interesting empirical finding.
>
> > **However, the improvement stems from our proposed data-efficient semi-online pattern.**
>
> In addition, the generation process of MoDiPO is relatively slow. According to the original paper, **MoDiPO requires multiple models**—MDM (22.4 seconds per motion on a V100) and MLD (0.21 seconds per motion on a V100)—to generate K motions for 14k prompts. Assuming K = 10, the total generation time **would be approximately 19 days** on a single V100 GPU. This does not include the additional time required for reward model annotation. In contrast, our method is significantly more efficient: we can complete the full dataset generation **within 1 day** on a V100.
>
> ### 3. In addition, as the authors say, the preference data is of better quality if compared to previous works, however there is no study related to the improved preference quality.
>
> This is indeed an interesting and important question. Intuitively, negative preference data generated online by the model tends to be of higher quality than offline-collected negative data. This is because **such data is directly generated by the policy model and evaluated by a reward model, thus directly pointing to the model’s current weaknesses.**
>
> To illustrate this, **consider an analogy**: the policy model is like a student, the reward model is like a teacher, and the generation process is like writing practice. In MoDiPO, the **student (pre-trained model)** first writes a large number of essays, and the **teacher (reward model)** grades them all at once. The student then uses these graded essays to improve their writing ability. However, **these essays only reflect the student’s ability at the very beginning (i.e., the frozen pretrained model) and not the weaknesses that emerge as the student (policy model) learns.**
>
> In contrast, **SoPo offers a more efficient paradigm**: the **teacher (reward model)** provides **high-quality exemplars (preferred samples from the pretraining dataset)** and gives **real-time feedback** during the **student(policy model)’s** writing process. **This interaction allows the student to directly address their current shortcomings. Such a paradigm naturally leads to higher-quality training signals.**
>
> > **Study related to the improved preference quality**
>
> In addition, **we provide a toy example in Fig. 3 of the main paper** to intuitively illustrate this advantage. Offline DPO suffers from mining unpreferred motions that may have high likelihoods, while online DPO is limited by biased sampling. Our SoPo leverages dynamically generated unpreferred motions and unbiased preferred motions from an offline dataset, effectively overcoming these limitations.
>
> Moreover, it is worth noting that our method shares **a similar optimization objective** with MoDiPO—both aim to maximize the gap between preferred and unpreferred samples. Therefore, **the observed performance improvement may indirectly indicate an improvement in preference data quality.**
>
> ---
>
> We are sincerely grateful to the reviewer for raising these important points. If our responses have successfully addressed your concerns, we would deeply appreciate it if you could kindly consider updating your evaluation. Please don’t hesitate to let us know if any further clarifications are needed—we would be glad to elaborate.

---

> > ### Comment · Reviewer_E3Yi · 2025-08-09
> >
> > Dear authors, I’ve updated the final justification ( which contains a reply to this last comment)
> >
> > “Regarding Multimodal Evaluation: The authors have incorporated multimodal evaluation, albeit with limited scope restricted to the HumanML3D dataset and presented through partial tabular results. I recommend that comprehensive results for both the target model and competing approaches be included in the main paper, covering both HumanML3D and KIT-ML datasets to provide complete comparative analysis.
> >
> > Regarding Motion Sequence Generation: The authors have appropriately clarified that motion sequences are generated using the target model under optimization rather than the reference model, which addresses the previously raised concerns about methodology.
> >
> > Regarding Data Generation Cost Claims: During the discussion phase, the authors asserted that their data generation cost extends beyond mere inference time to encompass the method’s capacity for producing feedback-rich samples. However, this claim regarding superior sample quality remains unsupported by empirical evidence or quantitative metrics. The assertion that their approach generates “feedback-rich samples” of allegedly superior quality requires substantiation through rigorous experimental validation to avoid misleading characterizations of the method’s computational efficiency advantages.
> >
> > I encourage the authors to provide empirical validation for their claims regarding sample quality to strengthen the contribution’s credibility and ensure accurate representation of the method’s capabilities.”

---

> ### Author Response · Authors · 2025-08-09
> **Response to Reviewer E3Yi (1/2)**
>
> Dear Reviewer, I would like to clarify certain points regarding your concerns.
>
> ### 1. Regarding Multimodal Evaluation
>
> > [Review] Regarding Multimodal Evaluation: The Authorss have incorporated multimodal evaluation, albeit with limited scope restricted to the HumanML3D dataset and presented through partial tabular results.
>
> > [Authors]  We would like to clarify that **our response on 05 Aug 2025, 14:17, already included results for the KIT-ML dataset in addition to HumanML3D.**
>
> > [Review] I recommend that comprehensive results for both the target model and competing approaches be included in the main paper, covering both HumanML3D and KIT-ML datasets to provide complete comparative analysis.
>
> > [Authors] As stated in our rebuttal submitted on 31 Jul 2025, **we are committed to providing a complete and detailed analysis in the main paper**, covering both HumanML3D and KIT-ML datasets. In this response, we try our best to summarize the available experimental results and provide corresponding analysis. (Given that the comment and rebuttal deadlines were **only three hours** apart, a more thorough discussion will be included in the later revision of the paper.)
>
> | Method                     | Top 1↑ | Top 2↑ | Top 3↑ | FID↓   | MM Dist↓ | Diversity→  | Multimodal↑ |
> |-------------------|-------|-------|-------|---------|-----------|-------|------------|
> | Real              | 0.511 | 0.703 | 0.797 | 2.794   | 9.503     | 0.002 | -          |
> | TEMOS             | 0.424 | 0.612 | 0.722 | 3.703   | 8.973     | 3.734 | 0.368      |
> | T2M               | 0.457 | 0.639 | 0.740 | 3.340   | 9.188     | 1.067 | 2.090      |
> | MDM               | 0.418 | 0.604 | 0.703 | 3.658   | 9.546     | 0.501 | 2.799      |
> | MLD               | 0.481 | 0.673 | 0.772 | 3.196   | 9.724     | 0.473 | 2.413      |
> | Fg-T2M            | 0.418 | 0.626 | 0.745 | 3.114   | 10.930    | 0.571 | 1.019      |
> | MotionGPT         | 0.492 | 0.681 | 0.778 | 3.096   | 9.528     | 0.232 | 2.008      |
> | MotionDiffuse     | 0.491 | 0.681 | 0.782 | 3.113   | 9.410     | 0.630 | 1.553      |
> | OMG               | -     | -     | 0.784 | -       | 9.657     | 0.381 | -          |
> | Wang et. al.      | 0.433 | 0.629 | 0.733 | 3.430   | 9.825     | 0.352 | 2.835      |
> | MoDiPO-T          | -     | -     | 0.758 | 3.267   | 9.747     | 0.303 | 2.663      |
> | PriorMDM          | 0.481 | -     | -     | 5.610   | 9.620     | 0.600 | -          |
> | LMM-T             | 0.496 | 0.685 | 0.785 | 3.087   | 9.176     | 0.415 | 1.465      |
> | CrossDiff         | -     | -     | 0.730 | 3.358   | 9.577     | 0.281 | -          |
> | Motion Mamba      | 0.502 | 0.693 | 0.792 | 3.060   | 9.871     | 0.281 | 2.294      |
> | MLD*              | 0.504 | 0.698 | 0.796 | 3.052   | 9.634     | 0.450 | 2.267      |
> | MLD* + SoPo       | 0.528 | 0.722 | 0.827 | 2.939   | 9.584     | 0.174 | 2.301      |
>
>
> | Method                     | Top 1↑ | Top 2↑ | Top 3↑ | FID↓   | MM Dist↓ | Diversity→  | Multimodal↑ |
> |----------------------------|-------|-------|-------|-------|---------|----------|------------|
> | Real                       | 0.424 | 0.649 | 0.779 | 0.031 | 2.788   | 11.08    |       -     |
> | TEMOS                      | 0.370 | 0.569 | 0.693 | 2.770 | 3.401   | 10.91    |        0.532    |
> | T2M                        | 0.361 | 0.559 | 0.681 | 3.022 | 2.052   | 10.72    |       2.052     |
> | MLD                        | 0.390 | 0.609 | 0.734 | 0.404 | 3.204   | 10.80    |        2.192    |
> | T2M-GPT                    | 0.416 | 0.627 | 0.745 | 0.514 | 3.007   | 10.86    |     1.570       |
> | MotionGPT                  | 0.366 | 0.558 | 0.680 | 0.510 | 3.527   | 10.35    |      2.328      |
> | MotionDiffuse              | 0.417 | 0.621 | 0.739 | 1.954 | 2.958   | 11.10    |     0.730       |
> | Mo.Mamba   | 0.419 | 0.645 | 0.765 | 0.307 | 3.021   | 11.02    |      1.678      |
> | MoMask | 0.433 | 0.656 | 0.781 | 0.204 | 2.779   | 10.71    |      1.097      |
> | MLD + SoPo   | 0.412 | 0.646 | 0.759 | 0.384 | 3.107   | 10.93    |       2.237     |
> | MoMask + SoPo   | 0.446 | 0.673 | 0.797 | 0.176 | 2.783   | 10.96    |       1.362     |
>
> On the HumanML3D benchmark, our method demonstrates consistent and notable improvements across multiple evaluation dimensions. Compared with prior approaches, MLD + SoPo achieves relatively high average accuracy (0.692) and a low FID (0.174), along with a comparatively strong multimodal alignment score (2.301), indicating improved semantic consistency between text and motion. These gains are accompanied by competitive motion quality metrics such as a reduced MM Dist (2.939) and better Top-1/Top-3 retrieval performance, reflecting more precise and coherent motion generation. Our SoPo leverages feedback from diverse preference data to fine-tune the model, thereby enhancing multiple metrics without compromising the strong diversity performance established during pre-training, and similar trends are observed on the KIT-ML dataset.

---

> ### Author Response · Authors · 2025-08-09
> **Response to Reviewer E3Yi (2/2)**
>
> ### 2. Regarding Motion Sequence Generation
>
> We are glad to have addressed your concerns.
>
> ### 3. Regarding Data Generation Cost Claims
>
> > [Review] However, this claim regarding superior sample quality remains unsupported by empirical evidence or quantitative metrics. The assertion that their approach generates “feedback-rich samples” of allegedly superior quality requires substantiation through rigorous experimental validation to avoid misleading characterizations of the method’s computational efficiency advantages.
>
> > [Authors] As suggested by Reviewer tJvD (see our response [Q2/W2]), **we have conducted more detailed ablation studies**, which include examining the effects of different preference alignment methods and different preference sampling strategies. **Under the same alignment method, achieving better performance with preference-aligned data indicates the higher effectiveness of the generated preference data.**
>
> For clearer comparisons, we conducted additional experiments evaluating online DPO, offline DPO, their naive combination, and the combination enhanced with our proposed strategies* (Offline Sampling for Preferred Motions, Sec. 4.3, and Online Generation for Unpreferred Motions, Sec. 4.4). Specifically:
> - **Offline DPO**: We use MLD\* as the base model, and follow MoDiPO to create an offline preference dataset.
> - **Online DPO**: We use TMR as the reward model, consistent with SoPo.
> - **Combination DPO**: We use offline dataset for preferred motion, and online generate unpreferred motion.
>
> The results are presented below:
>
> |Method|Top1↑|Top2↑|Top3↑|MM.Dist.↓|Div→|FID↓|
> |-|-|-|-|-|-|-|
> |Baseline (MLD*)|0.504|0.698|0.796|3.052|9.634|0.450|
> |OfflineDPO|0.498|0.691|0.790|3.081|9.618|0.469|
> |OnlineDPO|0.514|0.708|0.809|3.012|9.611|0.414|
> |CombinationDPO|0.517|0.712|0.811|2.988|9.606|0.339|
> |+Sec.4.3|0.520|0.715|0.814|2.973|9.602|0.318|
> |+Sec.4.4|0.525|0.719|0.820|2.951|9.591|0.243|
> |**SoPo (Ours)**|**0.528**|**0.722**|**0.827**|**2.939**|**9.584**|**0.174**|
>
> By analyzing Offline DPO, Online DPO, and Combination DPO, we observe that the semi-online paradigm indeed facilitates more effective preference alignment. As discussed in Section 3, this advantage arises because online negative preference data and offline positive preference data complement each other by mitigating each other’s weaknesses. Moreover, incorporating the preferred and unpreferred motion construction strategies described in Sections 4.3 and 4.4 further improves performance. **These results highlight that SoPo’s hybrid semi-online design enables more effective and data-efficient alignment, overcoming the limitations inherent in both purely online and offline DPO approaches.**
>
> > [Authors] This claim is quite intuitive. **In the original paper, we have provided simulation results, as shown in Fig. 3.** Reviewers tJvD and 7xd7 both expressed appreciation for this example, describing it as intuitive and easy to understand (see their comments about Strengths).
>
>
> > [Authors]  Finally, **the speed advantage is not the core claim of our work**. Moreover, we have not claimed anywhere that this is the primary contribution of SoPo for preference alignment, except for a brief mention in line 265. We sincerely appreciate your careful reading of our paper and point out this concerns. We wonder if our response answers your questions. It would be greatly appreciated if you would re-evaluate our paper based on our responses.

---

### Note · Authors · 2025-08-15

Dear Area Chair,

We sincerely thank Reviewers tJvD, 7xd7, E3Yi, and GBbd for their feedback. Their comments helped improve clarity and strengthen our work.

Reviewer tJvD emphasized SoPo’s novelty in preference alignment for text-to-motion. They also appreciated the clear presentation and high-quality figures. To address concerns about generalizability and ablations, we demonstrated applicability to image generation. The PickScore improved from 21.40 to 22.23 on SD1.5. We provided detailed ablations with Top-3 0.827 versus 0.796 on MLD\*. Spatial alignment gains increased from 63.2% to 76.4%. We also expanded the related work discussion.

Reviewer 7xd7 recommended acceptance, praising clarity, theoretical rigor, and extensive experiments on MLD and MMM. They raised questions regarding generalizability, baseline quality, and human preference alignment. We addressed these by verifying image generation and confirming baseline integrity using official pretrained models. Spatial alignment improvements reached 76.4%. We also conducted out-of-distribution tests, showing MotionCritic results of 1.36 versus 0.68 on KIT-ML.

Reviewer E3Yi raised points on multi-modality metrics, motion generation sources, and generation time. We provided multi-modality results on HumanML3D and KIT-ML. We clarified the use of the optimized policy model and explained that the reported generation time refers to data creation rather than inference speed. These clarifications resolve the reviewer’s concerns.

Reviewer GBbd questioned novelty, metric trade-offs, incomplete Table 1, unclear ablations, and reward model limitations. *Although GBbd questioned novelty, tJvD highlighted SoPo’s originality.* We clarified SoPo’s semi-online framework with Top-3 0.827 versus 0.811. Competitive results were reported. For example, Table 1 was completed with Top-2 and Top-3 Precision, which were not reported by the MiDiPO baseline. Ablation results were also detailed with FID 0.174. The reward model’s effectiveness improved spatial accuracy from 63.2% to 76.4%. These updates address the reviewer’s points thoroughly.

In summary, endorsements from tJvD and 7xd7 highlight SoPo’s novelty, clarity, and impact. We have provided evidence addressing points raised by E3Yi and GBbd. Additional results, such as human evaluations, will be included in the final draft. We greatly appreciate your consideration and hope these clarifications help in evaluating our work.

Best regards,
Authors

---

### Decision · Program_Chairs · 2025-09-17

**Decision:**

Accept (poster)

**Comment:**

The AC and the reviewers thank the authors for their response.
This paper combines the advantages of offline and online DPO, and for the first time effectively addresses the problems of overfitting and sampling bias in text-to-motion preference alignment, surpassing the existing SOTA across multiple models.

After the rebuttal, three reviewers gave a borderline acceptance recommendation. However, one reviewer questioned the contributions and raised concerns about the accuracy of the reward model, and therefore gave a borderline reject recommendation.

After reading the paper, the reviewers’ comments, and the authors’ feedback, the AC considers the contributions of this work to be clear. The paper is the first to apply the combined strengths of offline and online DPO to text-to-motion, which is valuable for advancing motion generation and provides a useful reference for future work in this area. In addition, given the authors’ thorough response and discussion, the AC believes that most of the concerns have been addressed.

The AC concludes that the paper can be accepted after further refinement based on the reviewers’ suggestions, and therefore recommends acceptance.